# Credit Assignment in Neural Networks through Deep Feedback Control

**Alexander Meulemans,**\* **Matilde Tristany Farinha**\*, **Javier García Ordóñez,**
**Pau Vilimelis Aceituno, João Sacramento, Benjamin F. Grewe**

Institute of Neuroinformatics, University of Zürich and ETH Zürich
`ameulema@ethz.ch`

## Abstract

The success of deep learning sparked interest in whether the brain learns by using similar techniques for assigning credit to each synaptic weight for its contribution to the network output. However, the majority of current attempts at biologically-plausible learning methods are either non-local in time, require highly specific connectivity motifs, or have no clear link to any known mathematical optimization method. Here, we introduce Deep Feedback Control (DFC), a new learning method that uses a feedback controller to drive a deep neural network to match a desired output target and whose control signal can be used for credit assignment. The resulting learning rule is fully local in space and time and approximates Gauss-Newton optimization for a wide range of feedback connectivity patterns. To further underline its biological plausibility, we relate DFC to a multi-compartment model of cortical pyramidal neurons with a local voltage-dependent synaptic plasticity rule, consistent with recent theories of dendritic processing. By combining dynamical system theory with mathematical optimization theory, we provide a strong theoretical foundation for DFC that we corroborate with detailed results on toy experiments and standard computer-vision benchmarks.

## 1 Introduction

The error backpropagation (BP) algorithm [1, 2, 3] is currently the gold standard to perform credit assignment (CA) in deep neural networks. Although deep learning was inspired by biological neural networks, an exact mapping of BP onto biology to explain learning in the brain leads to several inconsistencies with experimental results that are not yet fully addressed [4, 5, 6]. First, BP requires an exact symmetry between the weights of the forward and feedback pathways [5, 6], also called the weight transport problem. Another issue of relevance is that, in biological networks, feedback also changes each neuron's activation and thus its immediate output [7, 8], which does not occur in BP.

Lillicrap et al. [9] convincingly showed that the weight transport problem can be sidestepped in modest supervised learning problems by using random feedback connections. However, follow-up studies indicated that random feedback paths cannot provide precise CA in more complex problems [10, 11, 12, 13], which can be mitigated by learning feedback weights that align with the forward pathway [14, 15, 16, 17, 18] or approximate its inverse [19, 20, 21, 22]. However, this precise alignment imposes strict constraints on the feedback weights, whereas more flexible constraints could provide the freedom to use feedback also for other purposes besides learning, such as attention and prediction [8].

A complementary line of research proposes models of cortical microcircuits which propagate CA signals through the network using dynamic feedback [23, 24, 25] or multiplexed neural codes [26], thereby directly influencing neural activations with feedback. However, these models introduce

---

\*Equal contribution

35th Conference on Neural Information Processing Systems (NeurIPS 2021).

highly specific connectivity motifs and tightly coordinated plasticity mechanisms. Whether these constraints can be fulfilled by cortical networks is an interesting experimental question. Another line of work uses adaptive control theory [27] to derive learning rules for non-hierarchical recurrent neural networks (RNNs) based on error feedback, which drives neural activity to track a reference output [28, 29, 30, 31]. These methods have so far only been used to train single-layer RNNs with fixed output and feedback weights, making it unclear whether they can be extended to deep neural networks. Finally, two recent studies [32, 33] use error feedback in a dynamical setting to invert the forward pathway, thereby enabling errors to flow backward. These approaches rely on a learning rule that is non-local in time and it remains unclear whether they approximate any known optimization method. Addressing the latter, two recent studies take a first step by relating learned (non-dynamical) inverses of the forward pathway [21] and iterative inverses restricted to invertible networks [22] to approximate Gauss-Newton optimization.

Inspired by the Dynamic Inversion method [32], we introduce Deep Feedback Control (DFC), a new biologically-plausible CA method that addresses the above-mentioned limitations and extends the control theory approach to learning [28, 29, 30, 31] to deep neural networks. DFC uses a feedback controller that drives a deep neural network to match a desired output target. For learning, DFC then simply uses the dynamic change in the neuron activations to update their synaptic weights, resulting in a learning rule fully local in space and time. We show that DFC approximates Gauss-Newton (GN) optimization and therefore provides a fundamentally different approach to CA compared to BP. Furthermore, DFC does not require precise alignment between forward and feedback weights, nor does it rely on highly specific connectivity motifs. Interestingly, the neuron model used by DFC can be closely connected to recent multi-compartment models of cortical pyramidal neurons. Finally, we provide detailed experimental results, corroborating our theoretical contributions and showing that DFC does principled CA on standard computer-vision benchmarks in a way that fundamentally differs from standard BP.

## 2 The Deep Feedback Control method

Here, we introduce the core parts of DFC. In contrast to conventional feedforward neural network models, DFC makes use of a dynamical neuron model (Section 2.1). We use a feedback controller to drive the neurons of the network to match a desired output target (Section 2.2), while simultaneously updating the synaptic weights using the change in neuronal activities (Section 2.3). This combination of dynamical neurons and controller leads to a simple but powerful learning method, that is linked to GN optimization and offers a flexible range of feedback connectivity (see Section 3).

### 2.1 Neuron and network dynamics

The first main component of DFC is a dynamical multilayer network, in which every neuron integrates its forward and feedback inputs according to the following dynamics:

$$\tau_v \frac{\mathrm{d}}{\mathrm{d}t} \mathbf{v}_i(t) = -\mathbf{v}_i(t) + W_i \phi\big(\mathbf{v}_{i-1}(t)\big) + Q_i \mathbf{u}(t) \quad 1 \leq i \leq L, \tag{1}$$

with $\mathbf{v}_i$ a vector containing the pre-nonlinearity activations of the neurons in layer $i$, $W_i$ the forward weight matrix, $\phi$ a smooth nonlinearity, $\mathbf{u}$ a feedback input, $Q_i$ the feedback weight matrix, and $\tau_v$ a time constant. See Fig. 1B for a schematic representation of the network. To simplify notation, we define $\mathbf{r}_i = \phi(\mathbf{v}_i)$ as the post-nonlinearity activations of layer $i$. The input $\mathbf{r}_0$ remains fixed throughout the dynamics (1). Note that in the absence of feedback, i.e., $\mathbf{u} = 0$, the equilibrium state of the network dynamics (1) corresponds to a conventional multilayer feedforward network state, which we denote with superscript '-':

$$\mathbf{r}_i^- = \phi(\mathbf{v}_i^-) = \phi(W_i \mathbf{r}_{i-1}^-), \quad 1 \leq i \leq L, \quad \text{with } \mathbf{r}_0^- = \mathbf{r}_0. \tag{2}$$

### 2.2 Feedback controller

The second core component of DFC is a feedback controller, which is only active during learning. Instead of a single backward pass for providing feedback, DFC uses a feedback controller to continuously drive the network to an output target $\mathbf{r}_L^*$ (see Fig. 1D). Following the Target Propagation framework [20, 21, 22], we define $\mathbf{r}_L^*$ as the feedforward output nudged towards lower loss:

$$\mathbf{r}_L^* \triangleq \mathbf{r}_L^- - \lambda \frac{\partial \mathcal{L}(\mathbf{r}_L, \mathbf{y})}{\partial \mathbf{r}_L}\bigg|_{\mathbf{r}_L = \mathbf{r}_L^-} = \mathbf{r}_L^- + \boldsymbol{\delta}_L, \tag{3}$$

with $\mathcal{L}(\mathbf{r}_L, \mathbf{y})$ a supervised loss function defining the task, $\mathbf{y}$ the label of the training sample, $\lambda$ a stepsize, and $\boldsymbol{\delta}_L$ shorthand notation. Note that (3) only needs the easily obtained loss gradient w.r.t. the output, e.g., for an $L^2$ output loss, one obtains the convex combination $\mathbf{r}_L^* = (1 - 2\lambda)\mathbf{r}_L^- + 2\lambda\mathbf{y}$.

The feedback controller produces a feedback signal $\mathbf{u}(t)$ to drive the network output $\mathbf{r}_L(t)$ towards its target $\mathbf{r}_L^*$, using the control error $\mathbf{e}(t) \triangleq \mathbf{r}_L^* - \mathbf{r}_L(t)$. A standard approach in designing a feedback controller is the Proportional-Integral-Derivative (PID) framework [34]. While DFC is compatible with various controller types, such as a full PID controller or a pure proportional controller (see Appendix A.8), we use a PI controller for a combination of simplicity and good performance, resulting in the following controller dynamics (see also Fig. 1A):

$$\mathbf{u}(t) = K_I \mathbf{u}^{\text{int}}(t) + K_P \mathbf{e}(t), \quad \tau_u \frac{\mathrm{d}}{\mathrm{d}t} \mathbf{u}^{\text{int}}(t) = \mathbf{e}(t) - \alpha \mathbf{u}^{\text{int}}(t), \tag{4}$$

where a leakage term is added to constrain the magnitude of $\mathbf{u}^{\text{int}}$. For mathematical simplicity, we take the control matrices equal to $K_I = I$ and $K_P = k_p I$ with $k_p \geq 0$ the proportional control constant. This PI controller adds a leaky integration of the error $\mathbf{u}^{\text{int}}$ to a scaled version of the error $k_p \mathbf{e}$ which could be implemented by a dedicated neural microcircuit (for a discussion see App. I). Drawing inspiration from the Target Propagation framework [19, 20, 21, 22] and the Dynamic Inversion framework [32], one can think of the controller and network dynamics as performing a *dynamic inversion* of the output target $\mathbf{r}_L^*$ towards the hidden layers, as the controller dynamically changes the activation of the hidden layers until the output target is reached.

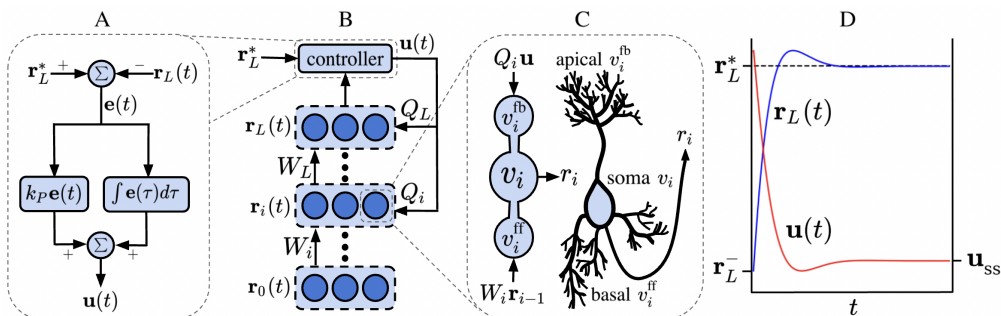

Figure 1: (A) A block diagram of the controller, where we omitted the leakage term of the integral controller. (B) Schematic illustration of DFC. (C) Schematic illustration of the multi-compartment neuron used by DFC, compared to a cortical pyramidal neuron sketch (see also Discussion). (D) Illustration of the output $\mathbf{r}_L(t)$ and the controller dynamics $\mathbf{u}(t)$ in DFC.

## 2.3 Forward weight updates

The update rule for the feedforward weights has the form:

$$\tau_W \frac{\mathrm{d}}{\mathrm{d}t} W_i(t) = \left( \phi(\mathbf{v}_i(t)) - \phi(W_i \mathbf{r}_{i-1}(t)) \right) \mathbf{r}_{i-1}(t)^T. \tag{5}$$

This learning rule simply compares the neuron's controlled activation to its current feedforward input and is thus local in space and time. Furthermore, it can be interpreted most naturally by compartmentalizing the neuron into the central compartment $\mathbf{v}_i$ from (1) and a feedforward compartment $\mathbf{v}_i^{\text{ff}} \triangleq W_i \mathbf{r}_{i-1}$ that integrates the feedforward input. Now, the forward weight dynamics (5) represents a delta rule using the difference between the actual firing rate of the neuron, $\phi(\mathbf{v}_i)$, and its estimated firing rate, $\phi(\mathbf{v}_i^{\text{ff}})$, based on the feedforward inputs. Note that we assume $\tau_W$ to be a large time constant, such that the network (1) and controller dynamics (4) are not influenced by the weight dynamics, i.e., the weights are considered fixed in the timescale of the controller and network dynamics.

In Section 5, we show how the feedback weights $Q_i$ can also be learned locally in time and space for supporting the stability of the network dynamics and the learning of $W_i$. This feedback learning rule needs a feedback compartment $\mathbf{v}_i^{\text{fb}} \triangleq Q_i \mathbf{u}$, leading to the *three-compartment neuron* schematized in Fig. 1C, inspired by recent multi-compartment models of the pyramidal neuron (see Discussion). Now, that we introduced the DFC model, we will show that (i) the weight updates (5) can properly optimize a loss function (Section 3), (ii) the resulting dynamical system is stable under certain conditions (Section 4), and (iii) learning the feedback weights facilitates (i) and (ii) (Section 5).

## 3 Learning theory

To understand how DFC optimizes the feedforward mapping (2) on a given loss function, we link the weight updates (5) to mathematical optimization theory. We start by showing that DFC dynamically inverts the output error to the hidden layers (Section 3.1), which we link to GN optimization under flexible constraints on the feedback weights $Q_i$ and on layer activations (Section 3.2). In Section 3.3, we relax some of these constraints, and show that DFC still does principled optimization by using *minimum norm* (MN) updates for $W_i$. During this learning theory section, we assume stable dynamics, which we investigate in more detail in Section 4. All theoretical results of this section are tailored towards a PI controller, and they can be easily extended to pure proportional or integral control (see App. A.8).

### 3.1 DFC dynamically inverts the output error

To understand how the weight update (5) can access error information, we start by investigating the steady state of the network dynamics (1) and the controller dynamics (4), assuming that all weights are fixed (hence, a separation of timescales). As the feedback controller controls all layers simultaneously, we introduce a compact notation for: concatenated neuron activations $\mathbf{v} \triangleq [\mathbf{v}_1^T, ..., \mathbf{v}_L^T]^T$, feedforward compartments $\mathbf{v}^{\text{ff}} \triangleq [\mathbf{v}_1^{\text{ff},T}, ..., \mathbf{v}_L^{\text{ff},T}]^T$, and feedback weights $Q \triangleq [Q_1^T ... Q_L^T]^T$. Lemma 1 shows a first-order Taylor approximation of the steady-state solution (full proof in App. A.1).

**Lemma 1.** *Assuming stable dynamics, a small target stepsize $\lambda$, and $W_i$ and $Q_i$ fixed, the steady-state solutions of the dynamical systems* (1) *and* (4) *can be approximated by:*

$$\mathbf{u}_{\text{ss}} = (JQ + \tilde{\alpha}I)^{-1}\boldsymbol{\delta}_L + \mathcal{O}(\lambda^2), \quad \mathbf{v}_{\text{ss}} = \mathbf{v}_{\text{ss}}^{\text{ff}} + Q(JQ + \tilde{\alpha}I)^{-1}\boldsymbol{\delta}_L + \mathcal{O}(\lambda^2), \quad (6)$$

*with $J \triangleq \left[ \frac{\partial \mathbf{r}_L^-}{\partial \mathbf{v}_1}, ..., \frac{\partial \mathbf{r}_L^-}{\partial \mathbf{v}_L} \right]\Big|_{\mathbf{v}=\mathbf{v}^-}$ the Jacobian of the network output w.r.t. all $\mathbf{v}_i$, evaluated at the network equilibrium without feedback, $\boldsymbol{\delta}_L$ the output error as defined in (3), $\mathbf{v}_{i,ss}^{\text{ff}} = W_i\phi(\mathbf{v}_{i-1,\text{ss}})$, and $\tilde{\alpha} = \alpha/(1 + \alpha k_p)$.*

To get a better intuition of what this steady state represents, consider the scenario where we want to nudge the network activation $\mathbf{v}$ with $\Delta\mathbf{v}$, i.e., $\mathbf{v}_{\text{ss}} = \mathbf{v}_{\text{ss}}^{\text{ff}} + \Delta\mathbf{v}$, such that the steady-state network output equals its target $\mathbf{r}_L^*$. With linearized network dynamics, this results in solving the linear system $J\Delta\mathbf{v} = \boldsymbol{\delta}_L$. As $\Delta\mathbf{v}$ is of much higher dimension than $\boldsymbol{\delta}_L$, this is an underdetermined system with infinitely many solutions. Constraining the solution to the column space of $Q$ leads to the unique solution $\Delta\mathbf{v} = Q(JQ)^{-1}\boldsymbol{\delta}_L$, corresponding to the steady-state solution in Lemma 1 minus a small damping constant $\tilde{\alpha}$. Hence, similar to Podlaski and Machens [32], through an interplay between the network and controller dynamics, the controller *dynamically inverts* the output error $\boldsymbol{\delta}_L$ to produce feedback that exactly drives the network output to its desired target.

### 3.2 DFC approximates Gauss-Newton optimization

To understand the optimization characteristics of DFC, we show that under flexible conditions on $Q_i$ and the layer activations, DFC approximates GN optimization. We first briefly review GN optimization and introduce two conditions needed for the main theorem.

**Gauss-Newton optimization** [35] is an approximate second-order optimization method used in nonlinear least-squares regression. The GN update for the model parameters $\boldsymbol{\theta}$ is computed as:

$$\Delta\boldsymbol{\theta} = J_\theta^\dagger \mathbf{e}_L, \quad (7)$$

with $J_\theta$ the Jacobian of the model output w.r.t. $\boldsymbol{\theta}$ concatenated for all minibatch samples, $J_\theta^\dagger$ its Moore-Penrose pseudoinverse, and $\mathbf{e}_L$ the output errors.

**Condition 1.** *Each layer of the network, except from the output layer, has the same activation norm:*

$$\|\mathbf{r}_0\|_2 = \|\mathbf{r}_1\|_2 = ...\|\mathbf{r}_{L-1}\|_2 \triangleq \|\mathbf{r}\|_2. \tag{8}$$

Note that the latter condition considers a statistic $\|\mathbf{r}_i\|_2$ of a whole layer and does not impose specific constraints on single neural firing rates. This condition can be interpreted as each layer, except the output layer, having the same 'energy budget' for firing.

**Condition 2.** *The column space of $Q$ is equal to the row space of $J$.*

This more abstract condition imposes a flexible constraint on the feedback weights $Q_i$, that generalizes common learning rules with direct feedback connections [16, 21]. For instance, besides $Q = J^T$ (BP; [16]) and $Q = J^\dagger$ [21], many other instances of $Q$ which have not yet been explored in the literature fulfill Condition 2 (see Fig. 2), hence leading to principled optimization (see Theorem 2). With these conditions in place, we are ready to state the main theorem of this section (full proof in App. A).

**Theorem 2.** *Assuming Conditions 1 and 2 hold, $J$ is full rank, the task loss $\mathcal{L}$ is a $L^2$ loss, and $\lambda, \alpha \to 0$, then the following steady-state (ss) updates for the forward weights,*

$$\Delta W_{i,\text{ss}} = \eta(\mathbf{v}_{i,\text{ss}} - \mathbf{v}_{i,\text{ss}}^{\text{ff}})\mathbf{r}_{i-1,\text{ss}}^T , \tag{9}$$

*with $\eta$ a stepsize parameter, align with the weight updates for $W_i$ for the feedforward network* (2) *prescribed by the GN optimization method with a minibatch size of 1.*

In this theorem, we need Condition 2 such that the dynamical inversion $Q(JQ)^{-1}$ (6) equals the pseudoinverse of $J$ and we need Condition 1 to extend this pseudoinverse to the Jacobian of the output w.r.t. the network weights, as in eq. (7). Theorem 2 links the DFC method to GN optimization, thereby showing that it does principled optimization, while being fundamentally different from BP. In contrast to recent work that connects target propagation to GN [21, 22], we do not need to approximate the GN curvature matrix by a block-diagonal matrix but use the full curvature instead. Hence, one can use Theorem 2 in Cai et al. [36] to obtain convergence results for this setting of GN with a minibatch size of 1, in highly overparameterized networks. Strikingly, the feedback path of DFC does not need to align with the forward path or its inverse to provide optimally aligned weight updates with GN, as long as it satisfies the flexible Condition 2 (see Fig. 2).

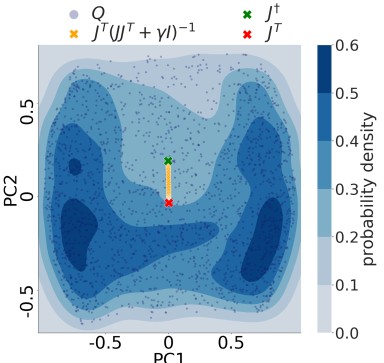

Figure 2: Randomly generated feedback matrices $Q$ (blue) that satisfy Conditions 2 and 3, and have unity norm, visualized by a principal component analysis, with density contours added for visual clarity. $J^T$, $J^\dagger$, and $J^T(JJ^T + \gamma I)^{-1}$, $\gamma \in [10^{-5}, 10^2]$, are added, highlighting that the optimal feedback configurations for DFC (blue) span a much wider space compared to conventional CA methods.

The steady-state updates (9) used in Theorem 2 differ from the actual updates (5) in two nuanced ways. First, the plasticity rule (5) uses a nonlinearity, $\phi$, of the compartment activations, whereas in Theorem 2 this nonlinearity is not included. There are two reasons for this: (i) the use of $\phi$ in (5) can be linked to specific biophysical mechanisms in the pyramidal cell [37] (see Discussion), and (ii) using $\phi$ makes sure that saturated neurons do not update their forward weights, which leads to better performance (see App. A.6). Second, in Theorem 2, the weights are only updated at steady state, whereas in (5) they are continuously updated during the dynamics of the network and controller. Before settling rapidly, the dynamics oscillate around the steady-state value (see Fig. 1D), and hence, the accumulated continuous updates (5) will be approximately equal to its steady-state equivalent, since the oscillations approximately cancel each other out and the steady state is quickly reached (see Section 6.1 and App. A.7). Theorem 2 needs a $L^2$ loss function and Condition 1 and 2 to hold for linking DFC with GN. In the following subsection, we relax these assumptions and show that DFC still does principled optimization.

### 3.3 DFC uses weighted minimum norm updates

GN optimization with a minibatch size of 1 is equivalent to MN updates [21], i.e., it computes the smallest possible weight update such that the network exactly reaches the current output target after

the update. These MN updates can be generalized to weighted MN updates for targets using arbitrary loss functions. The following theorem shows the connection between DFC and these weighted MN updates, while removing the need for Condition 1 and an $L^2$ loss (full proof in App. A).

**Theorem 3.** *Assuming stable dynamics, Condition 2 holds and $\lambda, \alpha \to 0$, the steady-state weight updates* (9) *are proportional to the weighted MN updates of $W_i$ for letting the feedforward output $\mathbf{r}_L^-$ reach $\mathbf{r}_L^*$, i.e., the solution to the following optimization problem:*

$$\underset{\Delta W_i, i \in [1,..,L]}{\arg\min} \quad \sum_{i=1}^{L} \|\mathbf{r}_{i-1}^{-(m)}\|_2^2 \|\Delta W_i\|_F^2 \qquad s.t. \quad \mathbf{r}_L^{-(m+1)} = \mathbf{r}_L^{*(m)}, \tag{10}$$

*with $m$ the iteration and $\mathbf{r}_L^{-(m+1)}$ the network output without feedback after the weight update.*

Theorem 3 shows that Condition 2 enables the controller to drive the network towards its target $\mathbf{r}_L^*$ with MN activation changes, $\Delta \mathbf{v} = \mathbf{v} - \mathbf{v}^{\text{ff}}$, which combined with the steady-state weight update (9) result in weighted MN updates $\Delta W_i$ (see also App. A.4). When the feedback weights do not have the correct column space, the weight updates will not be MN. Nevertheless, the following proposition shows that the weight updates still follow a descent direction given arbitrary feedback weights.

**Proposition 4.** *Assuming stable dynamics and $\lambda, \alpha \to 0$, the steady-state weight updates* (9) *with a layer-specific learning rate $\eta_i = \eta / \|r_{i-1}\|_2^2$ lie within 90 degrees of the loss gradient direction.*

## 4 Stability of DFC

Until now, we assumed that the network dynamics are stable, which is necessary for DFC, as an unstable network will diverge, making learning impossible. In this section, we investigate the conditions on the feedback weights $Q_i$ necessary for stability. To gain intuition, we linearize the network around its feedforward values, assume a separation of timescales between the controller and the network ($\tau_u \gg \tau_v$), and only consider integrative control ($k_p = 0$). This results in the following dynamics (see App. B for the derivation):

$$\tau_u \frac{\mathrm{d}}{\mathrm{d}t} \mathbf{u}(t) = -(JQ + \alpha I)\mathbf{u}(t) + \delta_L. \tag{11}$$

Hence, in this simplified case, the local stability of the network around the equilibrium point depends on the eigenvalues of $JQ$, which is formalized in the following condition and proposition.

**Condition 3.** *Given the network Jacobian evaluated at the steady state, $J_{\text{ss}} \triangleq \left[ \frac{\partial \mathbf{r}_L^-}{\partial \mathbf{v}_1}, ..., \frac{\partial \mathbf{r}_L^-}{\partial \mathbf{v}_L} \right]\Big|_{\mathbf{v}=\mathbf{v}_{\text{ss}}}$, the real parts of the eigenvalues of $J_{\text{ss}}Q$ are all greater than $-\alpha$.*

**Proposition 5.** *Assuming $\tau_u \gg \tau_v$ and $k_p = 0$, the network and controller dynamics are locally asymptotically stable around its equilibrium iff Condition 3 holds.*

This proposition follows directly from Lyapunov's Indirect Method [38]. When assuming the more general case where $\tau_v$ is not negligible and $k_p > 0$, the stability criteria quickly become less interpretable (see App. B). However, experimentally, we see that Condition 3 is a good proxy condition for guaranteeing stability in the general case where $\tau_v$ is not negligible and $k_p > 0$ (see Section 6 and App. B).

## 5 Learning the feedback weights

Condition 2 and 3 emphasize the importance of the feedback weights for enabling efficient learning and ensuring stability of the network dynamics, respectively. As the forward weights, and hence the network Jacobian, $J$, change during training, the set of feedback configurations that satisfy Conditions 2 and 3 also change. This creates the need to adapt the feedback weights accordingly to ensure efficient learning and network stability. We solve this challenge by learning the feedback weights, such that they can adapt to the changing network during training. We separate forward and feedback weight training in alternating wake-sleep phases [39]. Note that in practice, a fast alternation between the two phases is not required (see Section 6).

Inspired by the Weight Mirror method [14], we learn the feedback weights by inserting independent zero-mean noise $\epsilon$ in the system dynamics:

$$\tau_v \frac{\mathrm{d}}{\mathrm{d}t} \mathbf{v}_i(t) = -\mathbf{v}_i(t) + W_i \phi(\mathbf{v}_{i-1}(t)) + Q_i \mathbf{u}(t) + \sigma \boldsymbol{\epsilon}_i. \tag{12}$$

The noise fluctuations propagated to the output carry information from the network Jacobian, $J$. To let $\mathbf{e}$, and hence $\mathbf{u}$, incorporate this noise information, we set the output target $\mathbf{r}_L^*$ to the average network output $\mathbf{r}_L^-$. As the network is continuously perturbed by noise, the controller will try to counteract the noise and regulate the network towards the output target $\mathbf{r}_L^-$. The feedback weights can then be trained with a simple anti-Hebbian plasticity rule with weight decay, which is local in space and time:

$$\tau_Q \frac{\mathrm{d}}{\mathrm{d}t} Q_i(t) = -\mathbf{v}_i^{\mathrm{fb}}(t)\mathbf{u}(t)^T - \beta Q_i, \tag{13}$$

where $\beta$ is the scale factor of the weight decay term and where we assume that a subset of the noise input $\boldsymbol{\epsilon}_i$ enters through the feedback compartment, i.e., $\mathbf{v}_i^{\mathrm{fb}} = Q_i\mathbf{u} + \sigma_{\mathrm{fb}}\boldsymbol{\epsilon}_i^{\mathrm{fb}}$. The correlation between the noise in $\mathbf{v}_i^{\mathrm{fb}}$ and noise fluctuations in $\mathbf{u}$ provides the teaching signal for $Q_i$. Theorem 6 shows under simplifying assumptions that the feedback learning rule (13) drives $Q_i$ to satisfy Condition 2 and 3 (see App. C for the full theorem and its proof).

**Theorem 6** (Short version). *Assume a separation of timescales $\tau_v \ll \tau_u \ll \tau_Q$, $\alpha$ big, $k_p = 0$, $\mathbf{r}_L^* = \mathbf{r}_L^-$, and Condition 3 holds. Then, for a fixed input sample and $\sigma \to 0$, the first moment of $Q$ converges approximately to:*

$$\lim_{\sigma \to 0} \mathbb{E}[Q_{\mathrm{ss}}] \underset{\sim}{\propto} J^T(JJ^T + \gamma I)^{-1}, \tag{14}$$

*for some $\gamma > 0$. Furthermore, $\mathbb{E}[Q_{\mathrm{ss}}]$ satisfies Conditions 2 and 3, even if $\alpha = 0$ in the latter.*

Theorem 6 shows that under simplifying assumptions, $Q$ converges towards a damped pseudoinverse of $J$, which satisfies Conditions 2 and 3. Empirically, we see that this also approximately holds for more general settings where $\tau_v$ is not negligible, $k_p > 0$, and small $\alpha$ (see Section 6 and App. C).

The above theorem leaves two questions unanswered. First, it assumes that Condition 3 holds, however, the task of the feedback weight training is to make unstable network dynamics stable, resulting in a chicken-and-egg problem. The solution we use is to take $\alpha$ big enough to make the network stable during early training, after which the feedback weights align according to (14) and $\alpha$ can be decreased. Second, Theorem 6 considers training the feedback weights to convergence over one fixed input sample. However, in reality many different input samples will be considered during learning. When the network is linear, $J$ is the same for each input sample and eq. (14) holds exactly. However, for nonlinear networks, $J$ will be different for each sample, causing the feedback weights to align with an average of $J^T(JJ^T + \gamma I)^{-1}$ over many samples.

## 6 Experiments

We evaluate DFC in detail on toy experiments to showcase that our theoretical results translate to practice (Section 6.1) and on a modest range of computer vision benchmarks – MNIST classification and autoencoding [40], and Fashion MNIST classification [41] – to show that DFC can do precise CA in more challenging settings (Section 6.2). Alongside DFC, we test two variants: (i) **DFC-SS** which only updates its feedforward weights $W_i$ after the steady state (SS) of (1) and (4) is reached; and (ii) **DFC-SSA** which analytically computes the linearized steady state of (1) and (4) according to Lemma 1. To investigate whether learning the feedback weights is crucial for DFC, we compare for all three settings: (i) learning the feedback weights $Q_i$ according to (13); and (ii) fixing the feedback weights to the initialization $Q_i = \prod_{k=i+1}^{L} W_k^T$, which approximately satisfies Condition 2 and 3 at the beginning of training (see App. F), denoted with suffix (**fixed**). For the former, we pre-train the feedback weights according to (13) to ensure stability. During training, we iterate between 1 epoch of forward weight training and $X$ epochs of feedback weight training (if applicable), where $X \in [1, 2, 3]$ is a hyperparameter. We compare all variants to Direct Feedback Alignment (**DFA**) [42] as a control for direct feedback connectivity. DFC is simulated with the Euler-Maruyama method, which is the equivalent of forward Euler for stochastic differential equations [43]. We initialize the network to its feedforward activations (2) for each datasample and, for computational efficiency, we buffer the weight updates (5) and (13) and apply them once at the end of the simulation for the considered datasample. App. E and F provide further details on the implementation of all experiments.[2]

---

[2]PyTorch implementation of all methods is available at `https://github.com/meulemansalex/deep_feedback_control`.

## 6.1 Empirical verification of the theory

Figure 3 visualizes the theoretical results of Theorems 2 and 3 and Conditions 1, 2 and 3, in an empirical setting of nonlinear student teacher regression, where a randomly initialized teacher network generates synthetic training data for a student network. We see that Condition 2 is approximately satisfied for all DFC variants that learn their feedback weights (Fig. 3A), leading to close alignment with the ideal weighted MN updates of Theorem 3 (Fig. 3B). For nonlinear networks and linear direct feedback, it is in general not possible to perfectly satisfy Condition 2 as the network Jacobian $J$ varies for each datasample, while $Q_i$ remains the same. However, the results indicate that feedback learning finds a configuration for $Q_i$ that approximately satisfies Condition 2 for all datasamples. When the feedback weights are fixed, Condition 2 is approximately satisfied in the beginning of training due to a good initialization. However, as the network changes during training, Condition 2 degrades modestly, which results in worse alignment compared to DFC with trained feedback weights (Fig. 3B).

For having GN updates, both Conditions 1 and 2 need to be satisfied. Although we do not enforce Condition 1 during training, we see in Fig. 3C that it is crudely satisfied, which can be explained by the saturating properties of the $\tanh$ nonlinearity. This is reflected in the alignment with the ideal GN updates in Fig. 3D that follows the same trend as the alignment with the MN updates. Fig. 3E shows that all DFC variants remain stable throughout training, even when the feedback weights are fixed. In App. B, we indicate that Condition 3 is a good proxy for the stability shown in Fig. 3E. Finally, we see in Fig. 3F that the weight updates of DFC and DFC-SS align well with the analytical steady-state solution of Lemma 1, confirming that our learning theory of Section 3 applies to the continuous weight updates (5) of DFC.

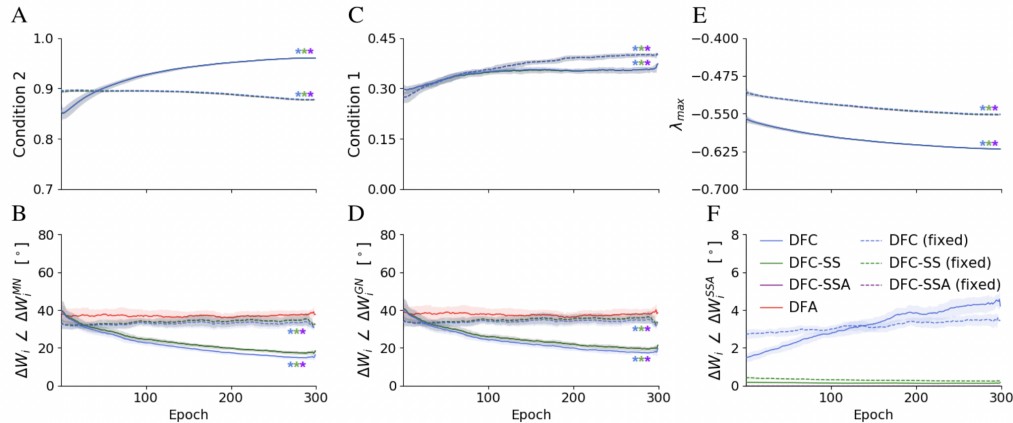

Figure 3: Results for nonlinear student-teacher regression task with layer sizes (15-10-10-5), $\tanh$ nonlinearities, a linear output layer, $k_p = 1.5$, $\lambda = 0.05$, and $\alpha = 0.0015$. (**A**) Ratio between the norms of $Q$ projected into the row space of $J$, and $Q$, with a value of 1 indicating perfect compliance of Condition 2. (**B,D,F**) Angle in degrees between the concatenated parameter updates of the whole network and: (**B**) the ideal weighted MN parameter updates (Theorem 3); (**D**) the ideal GN parameter updates (Theorem 2); and (**F**) the DFC-SSA parameter updates (see App. F.1 for all definitions). (**C**) The standard deviation of the layer norms $\|\mathbf{r}_i\|_2$, divided by the average layer norm, with a value of zero indicating perfect compliance to Condition 1. (**E**) The maximum real part of the eigenvalues of the total system dynamics matrix evaluated at equilibrium (see App. F.1), with negative real parts indicating local stability. For all measures, a window-average is plotted together with the window-std (shade). Stars indicate overlapping plots.

In Fig. 4, we show that the alignment with MN updates remains robust for $\lambda \in [10^{-3} : 10^{-1}]$ and $\alpha \in [10^{-4} : 10^{-1}]$, highlighting that our theory explains the behavior of DFC robustly when the limit of $\lambda$ and $\alpha$ to zero does not hold. When we clamp the output target to the label ($\lambda = 0.5$), the alignment with the MN updates decreases as expected (see Fig. 4), because the linearization of Lemma 1 becomes less accurate and the strong feedback changes the neural activations more significantly, thereby changing the pre-synaptic factor of the update rules (c.f. eq. 9). However, performance results on MNIST, provided in Table 2, show that the performance of DFC remains robust for a wide range of $\lambda$s and $\alpha$s, including $\lambda = 0.5$, suggesting that DFC can also provide principled

CA in this setting of strong feedback, which motivates future work to design a complementary theory for DFC focused on this extreme case.

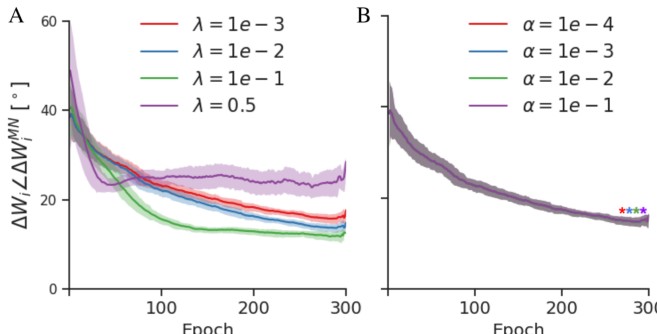

Figure 4: Comparison of the alignment between the DFC weight updates and the MN updates for variable values of $\lambda$ (A) and $\alpha$ (B), when performing the nonlinear student-teacher regression task described in Fig. 3. Stars indicate overlapping plots.

## 6.2 Performance of DFC on computer vision benchmarks

The classification results on MNIST and Fashion-MNIST (Table 1) show that the performances of DFC and its variants, but also its controls, lie close to the performance of BP, indicating that they perform proper CA in these tasks. To see significant differences between the methods, we consider the more challenging task of training an autoencoder on MNIST, where it is known that DFA fails to provide precise CA [9, 16, 32]. The results in Table 1 show that the DFC variants with trained feedback weights clearly outperform DFA and have close performance to BP. The low performance of the DFC variants with fixed feedback weights show the importance of learning the feedback weights continuously during training to satisfy Condition 2. Finally, to disentangle optimization performance from implicit regularization mechanisms, which both influence the test performance, we investigate the performance of all methods in minimizing the training loss of MNIST.[3] The results in Table 1 show improved performance of the DFC method with trained feedback weights compared to BP and controls, suggesting that the approximate MN updates of DFC can faster descend the loss landscape for this simple dataset.

Table 1: Test errors (classification) and test loss (autoencoder) corresponding to the epoch with the best validation result (for 5000 validation samples) over a training of 100 epochs (classification) or 25 epochs (autoencoder). Training loss after 100 epochs (MNIST train loss). We use the Adam optimizer [44]. Architectures: 3x256 fully connected (FC) tanh hidden layers and softmax output (classification), 256-32-256 FC hidden layers for autoencoder MNIST with tanh-linear-tanh nonlinearities, and a linear output. Mean ± std (5 random seeds). Best results (except BP) are displayed in bold.

|  | MNIST | Fashion-MNIST | MNIST-autoencoder | MNIST (train loss) |
|---|---|---|---|---|
| BP | $2.08^{\pm 0.15}\%$ | $10.60^{\pm 0.34}\%$ | $9.42^{\pm 0.09} \cdot 10^{-2}$ | $1.53^{\pm 0.19} \cdot 10^{-7}$ |
| DFC | $2.25^{\pm 0.094}\%$ | $11.17^{\pm 0.27}\%$ | $11.28^{\pm 0.18} \cdot 10^{-2}$ | $7.61^{\pm 0.65} \cdot 10^{-8}$ |
| DFC-SSA | $\mathbf{2.18^{\pm 0.16}}\%$ | $11.28^{\pm 0.27}\%$ | $11.27^{\pm 0.09} \cdot 10^{-2}$ | $4.89^{\pm 1.26} \cdot 10^{-8}$ |
| DFC-SS | $2.29^{\pm 0.097}\%$ | $\mathbf{11.15^{\pm 0.32}}\%$ | $\mathbf{11.21^{\pm 0.04} \cdot 10^{-2}}$ | $\mathbf{4.80^{\pm 0.70} \cdot 10^{-8}}$ |
| DFC (fixed) | $2.47^{\pm 0.12}\%$ | $11.62^{\pm 0.30}\%$ | $33.37^{\pm 0.60} \cdot 10^{-2}$ | $1.30^{\pm 0.15} \cdot 10^{-6}$ |
| DFC-SSA (fixed) | $2.46^{\pm 0.11}\%$ | $11.44^{\pm 0.14}\%$ | $31.90^{\pm 0.77} \cdot 10^{-2}$ | $1.73^{\pm 0.39} \cdot 10^{-6}$ |
| DFC-SS (fixed) | $2.39^{\pm 0.22}\%$ | $11.55^{\pm 0.42}\%$ | $32.31^{\pm 0.37} \cdot 10^{-2}$ | $1.67^{\pm 0.70} \cdot 10^{-6}$ |
| DFA | $2.69^{\pm 0.11}\%$ | $11.38^{\pm 0.25}\%$ | $29.95^{\pm 0.36} \cdot 10^{-2}$ | $7.09^{\pm 1.11} \cdot 10^{-7}$ |

Table 2: Test errors on MNIST with variable $\lambda$ values and fixed $\alpha = 0.0015$ (left), and with variable $\alpha$ values and fixed $\lambda = 0.08$ (right). Same experimental setting as in Table 1.

| $\lambda$ | DFC-SS | DFC | $\alpha$ | DFC-SS | DFC |
|---|---|---|---|---|---|
| $1e^{-3}$ | $2.26^{\pm 0.11}\%$ | $2.29^{\pm 0.04}\%$ | $1e^{-4}$ | $2.31^{\pm 0.12}\%$ | $2.28^{\pm 0.06}\%$ |
| $1e^{-2}$ | $2.25^{\pm 0.05}\%$ | $2.31^{\pm 0.04}\%$ | $1e^{-3}$ | $2.28^{\pm 0.15}\%$ | $2.31^{\pm 0.11}\%$ |
| $1e^{-1}$ | $2.27^{\pm 0.07}\%$ | $2.30^{\pm 0.06}\%$ | $1e^{-2}$ | $2.26^{\pm 0.05}\%$ | $2.32^{\pm 0.12}\%$ |
| $0.5$ | $2.31^{\pm 0.15}\%$ | $2.34^{\pm 0.15}\%$ | $1e^{-1}$ | $2.28^{\pm 0.11}\%$ | $2.34^{\pm 0.16}\%$ |

---

[3]We used separate hyperparameter configurations, selected for minimizing the training loss.

# 7 Discussion

We introduced DFC as an alternative biologically-plausible learning method for deep neural networks. DFC uses error feedback to drive the network activations to a desired output target. This process generates a neuron-specific learning signal which can be used to learn both forward and feedback weights locally in time and space. In contrast to other recent methods that learn the feedback weights and aim to approximate BP [14, 15, 16, 17, 26], we show that DFC approximates GN optimization, making it fundamentally different from BP approximations.

DFC is optimal – i.e., Conditions 2 and 3 are satisfied – for a wide range of feedback connectivity strengths. Thus, we prove that principled learning can be achieved with local rules and without symmetric feedforward and feedback connectivity by leveraging the network dynamics. This finding has interesting implications for experimental neuroscientific research looking for precise patterns of symmetric connectivity in the brain. Moreover, from a computational standpoint, the flexibility that stems from Conditions 2 and 3 might be relevant for other mechanisms besides learning, such as attention and prediction [8].

To present DFC in its simplest form, we used direct feedback mappings from the output controller to all hidden layers. Although numerous anatomical studies of the mammalian neocortex reported the occurrence of such direct feedback connections [45, 46], it is unlikely that all feedback pathways are direct. We note that DFC is also compatible with other feedback mappings, such as layerwise connections or separate feedback pathways with multiple layers of neurons (see App. H).

Interestingly, the three-compartment neuron is closely linked to recent multi-compartment models of the cortical pyramidal neuron [23, 25, 26, 47]. In the terminology of these models, our central, feedforward, and feedback compartments, correspond to the somatic, basal dendritic, and apical dendritic compartments of pyramidal neurons, respectively (see Fig. 1C). In line with DFC, experimental observations [48, 49] suggest that feedforward connections converge onto the basal compartment and feedback connections onto the apical compartment. Moreover, our plasticity rule for the forward weights (5) belongs to a class of dendritic predictive plasticity rules for which a biological implementation based on backpropagating action potentials has been put forward [37].

**Limitations and future work.** In practice, the forward weight updates are not exactly equal to GN or MN updates (Theorems 2 and 3), due to (i) the nonlinearity $\phi$ in the weight update rule 5, (ii) non-infinitesimal values for $\alpha$ and $\lambda$, (iii) limited training iterations for the feedback weights, and (iv) the limited capacity of linear feedback mappings to satisfy Condition 2 for each datasample. Figs. 3 and 4, and Table 2 show that DFC approximates the theory well in practice and has robust performance, however, future work can improve the results further by investigating new feedback architectures (see App. H). We note that, even though GN optimization has desirable approximate second-order optimization properties, it is presently unclear whether these second-order characteristics translate to our setting with a minibatch size of 1. Currently, our proposed feedback learning rule (13) aims to approximate one specific configuration and hence does not capitalize on the increased flexibility of DFC and Condition 2. Therefore, an interesting future direction is to design more flexible feedback learning rules that aim to satisfy Conditions 2 and 3 without targeting one specific configuration. Furthermore, DFC needs two separate phases for training the forward weights and feedback weights. Interestingly, if the feedback plasticity rule (13) uses a high-passed filtered version of the presynaptic input $\mathbf{u}$, both phases can be merged into one, with plasticity always on for both forward and feedback weights (see App. C.3). Finally, as DFC is dynamical in nature, it is costly to simulate on commonly used hardware for deep learning, prohibiting us from testing DFC on large-scale problems such as those considered by Bartunov et al. [10]. A promising alternative is to implement DFC on analog hardware, where the dynamics of DFC can correspond to real physical processes on a chip. This would not only make DFC resource-efficient, but also position DFC as an interesting training method for analog implementations of deep neural networks, commonly used in Edge AI and other applications where low energy consumption is key [50, 51].

To conclude, we show that DFC can provide principled CA in deep neural networks by actively using error feedback to drive neural activations. The flexible requirements for feedback mappings combined with the strong link between DFC and GN, underline that it is possible to do principled CA in neural networks without adhering to the symmetric layer-wise feedback structure imposed by BP.

## Acknowledgments and Disclosure of Funding

This work was supported by the Swiss National Science Foundation (B.F.G. CRSII5-173721 and 315230_189251), ETH project funding (B.F.G. ETH-20 19-01), the Human Frontiers Science Program (RGY0072/2019) and funding from the Swiss Data Science Center (B.F.G, C17-18, J. v. O. P18-03). João Sacramento was supported by an Ambizione grant (PZ00P3_186027) from the Swiss National Science Foundation. Pau Vilimelis Aceituno was supported by an ETH Zürich Postdoc fellowship. Javier García Ordóñez received support from La Caixa Foundation through the Postgraduate Studies in Europe scholarship. We would like to thank Anh Duong Vo and Nicolas Zucchet for feedback, William Podlaski, Jean-Pascal Pfister and Aditya Gilra for insightful discussions, and Simone Surace for his detailed feedback on Appendix C.1.

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
