# Supplementary Material

**Alexander Meulemans**[*], **Matilde Tristany Farinha**[*], **Javier García Ordóñez**,
**Pau Vilimelis Aceituno, João Sacramento, Benjamin F. Grewe**

Institute of Neuroinformatics, University of Zürich and ETH Zürich
`ameulema@ethz.ch`

## Table of Contents

# A Proofs and extra information for Section 3: Learning theory

## A.1 Linearized dynamics and fixed points

In this section, we linearize the network dynamics around the feedforward voltage levels $\mathbf{v}_i^-$ (i.e., the equilibrium of the network when no feedback is present) and study the equilibrium points resulting from the feedback input from the controller.

**Notation.** First, we introduce some shorthand notations:

$$\mathbf{v} \triangleq [\mathbf{v}_1^T...\mathbf{v}_L^T]^T \tag{15}$$

$$Q \triangleq [Q_1^T...Q_L^T]^T \tag{16}$$

$$f_i(\mathbf{v}_{i-1}) \triangleq W_i\phi(\mathbf{v}_{i-1}) \tag{17}$$

$$\Delta^-\mathbf{v}_i \triangleq \mathbf{v}_i - \mathbf{v}_i^- \tag{18}$$

$$\Delta\mathbf{v}_i \triangleq \mathbf{v}_i - W_i\phi(\mathbf{v}_{i-1}) = \mathbf{v}_i - \mathbf{v}_i^{\mathrm{ff}} \tag{19}$$

$$\Delta\mathbf{v} \triangleq [\Delta\mathbf{v}_1^T...\Delta\mathbf{v}_L^T]^T \tag{20}$$

$$J_{i,k} \triangleq \frac{\partial\mathbf{v}_i}{\partial\mathbf{v}_k}\bigg|_{\mathbf{v}_k=\mathbf{v}_k^-} = \frac{\partial f_i(..f_{k+1}(\mathbf{v}_k)..)}{\partial\mathbf{v}_k}\bigg|_{\mathbf{v}_k=\mathbf{v}_k^-} \tag{21}$$

$$J_i \triangleq \frac{\partial\mathbf{r}_L}{\partial\mathbf{v}_i}\bigg|_{\mathbf{v}_i=\mathbf{v}_i^-} \tag{22}$$

$$J \triangleq [J_1...J_L] = \frac{\partial\mathbf{r}_L}{\partial\Delta\mathbf{v}}\bigg|_{\Delta\mathbf{v}=0} \tag{23}$$

$$\boldsymbol{\delta}_L \triangleq -\lambda\frac{\partial\mathcal{L}}{\partial\mathbf{r}_L}\bigg|_{\mathbf{r}_L=\mathbf{r}_L^-} \tag{24}$$

$$\mathbf{r}_L^* \triangleq \mathbf{r}_L^- + \boldsymbol{\delta}_L \tag{25}$$

To investigate the steady state of the network and controller dynamics, we start by proving Lemma 1, which we restate here for convenience.

**Lemma S1.** *Assuming stable dynamics, a small target stepsize $\lambda$, and $W_i$ and $Q_i$ fixed, the steady-state solutions of the dynamical systems* (1) *and* (4) *can be approximated by*

$$\mathbf{u}_{\mathrm{ss}} = (JQ+\tilde{\alpha}I)^{-1}\boldsymbol{\delta}_L + \mathcal{O}(\lambda^2), \quad \mathbf{v}_{\mathrm{ss}} = \mathbf{v}_{\mathrm{ss}}^{\mathrm{ff}} + Q(JQ+\tilde{\alpha}I)^{-1}\boldsymbol{\delta}_L + \mathcal{O}(\lambda^2), \tag{26}$$

*with $J \triangleq \frac{\partial\mathbf{r}_L^-}{\partial\mathbf{v}}\big|_{\mathbf{v}=\mathbf{v}^-}$ the Jacobian of the network output w.r.t. $\mathbf{v}$, evaluated at the network equilibrium without feedback, $\boldsymbol{\delta}_L$ the output error as defined in* (3), *$\mathbf{v}_{i,ss}^{\mathrm{ff}} = W_i\phi(\mathbf{v}_{i-1,\mathrm{ss}})$, and $\tilde{\alpha} = \alpha/(1+\alpha k_p)$.*

*Proof.* The proof is ordered as follows: first, we linearize the network dynamics around the feedforward equilibrium of (2). Then, we solve the algebraic set of linear equilibrium equations.

With the introduced shorthand notation, we can combine (1) for $i = 1, ..., L$ into a single dynamical equation for the whole network:

$$\tau\frac{\mathrm{d}\mathbf{v}}{\mathrm{d}t} = -\Delta\mathbf{v} + Q\mathbf{u}. \tag{27}$$

By linearizing the dynamics, we can derive the control error $\mathbf{e}(t) \triangleq \mathbf{r}_L^* - \mathbf{r}_L(t)$ as an affine transformation of $\Delta\mathbf{v}$. First, note that

$$\Delta^-\mathbf{v}_i = \mathbf{v}_i - W_i\phi(\mathbf{v}_{i-1}) + W_i\phi(\mathbf{v}_{i-1}) - \mathbf{v}_i^- \tag{28}$$

$$= \Delta\mathbf{v}_i + W_i\phi(\mathbf{v}_{i-1}^- + \Delta^-\mathbf{v}_{i-1}) - \mathbf{v}_i^- \tag{29}$$

$$= \Delta\mathbf{v}_i + J_{i,i-1}\Delta^-\mathbf{v}_{i-1} + \mathcal{O}(\lambda^2). \tag{30}$$

By recursion, we have that

$$\Delta^- \mathbf{v}_i = \Delta \mathbf{v}_i + \sum_{k=1}^{i-1} J_{i,k} \Delta \mathbf{v}_k + \mathcal{O}(\lambda^2), \tag{31}$$

with $\Delta \mathbf{v}_1 = \Delta^- \mathbf{v}_1 = \mathbf{v}_1 - \mathbf{v}_1^-$ because the input to the network is not influenced by the controller, i.e., $\mathbf{v}_0 = \mathbf{v}_0^-$.

The control error given by

$$\mathbf{e} \triangleq \mathbf{r}_L^* - \mathbf{r}_L = \mathbf{r}_L^* - \mathbf{r}_L^- + \mathbf{r}_L^- - \mathbf{r}_L \tag{32}$$

$$= \boldsymbol{\delta}_L - J_L \Delta^- \mathbf{v}_L + \mathcal{O}(\lambda^2) \tag{33}$$

$$= \boldsymbol{\delta}_L - J_L \Big( \Delta \mathbf{v}_L + \sum_{k=1}^{L-1} J_{L,k} \Delta \mathbf{v}_k \Big) + \mathcal{O}(\lambda^2) \tag{34}$$

$$= \boldsymbol{\delta}_L - \sum_{k=1}^{L} J_k \Delta \mathbf{v}_k + \mathcal{O}(\lambda^2) \tag{35}$$

$$= \boldsymbol{\delta}_L - J \Delta \mathbf{v} + \mathcal{O}(\lambda^2). \tag{36}$$

The controller dynamics are given by

$$\mathbf{u}(t) = \mathbf{u}^{\text{int}}(t) + k_p \mathbf{e}(t) \tag{37}$$

$$\tau_u \frac{\mathrm{d}}{\mathrm{d}t} \mathbf{u}^{\text{int}}(t) = \mathbf{e}(t) - \alpha \mathbf{u}^{\text{int}}(t) \tag{38}$$

By differentiating (37) and using $\mathbf{u}^{\text{int}} = \mathbf{u} - k_p \mathbf{e}$ we get the following controller dynamics for $\mathbf{u}$:

$$\tau_u \frac{\mathrm{d}}{\mathrm{d}t} \mathbf{u}(t) = (1 + \alpha k_p) \mathbf{e}(t) + k_p \tau_u \frac{\mathrm{d}}{\mathrm{d}t} \mathbf{e}(t) - \alpha \mathbf{u}(t). \tag{39}$$

The system of equations (27) and (39) can be solved in steady state as follows. From (27) at steady state, we have

$$\Delta \mathbf{v}_{\text{ss}} = Q \mathbf{u}_{\text{ss}}. \tag{40}$$

Substituting $\Delta \mathbf{v}_{\text{ss}}$ into the steady state of (39) while using the linearized control error (32) gives

$$\mathbf{u}_{\text{ss}} = (JQ + \tilde{\alpha} I)^{-1} \boldsymbol{\delta}_L + \mathcal{O}(\lambda^2) \tag{41}$$

$$\Delta \mathbf{v}_{\text{ss}} = Q(JQ + \tilde{\alpha} I)^{-1} \boldsymbol{\delta}_L + \mathcal{O}(\lambda^2), \tag{42}$$

with $\tilde{\alpha} \triangleq \frac{\alpha}{1 + \alpha k_p}$. Using $\mathbf{v} = \mathbf{v}^{\text{ff}} + \Delta \mathbf{v}$ concludes the proof. $\square$

In the next section, we will investigate how this steady-state solution can result in useful weight updates (plasticity) for the forward weights $W_i$.

## A.2 DFC approximates Gauss-Newton optimization

In this subsection, we will investigate the steady state (42) for $\alpha \to 0$ and link the resulting weight updates to Gauss-Newton (GN) optimization. Before proving Theorem 2, we need to introduce and prove some lemmas. First, we need to show that $\lim_{\alpha \to 0} \Delta \mathbf{v}_{\text{ss}} = J^\dagger \boldsymbol{\delta}_L$ under Condition 2, with $J^\dagger$ the Moore-Penrose pseudoinverse of $J$ [52, 53]. This is done in the following Lemma.

**Lemma S2.** *Assuming $J$ has full rank,*

$$\lim_{\alpha \to 0} Q(JQ + \alpha I)^{-1} = J^\dagger \tag{43}$$

*iff Condition 2 holds, i.e., $Col(Q) = Row(J)$.*

*Proof.* We begin by stating the Moore-Penrose conditions [53]:

**Condition S1.**  $B = A^\dagger$ *iff*

1.  $ABA = A$

2.  $BAB = B$

3.  $AB = (AB)^T$

4.  $BA = (BA)^T$

In this proof, we need to consider 2 general cases: (i) $J$ has full rank and $Q$ does not and (ii) $Q$ and $J$ have both full rank. As $J^T$ and $Q$ have much more rows than columns, they will almost always be of full rank, however, we consider both cases for completeness.

In case (i), where $Q$ has lower rank than $J$, we have that $\mathrm{rank}\big(Q(JQ + \alpha I)^{-1}\big) \leq \mathrm{rank}(Q)$. As $\mathrm{rank}(J^\dagger) = \mathrm{rank}(J^T)$, $Q(JQ + \alpha I)^{-1}$ can never be the pseudoinverse of $J$, thereby proving that a necessary condition for (43) is that $\mathrm{rank}(Q) \geq \mathrm{rank}(J)$ (note that this condition is satisfied by Condition 2). Now, that we showed that it is a necessary condition that $Q$ is full rank (as $J$ is full rank by assumption of the lemma) for eq. (43) to hold, we proceed with the second case.

In case (ii), where $Q$ and $J^T$ have both full rank, we need to prove under which conditions on $Q$, $S \triangleq \lim_{\alpha \to 0} Q(JQ + \alpha I)^{-1}$ is equal to $J^\dagger$. As $Q$ and $J^T$ have both full rank, $JQ$ is of full rank and we have

$$JS = \lim_{\alpha \to 0} JQ(JQ + \alpha I)^{-1} = I. \tag{44}$$

Hence, conditions S1.1, S1.2 and S1.3 are trivially satisfied:

1.  $JSJ = IJ = J$

2.  $SJS = SI = S$

3.  $JS = I = I^T = (JS)^T$

Condition S1.4 will only be satisfied under certain constraints on $Q$. We first assume Condition 2 holds to show its sufficiency after which we continue to show its necessity.

Consider $U_J$ as an orthogonal basis of the column space of $J^T$. Then, we can write

$$J = M_J U_J^T \tag{45}$$

for some full rank square matrix $M_J$. As we assume Condition 2 holds, we can similarly write $Q$ as

$$Q = U_J M_Q \tag{46}$$

for some full rank square matrix $M_Q$. Condition S1.4 can now be written as

$$SJ = Q(JQ)^{-1}J \tag{47}$$

$$= U_J M_Q (M_J M_Q)^{-1} M_J U_J^T \tag{48}$$

$$= U_J U_J^T \tag{49}$$

$$= (SJ)^T, \tag{50}$$

showing that S is indeed the pseudoinverse of $J$ if Condition 2 holds, proving its sufficiency.

For showing the necessity of Condition 2, we use a proof by contradiction. We now assume that Condition 2 does not hold and hence the column space of $Q$ is not equal to that of $J$. Similar as before, consider $U_Q$ and orthogonal basis of the column space of $Q$. Furthermore, consider the square orthogonal matrix $\bar{U}_J \triangleq [U_J \tilde{U}_J]$ with $U_J$ as defined in (45) and $\tilde{U}_J$ orthogonal on $U_J$. We can now decompose $Q$ into a part inside the column space of $J^T$ and outside of that column space:

$$Q = U_Q M_Q \tag{51}$$

$$= \bar{U}_J \bar{U}_J^T U_Q M_Q \tag{52}$$

$$= U_J P_Q + \tilde{U}_J \tilde{P}_Q, \tag{53}$$

with $M_Q$ a square full rank matrix, $P_Q \triangleq U_J^T U_Q M_Q$, and $\tilde{P}_Q \triangleq \tilde{U}_J^T U_Q M_Q$. The first part of (51) represents the part of $Q$ inside the column space of $J^T$ and the second part represents the part of $Q$ outside of this column space. For clarity, we assume that $P_Q$ is full rank[4], which is true in all but degenerate cases. Note that $\tilde{P}_Q$ is different from zero, as we assume Condition 2 does not hold in this proof by contradiction. Using this decomposition of $Q$, we can write $SJ$ used in Condition S1.4 as

$$SJ = Q(JQ)^{-1}J \tag{54}$$
$$= (U_J P_Q + \tilde{U}_J \tilde{P}_Q)(M_J P_Q)^{-1}(M_J U_J^T) \tag{55}$$
$$= U_J U_J^T + \tilde{U}_J \tilde{P}_Q P_Q^{-1} U_J^T. \tag{56}$$

The first part of the last equation is always symmetric, hence Condition S1.4 boils down to the second part being symmetric:

$$\tilde{U}_J \tilde{P}_Q P_Q^{-1} U_J^T = U_J P_Q^{-T} \tilde{P}_Q^T \tilde{U}_J^T \tag{57}$$
$$\Rightarrow \quad \tilde{U}_J^T \tilde{U}_J \tilde{P}_Q P_Q^{-1} U_J^T = \tilde{U}_J^T U_J P_Q^{-T} \tilde{P}_Q^T \tilde{U}_J^T \tag{58}$$
$$\Rightarrow \quad \tilde{P}_Q P_Q^{-1} U_J^T = 0 \tag{59}$$
$$\Rightarrow \quad U_J P_Q^{-T} \tilde{P}_Q^T = 0. \tag{60}$$

As $U_J$ has a zero-dimensional null space and $P_Q$ is full rank, S1.4 can only hold when $\tilde{P}_Q = 0$. This contradicts with our initial assumption in this proof by contradiction, stating that Condition 2 does not hold and consequently $Q$ has components outside of the column space of $J$, thereby proving that Condition 2 is necessary.

$\square$

Theorem 2 states that the updates for $W_i$ in DFC at steady-state align with the updates $W_i$ prescribed by the GN optimization method for a feedforward neural network. We first formalize a *feedforward fully connected neural network*.

**Definition S1.** *A feedforward fully connected neural network with $L$ layers, input dimension $n_0$, output dimension $n_L$ and hidden layer dimensions $n_i$, $0 < i < L$ is defined by the following sequence of mappings:*

$$\mathbf{r}_i = \phi(W_i \mathbf{r}_{i-1}), \quad 0 < i < L \tag{61}$$
$$\mathbf{r}_L = \phi_L(W_i \mathbf{r}_{L-1}), \tag{62}$$

*with $\phi$ and $\phi_L$ activation functions, $\mathbf{r}_0$ the input of the network, and $\mathbf{r}_L$ the output of the network.*

The Lemma below shows that the network dynamics (1) at steady-state are equal to a feedforward neural network corresponding to Definition S1 in the absence of feedback.

**Lemma S3.** *In the absence of feedback ($\mathbf{u}(t) = 0$), the system dynamics (1) at steady-state are equivalent to a feedforward neural network defined by Definition S1.*

*Proof.* The proof is trivial upon noting that $Q\mathbf{u} = 0$ without feedback and computing the steady-state of (1) using $\mathbf{r}_i \triangleq \phi(\mathbf{v}_i)$. $\square$

Following the notation of eq. (2), we denote with $\mathbf{r}_i^-$ the firing rates of the network in steady-state when feedback is absent, hence corresponding to the activations of a conventional feedforward neural network. The following Lemma investigates what the GN parameter updates are for a feedforward neural network. Later, we then show that the updates at equilibrium of DFC approximate these GN updates. For clarity, we assume that the network has only weights and no biases in all the following theorems and proofs, however, all proofs can be easily extended to comprise both weights and biases. First, we need to introduce some new notation for vectorized matrices.

$$\vec{W}_i \triangleq \text{vec}(W_i) \tag{63}$$
$$\bar{W} \triangleq [\vec{W}_1^T ... \vec{W}_L^T]^T, \tag{64}$$

where $\text{vec}(W_i)$ denotes the concatenation of the columns of $W_i$ in a column vector.

---

[4]If $P_Q$ is not of full rank, $JQ$ is not of full rank and hence $\lim_{\alpha \to 0}(JQ + \alpha I)$ also not. Consequently, $\lim_{\alpha \to 0} Q(JQ + \alpha I)^{-1}$ will project $Q$ onto something of lower rank, making it impossible for $S$ to approximate $J^\dagger$, thereby showing that it is necessary that $P_Q$ is full rank.

**Lemma S4.** *Assuming an $L^2$ task loss and Condition 1 holds, the Gauss-Newton parameter updates for the weights of a feedforward network defined by Definition S1 for a minibatch size of 1 is given by*

$$\Delta \bar{W}^{GN} = \frac{1}{2\lambda \|\mathbf{r}\|_2^2} R J^\dagger \boldsymbol{\delta}_L, \tag{65}$$

*with $R$ defined in eq. (70).*

*Proof.* Consider the Jacobian of the output w.r.t. the network weights $W$ (in vectorized form as defined above), evaluated at the feedforward activation:

$$J_{\bar{W}} \triangleq \frac{\partial \mathbf{r}_L}{\partial \bar{W}} \Big|_{\mathbf{r}_L = \mathbf{r}_L^-}. \tag{66}$$

For a minibatch size of 1, the GN update for the parameters $\bar{W}$, assuming an $L^2$ output loss, is given by [35, 54]

$$\Delta \bar{W}^{GN} = J_{\bar{W}}^\dagger (\mathbf{r}_L^{\text{true}} - \mathbf{r}_L^-) = \frac{1}{2\lambda} J_{\bar{W}}^\dagger \boldsymbol{\delta}_L, \tag{67}$$

with $\mathbf{r}_L^{\text{true}}$ the true supervised output (e.g., the class label). The remainder of this proof will manipulate expression (67) in order to reach (65). Using $J_{\vec{W}_i} \triangleq \frac{\partial \mathbf{r}_L}{\partial \vec{W}_i} \big|_{\mathbf{r}_L = \mathbf{r}_L^-}$, $J_{\bar{W}}$ can be restructured as:

$$J_{\bar{W}} = [J_{\vec{W}_1} ... J_{\vec{W}_L}]. \tag{68}$$

Moreover, $J_{\vec{W}_i} = J_i \frac{\partial \mathbf{v}_i}{\partial \vec{W}_i} \big|_{\mathbf{v}_i = \mathbf{v}_i^-}$. Using Kronecker products, this becomes[5]

$$J_{\vec{W}_i} = J_i \big( (\mathbf{r}_{i-1}^-)^T \otimes I \big). \tag{69}$$

Using the structure of $J_{\bar{W}}$, this leads to

$$J_{\bar{W}} = J R^T \tag{70}$$

$$R^T \triangleq \begin{bmatrix} (\mathbf{r}_0^-)^T \otimes I & 0 & \cdots & 0 \\ 0 & (\mathbf{r}_1^-)^T \otimes I & \cdots & 0 \\ \vdots & 0 & \ddots & 0 \\ 0 & \cdots & 0 & (\mathbf{r}_{L-1}^-)^T \otimes I \end{bmatrix} \tag{71}$$

with the dimensions of $I$ such that the equality $J_{\bar{W}} = J R^T$ holds. What remains to be proven is that $J_{\bar{W}}^\dagger = \frac{1}{\|\mathbf{r}\|_2^2} R J^\dagger$, assuming that Condition 1 holds and knowing that $J_{\bar{W}} = J R^T$. To prove this, we need to know under which conditions $(J R^T)^\dagger = (R^T)^\dagger J^\dagger$. The following condition specifies when a pseudoinverse of a matrix product can be factorized [55].

**Condition S2.** *The Moore-Penrose pseudoinverse of a matrix product $(AB)^\dagger$ can be factorized as $(AB)^\dagger = B^\dagger A^\dagger$ if one of the following conditions hold:*

1. *A has orthonormal columns*

2. *B has orthonormal rows*

3. *$B = A^T$*

4. *A has all columns linearly independent and B has all rows linearly independent*

In our case, J has more columns than rows, hence conditions S2.1 and S2.4 can never be satisfied. Furthermore, condition S2.3 does not hold, which leaves us with condition S2.2. To investigate whether $R^T$ has orthonormal rows, we compute $R^T R$:

$$R^T R = \begin{bmatrix} \|\mathbf{r}_0^-\|_2^2 I & \cdots & 0 \\ \vdots & \ddots & \vdots \\ 0 & \cdots & \|\mathbf{r}_{L-1}^-\|_2^2 I \end{bmatrix} \tag{72}$$

---

[5]The Kronecker product leads to the following equality: $\text{vec}(ABC) = (C^T \otimes A)\text{vec}(B)$. Applied to our situation, this leads to the following equality: $\mathbf{v}_i = W_i \mathbf{r}_{i-1} = (\mathbf{r}_{i-1}^T \otimes I)\vec{W}_i$

If Condition 1 holds, we have $\|\mathbf{r}_0^-\|_2^2 = \ldots = \|\mathbf{r}_{L-1}^-\|_2^2 \triangleq \|\mathbf{r}\|_2^2$ such that:

$$R^T R = \|\mathbf{r}\|_2^2 I. \tag{73}$$

Hence, $\frac{1}{\|\mathbf{r}\|_2} R^T$ has orthonormal rows iff Condition 1 holds. From now on, we assume that Condition 1 holds. Next, we will compute $(R^T)^\dagger$. Consider $R^T = U\Sigma V^T$, the singular value decomposition (SVD) of $R^T$. Its pseudoinverse is given by $(R^T)^\dagger = V\Sigma^\dagger U^T$. As the SVD is unique and $\frac{1}{\|\mathbf{r}\|_2} R^T$ has orthonormal rows, we can construct the SVD manually:

$$R^T = \underbrace{I}_{=U} \underbrace{[\|\mathbf{r}\|_2 I \quad 0]}_{=\Sigma} \underbrace{\begin{bmatrix} \frac{1}{\|\mathbf{r}\|_2} R^T \\ \tilde{V}^T \end{bmatrix}}_{=V^T}, \tag{74}$$

with $\tilde{V}^T$ being a basis orthonormal to $\frac{1}{\|\mathbf{r}\|_2} R^T$. Hence, we have that

$$(R^T)^\dagger = V\Sigma^\dagger U^T = \frac{1}{\|\mathbf{r}\|_2^2} R. \tag{75}$$

Putting everything together and assuming that Condition 1 holds, we have that

$$\Delta \bar{W}^{GN} = \frac{1}{2\lambda} J_{\bar{W}}^\dagger \boldsymbol{\delta}_L = \frac{1}{2\lambda\|\mathbf{r}\|_2^2} R J^\dagger \boldsymbol{\delta}_L, \tag{76}$$

thereby concluding the proof. $\qquad\square$

Now, we are ready to prove Theorem 2.

**Theorem S5** (Theorem 2 in main manuscript). *Assuming Conditions 1 and 2 hold, $J$ is full rank, the task loss $\mathcal{L}$ is a $L^2$ loss, and $\lambda, \alpha \to 0$, then the following steady-state (ss) updates for the forward weights*

$$\Delta W_i = \eta(\mathbf{v}_{i,\mathrm{ss}} - \mathbf{v}_{i,\mathrm{ss}}^{\mathit{ff}})\mathbf{r}_{i-1,\mathrm{ss}}^T, \tag{77}$$

*with $\eta$ a stepsize parameter, align with the weight updates for $W_i$ for the feedforward network (2) prescribed by the GN optimization method with a minibatch size of 1.*

*Proof.* Lemma S3 shows that the dynamical network (1) at equilibrium in the absence of feedback is equivalent to a feedforward neural network. Lemma S4 provides the GN update step for such a feedforward network, and hence also for our dynamical network. To prove Theorem 2, we have to show that $\lim_{\alpha,\lambda\to 0} \eta(\mathbf{v}_{i,\mathrm{ss}}^S - \mathbf{v}_{i,\mathrm{ss}}^B)\mathbf{r}_{i-1,\mathrm{ss}}^T$ is aligned with the GN update. First, we combine the updates $\Delta W_i$ into their concatenated vectorized form:

$$\Delta W_i = \eta \Delta \mathbf{v}_{i,\mathrm{ss}} \mathbf{r}_{i-1,\mathrm{ss}}^T \tag{78}$$

$$\Delta \vec{W}_i = (\mathbf{r}_{i-1,\mathrm{ss}} \otimes I)\Delta \mathbf{v}_{i,\mathrm{ss}} \tag{79}$$

$$\Delta \bar{W} = \begin{bmatrix} \Delta \vec{W}_1 \\ \vdots \\ \Delta \vec{W}_L \end{bmatrix} = \eta R_{\mathrm{ss}} \Delta \mathbf{v}_{\mathrm{ss}} \tag{80}$$

with $R_{\mathrm{ss}}$ as defined in (70), but then with $\mathbf{r}_{i,\mathrm{ss}}$ instead of $\mathbf{r}_i^-$. From the linearized dynamics (42), combined with Lemma S2 while assuming $J$ is of full rank, we have that

$$\lim_{\alpha\to 0} \Delta \mathbf{v}_{\mathrm{ss}} = J^\dagger \boldsymbol{\delta}_L + \mathcal{O}(\lambda^2) \tag{81}$$

iff Condition 2 holds. Taking $\eta = \frac{1}{2\lambda\|\mathbf{r}\|_2^2}$ and assuming an $L^2$ task loss, we have (using Lemma S4):

$$\lim_{\alpha\to 0} \Delta \bar{W} = \frac{1}{2\lambda\|\mathbf{r}\|_2^2} R_{\mathrm{ss}} \Delta \mathbf{v}_{\mathrm{ss}} = \frac{1}{2\lambda\|\mathbf{r}\|_2^2} R_{\mathrm{ss}} J^\dagger 2\lambda(\mathbf{r}_L^{\mathrm{true}} - \mathbf{r}_L^-) + \mathcal{O}(\lambda^2) \tag{82}$$

$$\lim_{\alpha,\lambda\to 0} \Delta \bar{W} = \frac{1}{\|\mathbf{r}\|_2^2} R J^\dagger (\mathbf{r}_L^{\mathrm{true}} - \mathbf{r}_L^-) \tag{83}$$

where we used that $\lim_{\lambda\to 0} R_{\mathrm{ss}} = R$. By comparing $\lim_{\alpha,\lambda\to 0} \Delta \bar{W}$ to Lemma S4, we see that it is equal to the GN update for $\bar{W}$ for a minibatchsize of 1, iff Condition 1 and 2 hold and for an appropriate learning rate $\eta = \frac{1}{2\lambda\|\mathbf{r}\|_2^2}$. As $\eta$ is a scalar, we have that for arbitrary $\eta$, $\lim_{\alpha,\lambda\to 0} \Delta \bar{W}$ is proportional to the Gauss-Newton parameter update, thereby concluding the proof. $\qquad\square$

This theorem shows that for tasks with an $L^2$ loss and when Conditions 1 and 2 hold, DFC approximates Gauss-Newton updates with a minibatch size of 1, which becomes an exact equivalence in the limit of $\alpha$ and $\lambda$ to zero.

## A.3 DFC uses minimum norm updates

To remove the need for Condition 1 and a L2 task loss,[6] we show that the learning behavior of our network is mathematically sound under more relaxed conditions. Theorem 3 (restated below for convenience) shows that for arbitrary loss functions and without the need for Condition 1, our synaptic plasticity rule can be interpreted as a weighted minimum norm (MN) parameter update for reaching the output target, assuming linearized dynamics (which becomes exact in the limit of $\lambda \to 0$).

**Theorem S6.** *Assuming stable dynamics, Condition 2 holds and $\lambda, \alpha \to 0$, the steady-state weight updates (9) are proportional to the weighted MN updates of $W_i$ for letting the feedforward output $\mathbf{r}_L^-$ reach $\mathbf{r}_L^*$, i.e., the solution to the following optimization problem:*

$$\underset{\Delta W_i, i \in [1,..,L]}{\arg\min} \quad \sum_{i=1}^{L} \|\mathbf{r}_{i-1}^{-(m)}\|_2^2 \|\Delta W_i\|_F^2 \qquad s.t. \quad \mathbf{r}_L^{-(m+1)} = \mathbf{r}_L^{*(m)}, \tag{84}$$

*with $m$ the iteration and $\mathbf{r}_L^{-(m+1)}$ the network output without feedback after the weight update.*

*Proof.* Rewriting the optimization problem using

$$M = \begin{bmatrix} \|\mathbf{r}_0^-\|_2 I & \dots & 0 \\ & \ddots & \\ 0 & \dots & \|\mathbf{r}_{L-1}^-\|_2 I \end{bmatrix} \tag{85}$$

and the concatenated vectorized weights $\bar{W}$, we get:

$$\underset{\Delta \bar{W}}{\arg\min} \quad \|M \Delta \bar{W}\|_2^2 \tag{86}$$

$$\text{s.t.} \qquad \mathbf{r}_L^{-(m+1)} = \mathbf{r}_L^{*(m)} \tag{87}$$

Linearizing the feedforward dynamics around the current parameter values $\bar{W}^{(m)}$ and using Lemma S3, we get:

$$\mathbf{r}_L^{-(m+1)} = \mathbf{r}_L^{-(m)} + J_{\bar{W}} \Delta \bar{W} + \mathcal{O}(\|\Delta \bar{W}\|_2^2). \tag{88}$$

We will now assume that $\mathcal{O}(\|\Delta \bar{W}\|_2^2)$ vanishes in the limit of $\lambda \to 0$, relative to the other terms in this Taylor expansion, and check this assumption at the end of the proof. Using (88) to rewrite the constraints (87), we get:

$$\mathbf{r}_L^{-(m+1)} = \mathbf{r}_L^{*(m)} \tag{89}$$

$$\Leftrightarrow \quad J_{\bar{W}} \Delta \bar{W} = \boldsymbol{\delta}_L. \tag{90}$$

To solve the optimization problem, we construct its Lagrangian:

$$\mathbb{L} = \|M \Delta \bar{W}\|_2^2 + \boldsymbol{\mu}^T (J_{\bar{W}} \Delta \bar{W} - \boldsymbol{\delta}_L), \tag{91}$$

with $\boldsymbol{\mu}$ the Lagrange multipliers. As this is a convex optimization problem, the optimal solution can be found by solving the following set of equations:

$$\left(\frac{\partial \mathbb{L}}{\partial \boldsymbol{\mu}}\right)^T = J_{\bar{W}} \Delta \bar{W}^* - \boldsymbol{\delta}_L = 0 \tag{92}$$

$$\left(\frac{\partial \mathbb{L}}{\partial \Delta \bar{W}}\right)^T = 2M^2 \Delta \bar{W}^* + J_{\bar{W}}^T \boldsymbol{\mu}^* = 0 \quad \Rightarrow \Delta \bar{W}^* = -\frac{1}{2} M^{-2} J_{\bar{W}}^T \boldsymbol{\mu}^* \tag{93}$$

$$\Rightarrow \quad \boldsymbol{\mu}^* = -2(J_{\bar{W}} M^{-2} J_{\bar{W}}^T)^{-1} \boldsymbol{\delta}_L \tag{94}$$

$$\Rightarrow \quad \Delta \bar{W}^* = M^{-2} J_{\bar{W}}^T (J_{\bar{W}} M^{-2} J_{\bar{W}}^T)^{-1} \boldsymbol{\delta}_L = M^{-1} (J_{\bar{W}} M^{-1})^\dagger \boldsymbol{\delta}_L, \tag{95}$$

---

[6]The Gauss-Newton method can be generalized to other loss functions by using the Generalized Gauss-Newton method [56].

assuming $J_{\bar{W}}M^{-2}J_{\bar{W}}^T$ is invertible, which is highly likely, as $J_{\bar{W}}$ is a skinny horizontal matrix and $M$ full rank. As $\mathcal{O}(\|\Delta\bar{W}\|_2) = \mathcal{O}(\lambda)$ and $\mathcal{O}(\|\Delta\bar{W}\|_2^2) = \mathcal{O}(\lambda^2)$, the Taylor expansion error $\mathcal{O}(\|\Delta\bar{W}\|_2^2)$ vanishes in the limit of $\lambda \to 0$, relative to the zeroth and first order terms, thereby confirming our assumption.

Now, we proceed by factorizing $(J_{\bar{W}}M^{-1})^\dagger$ into $J^\dagger$ and some other term, similar as in Lemma S4. First, we note that $J_{\bar{W}}M^{-1} = JR^T M^{-1}$, with $R^T$ defined in eq. (70). Furthermore, we have that $(R^T M^{-1})(R^T M^{-1})^T = I$, hence $R^T M^{-1}$ has orthonormal rows. Following Condition S2, we can factorize $(J_{\bar{W}}M^{-1})^\dagger$ as follows:

$$\left(J_{\bar{W}}M^{-1}\right)^\dagger = \left(JR^T M^{-1}\right)^\dagger = \left(R^T M^{-1}\right)^\dagger J^\dagger = M^{-1}RJ^\dagger \tag{96}$$

$$\Rightarrow \quad \Delta\bar{W}^* = M^{-2}RJ^\dagger \boldsymbol{\delta}_L \tag{97}$$

$$\Rightarrow \quad \Delta W_i^* = \frac{1}{\|\mathbf{r}_{i-1}\|_2^2}\left[J^\dagger\boldsymbol{\delta}_L\right]_i \mathbf{r}_{i-1}^T, \tag{98}$$

with $\left[J^\dagger\boldsymbol{\delta}_L\right]_i$ the entries of the vector $J^\dagger\boldsymbol{\delta}_L$ corresponding to $\mathbf{v}_i$. We used $\left(R^T M^{-1}\right)^\dagger = M^{-1}R$, which has a similar derivation as the one used for $\left(R^T\right)^\dagger$ in Lemma S4.

We continue by showing that the weight update at equilibrium of DFC aligns with the MN solutions $\Delta W_i^*$. Adapting (83) from Theorem 2 to arbitrary loss functions, assuming 2 holds, and taking a layer-specific learning rate $\eta_i = \frac{1}{\|\mathbf{r}_{i-1}\|_2^2}$, we get that

$$\lim_{\alpha,\lambda\to 0}\Delta W_i = \frac{1}{\|\mathbf{r}_{i-1}\|_2^2}\left[J^\dagger\boldsymbol{\delta}_L\right]_i \mathbf{r}_{i-1}^T, \tag{99}$$

for which we used the same notation as in eq. (96) to divide the vector $J^\dagger\boldsymbol{\delta}_L$ in layerwise components. As the DFC update (99) is equal to the MN solution (96), we can conclude the proof. Note that because we used layer-specific learning rates $\eta_i = \frac{1}{\|\mathbf{r}_{i-1}\|_2^2}$ only the layerwise updates $\Delta W_i$ and $\Delta W_i^*$ align, not their concatenated versions $\Delta\bar{W}$ and $\Delta\bar{W}^*$. $\qquad\square$

Finally, we will remove Condition 2 and show in Proposition 4 (here repeated in Proposition S8 for convenience) that the weight updates still follow a descent direction for arbitrary feedback weights. Before proving Proposition 4, we need to introduce and prove the following Lemma.

**Lemma S7.** *Assuming $\tilde{J}_1$ is full rank,*

$$\lim_{\alpha\to 0} Q(JQ + \alpha I)^{-1} = U_Q \begin{bmatrix} \tilde{J}_1^{-1} \\ 0 \end{bmatrix} V_Q^T, \tag{100}$$

*with $U_Q$, $V_Q$ the left and right singular vectors of $Q$ and $\tilde{J}_1$ as defined as follows: consider $\tilde{J} = V_Q^T JU_Q$, the linear transformation of $J$ by the singular vectors of $Q$ which can be written in blockmatrix form $\tilde{J} = [\tilde{J}_1\tilde{J}_2]$ with $\tilde{J}_1$ a square matrix.*

*Proof.* For the proof, we use the singular value decomposition (SVD) of $Q$ and use it to rewrite $Q(JQ + \tilde{\alpha}I)^{-1}$. The SVD is given by $Q = U_Q\Sigma_Q V_Q^T$, with $V_Q$ and $U_Q$ square orthogonal matrices and $\Sigma_Q$ a rectangular diagonal matrix:

$$\Sigma_Q = \begin{bmatrix} \Sigma_Q^D \\ 0 \end{bmatrix}, \tag{101}$$

with $\Sigma_Q^D$ a square diagonal matrix, containing the singular values of $Q$. Now, let us define $\tilde{J}$ as

$$\tilde{J} \triangleq V_Q^T JU_Q, \tag{102}$$

such that $J = V_Q\tilde{J}U_Q^T$. $\tilde{J}$ can be structured into $\tilde{J} = [\tilde{J}_1\tilde{J}_2]$ with $\tilde{J}_1$ a square matrix. Now, we can rewrite $Q(JQ + \alpha I)^{-1}$ as

$$Q(JQ + \alpha I)^{-1} = U_Q\Sigma_Q V_Q^T\left(V_Q\tilde{J}U_Q^T U_Q\Sigma_Q V_Q^T + \alpha I\right)^{-1} \tag{103}$$

$$= U_Q\Sigma_Q\left(\tilde{J}\Sigma_Q + \alpha I\right)^{-1}V_Q^T \tag{104}$$

$$= U_Q \begin{bmatrix} \Sigma_Q^D \\ 0 \end{bmatrix}\left(\tilde{J}_1\Sigma_Q^D + \alpha I\right)^{-1}V_Q^T. \tag{105}$$

Assuming $\tilde{J}_1$ and $\Sigma_Q^D$ to be invertible (i.e., no zero singular values), this leads to:

$$\lim_{\alpha\to0} Q(JQ + \alpha I)^{-1} = U_Q \begin{bmatrix} \tilde{J}_1^{-1} \\ 0 \end{bmatrix} V_Q^T, \tag{106}$$

thereby concluding the proof. $\qquad\square$

This lemma shows clearly that $\lim_{\alpha\to0} Q(JQ + \alpha I)^{-1}$ is a generalized inverse of the forward Jacobian $J$, constrained by the column space of $Q$, which is represented by $U_Q$.

**Proposition S8.** *Assuming stable network dynamics and $\lambda, \alpha \to 0$, the steady-state weight updates $\Delta W_{i,\mathrm{ss}}$ (9) with a layer-specific learning rate $\eta_i = \eta/\|r_{i-1}\|_2^2$ lie always within 90 degrees of the loss gradient direction.*

*Proof.* First, we show that the steady-state weight update lies within 90 degrees of the loss gradient, after which we continue to prove convergence for linear networks. We define $\Delta\mathbf{v}_{\mathrm{ss}} \triangleq \mathbf{v}_{\mathrm{ss}} - \mathbf{v}_{\mathrm{ss}}^{\mathrm{ff}}$, which allows us to rewrite the steady-state update (9) as

$$\Delta\bar{W}_{\mathrm{ss}} = \eta M^{-2} R_{\mathrm{ss}} \Delta\mathbf{v}_{\mathrm{ss}}, \tag{107}$$

where we use the vectorized notation, $R_{\mathrm{ss}}$ defined in eq. (70) with steady-state activations, and $M$ defined in eq. (85) to represent the layer-specific learning rate $\eta_i = \eta/\|r_{i-1}\|_2^2$. Using Lemma 1 and S7, we have that

$$\lim_{\alpha\to0} \Delta\mathbf{v}_{\mathrm{ss}} = U_Q \begin{bmatrix} \tilde{J}_1^{-1} \\ 0 \end{bmatrix} V_Q^T \boldsymbol{\delta}_L. \tag{108}$$

Using the same vectorized notation, the negative gradient of the loss with respect to the network weights (i.e., the BP updates) can be written as:

$$\Delta\bar{W}^{BP} = \eta R J^T \boldsymbol{\delta}_L. \tag{109}$$

To show that the steady-state weight update lies within 90 degrees of the loss gradient, we prove that their inner product is greater than zero in the limit of $\lambda, \alpha \to 0$:

$$\lim_{\lambda,\alpha\to0} \langle \Delta\bar{W}^{BP}, \Delta\bar{W}_{\mathrm{ss}} \rangle = \lim_{\lambda,\alpha\to0} \eta^2 \boldsymbol{\delta}_L^T J R^T M^{-2} R_{\mathrm{ss}} \Delta\mathbf{v}_{\mathrm{ss}} \tag{110}$$

$$= \lim_{\lambda\to0} \eta^2 \boldsymbol{\delta}_L^T V_Q \tilde{J} \begin{bmatrix} \tilde{J}_1^{-1} \\ 0 \end{bmatrix} V_Q^T \boldsymbol{\delta}_L \tag{111}$$

$$= \lim_{\lambda\to0} \eta^2 \boldsymbol{\delta}_L^T \boldsymbol{\delta}_L > 0, \tag{112}$$

where we used that $\lim_{\lambda\to0} R^T M^{-2} R_{\mathrm{ss}} = I$ and took $\eta \propto 1/\lambda$ to have a limit different from zero, as $\boldsymbol{\delta}_L$ scales with $\lambda$.

$\qquad\square$

## A.4  An intuitive interpretation of Condition 2

In the previous sections, we showed that Condition 2 is needed to enable precise CA through GN or MN optimization. Here, we discuss a more intuitive interpretation of why Condition 2 is needed.

DFC has three main components that influence the feedback signals given to each neuron. First, we have the network dynamics (1) (here repeated for convenience).

$$\tau_v \frac{\mathrm{d}}{\mathrm{d}t} \mathbf{v}_i(t) = -\mathbf{v}_i(t) + W_i \phi\big(\mathbf{v}_{i-1}(t)\big) + Q_i \mathbf{u}(t) \quad 1 \le i \le L. \tag{113}$$

The first two terms $-\mathbf{v}_i(t) + W_i \phi\big(\mathbf{v}_{i-1}(t)\big)$ pull the neural activation $\mathbf{v}_i$ close to its feedforward compartment $\mathbf{v}_i^{\mathrm{ff}}$, while the third term $Q_i \mathbf{u}(t)$ provides an extra push such that the network output is driven to its target. This interplay between pulling and pushing is important, as it makes sure that $\mathbf{v}_i$ and $\mathbf{v}_i^{\mathrm{ff}}$ remain as close as possible together, while driving the output towards its target.

Second, we have the feedback weights $Q$. As $Q$ is of dimensions $\sum_{i=1}^{L} n_i \times n_L$, with $n_i$ the layer size, it has always much more rows than columns. Hence, the few but long columns of $Q$ can be seen as the 'modes' that the controller $\mathbf{u}$ can use to change network activations $\mathbf{v}$. Due to the low-dimensionality of $\mathbf{u}$ compared to $\mathbf{v}$, $Q\mathbf{u}$ cannot change the activations $\mathbf{v}$ in arbitrary directions, but is constrained by the column space of $Q$, i.e., the 'modes' of $Q$.

Third, we have the feedback controller, that through its own dynamics, combined with the network dynamics (1) and $Q$, selects an 'optimal' configuration for $\mathbf{u}$, i.e., $\mathbf{u}_{ss} = (JQ)^{-1}\boldsymbol{\delta}_L$, that selects and weights the different modes (columns) of $Q$ to push the output to its target in the 'most efficient manner'.

To make 'most efficient manner' more concrete, we need to define the *nullspace* of the network. As the dimension of $\mathbf{v}$ is much bigger than the output dimension, there exist changes in activation $\Delta\mathbf{v}$ that do not result in a change of output $\Delta\mathbf{r}_L$, because they lie in the nullspace of the network. In a linearized network, this is reflected by the network Jacobian $J$, as we have that $\Delta\mathbf{r}_L = J\Delta\mathbf{v}$. As J is of dimensions $n_L \times \sum_{i=1}^{L} n_i$, it has many more columns than rows and thus a non-zero nullspace. When $\Delta\mathbf{v}$ lies inside the nullspace of $J$, it will result in $\Delta\mathbf{r}_L = 0$. Now, if the column space of $Q$ overlaps partially with the nullspace of $J$, one could make $\mathbf{u}$, and hence $\Delta\mathbf{v} = Q\mathbf{u}$, arbitrarily big, while still making sure that the output is pushed exactly to its target, when the 'arbitrarily big' parts of $\Delta\mathbf{v}$ lie inside the nullspace of $J$ and hence do not influence $\mathbf{r}_L$. Importantly, the feedback controller combined with the network dynamics ensure that this does not happen, as $\mathbf{u}_{ss} = (JQ)^{-1}\boldsymbol{\delta}_L$ selects the smallest possible $\mathbf{u}_{ss}$ to push the output to its target.

However, when the column space of $Q$ partially overlaps with the nullspace of $J$, there will inevitably be parts of $\Delta\mathbf{v}$ that lie inside the nullspace of $J$, even though the controller selects the smallest possible $\mathbf{u}_{ss}$. This can easily be seen as in general, each column of $Q$ overlaps partially with the nullspace of $J$, so $\Delta\mathbf{v} = Q\mathbf{u}$, which is a linear combination of the columns of $Q$, will also overlap partially with the nullspace of $J$. This is where Condition 2 comes into play.

Condition 2 states that the column space of $Q$ is equal to the row space of $J$. When this condition is fulfilled, the column space of $Q$ does not overlap with the nullspace of $J$. Hence, all the feedback $Q\mathbf{u}$ produces a change in the network output and no unnecessary changes in activations $\Delta\mathbf{v}$ take place. With Condition 2 satisfied, the occurring changes in activations $\Delta\mathbf{v}$ are MN, as they lie fully in the row-space of $J$ and push the output exactly to its target. This interpretation lies at the basis of Theorem 3 and is also an important part of Theorem 2.

### A.5   Gauss-Newton optimization with a mini-batch size of 1

In this section, we review the GN optimization method and discuss the unique properties that arise when a mini-batch size of 1 is taken.

**Review of GN optimization.**   Gauss-Newton (GN) optimization is an iterative optimization method used for non-linear regression problems with an $L^2$ output loss, defined as follows:

$$\underset{\theta}{\arg\min} \quad \mathcal{L} = \frac{1}{2}\sum_{b=1}^{B} \|\boldsymbol{\delta}^{(b)}\|_2^2 \tag{114}$$

$$\boldsymbol{\delta}^{(b)} \triangleq \mathbf{y}^{(b)} - \mathbf{r}^{(b)}, \tag{115}$$

with B the minibatch size, $\boldsymbol{\delta}$ the regression error, $\mathbf{r}$ the model output, and $\mathbf{y}$ the corresponding regression target. There exist two main derivations of the GN optimization method: (i) through an approximation of the Newton-Raphson method and (ii) through linearizing the parametric model that is being optimized. We focus on the latter, as this derivation is closely connected to DFC.

GN is an iterative optimization method and hence aims to find a parameter update $\Delta\boldsymbol{\theta}$ that leads to a lower regression loss:

$$\boldsymbol{\theta}^{(m+1)} \leftarrow \boldsymbol{\theta}^{(m)} + \Delta\boldsymbol{\theta}, \tag{116}$$

with $m$ indicating the iteration number. The end goal of the optimization scheme is to find a local minimum of $\mathcal{L}$, hence, finding $\boldsymbol{\theta}^*$ for which holds

$$0 \overset{!}{=} \frac{\partial \mathcal{L}}{\partial \boldsymbol{\theta}}\Big|_{\boldsymbol{\theta}=\boldsymbol{\theta}^*}^T = J_{\boldsymbol{\theta}}^T \boldsymbol{\delta} \tag{117}$$

$$J_{\boldsymbol{\theta}} \triangleq \frac{\partial \mathbf{r}}{\partial \boldsymbol{\theta}}\Big|_{\boldsymbol{\theta}=\boldsymbol{\theta}^*}, \tag{118}$$

with $\boldsymbol{\delta}$ and $\mathbf{r}$ the concatenation of all $\boldsymbol{\delta}^{(b)}$ and $\mathbf{r}^{(b)}$, respectively. To obtain a closed-form expression for $\boldsymbol{\theta}^*$ that fulfills eq. (117) approximately, one can make a first-order Taylor approximation of the parameterize model around the current parameter setting $\boldsymbol{\theta}^{(m)}$:

$$\mathbf{r}^{(m+1)} \approx \mathbf{r}^{(m)} + J_{\boldsymbol{\theta}} \Delta \boldsymbol{\theta} \tag{119}$$

$$\boldsymbol{\delta}^{(m+1)} = \mathbf{y} - \mathbf{r}^{(m+1)} \approx \boldsymbol{\delta}^{(m)} - J_{\boldsymbol{\theta}} \Delta \boldsymbol{\theta}. \tag{120}$$

Filling this approximation into eq. (117), we get:

$$\frac{\partial \mathcal{L}}{\partial \boldsymbol{\theta}} \approx J_{\boldsymbol{\theta}}^T \big( \boldsymbol{\delta}^{(m)} - J_{\boldsymbol{\theta}} \Delta \boldsymbol{\theta} \big) = 0 \tag{121}$$

$$\Leftrightarrow \quad J_{\boldsymbol{\theta}}^T J_{\boldsymbol{\theta}} \Delta \boldsymbol{\theta} = J_{\boldsymbol{\theta}}^T \boldsymbol{\delta}^{(m)}. \tag{122}$$

In an under-parameterized setting, i.e., the dimension of $\boldsymbol{\delta}$ is bigger than the dimension of $\boldsymbol{\theta}$, $J_{\boldsymbol{\theta}}^T J_{\boldsymbol{\theta}}$ can be interpreted as an approximation of the loss Hessian matrix used in the Newton-Raphson method and is known as the *Gauss-Newton curvature matrix*. In the under-parameterized setting, $J_{\boldsymbol{\theta}}^T J_{\boldsymbol{\theta}}$ is invertible, leading to the update

$$\Delta \boldsymbol{\theta} = \big( J_{\boldsymbol{\theta}}^T J_{\boldsymbol{\theta}} \big)^{-1} J_{\boldsymbol{\theta}}^T \boldsymbol{\delta}^{(m)} \tag{123}$$

$$\Delta \boldsymbol{\theta} = J_{\boldsymbol{\theta}}^\dagger \boldsymbol{\delta}^{(m)}, \tag{124}$$

with $J_{\boldsymbol{\theta}}^\dagger$ the Moore-Penrose pseudoinverse of $J_{\boldsymbol{\theta}}$. In the under-parameterized setting, eq. (122) can be interpreted as a linear least-squares regression for finding a parameter update $\Delta \boldsymbol{\theta}$ that results in a least-squares solution on the linearized parametric model (119). Until now we considered the under-parameterized case. However, DFC is related to GN optimization with a mini-batch size of 1, which concerns the over-parameterized case.

**GN optimization with a mini-batch size of 1.** When the minibatch size $B = 1$, the dimension of $\boldsymbol{\delta}$ is smaller than the dimension of $\boldsymbol{\theta}$ in neural networks, hence we need to consider the over-parameterized case of GN [36, 57]. Now, the matrix $J_{\boldsymbol{\theta}}^T J_{\boldsymbol{\theta}}$ is not of full rank and hence an infinite amount of solutions exist for eq. (122). To enforce a unique solution for the parameter update $\Delta \boldsymbol{\theta}$, a common approach is to take the MN solution, i.e., the smallest possible solution $\Delta \boldsymbol{\theta}$ that satisfies (122). Using the MN properties of the Moore-Penrose pseudoinverse, this results in:

$$\Delta \boldsymbol{\theta} = J_{\boldsymbol{\theta}}^\dagger \boldsymbol{\delta}^{(m)}. \tag{125}$$

Although the solution has the same form as before (124), its interpretation is fundamentally different, as we did not use a linear least-squares solution, but a MN solution instead. In the under-parameterized case considered before, the parameter update $\Delta \boldsymbol{\theta}$ will not be able to drive $\boldsymbol{\delta}^{(m+1)}$ to zero (in the linearized model). In the over-parameterized case however, there exist many solutions for $\Delta \boldsymbol{\theta}$ that drive $\boldsymbol{\delta}^{(m+1)}$ exactly to zero, and GN picks the MN solution (125).

With this interpretation, we see clearly the connection to DFC. In DFC, the feedback controller drives the network activations (i.e., finds an 'activation update' solution) such that the output of the network reaches its target $\mathbf{r}_L^*$ (i.e., the error $\boldsymbol{\delta}^{(m+1)}$ is driven to zero). When Condition 2 holds, this activation update solution is the MN solution. Furthermore, when Condition 1 holds, this MN activation update results also in a MN parameter update $\Delta W_{i,\text{ss}}$ (9).

**DFC updates with larger batch sizes.** For computational efficiency, we average the DFC updates over a minibatch size bigger than 1. However, this averaging over a minibatch is distinct from doing Gauss-Newton optimization on a minibatch. The GN iteration with minibatch size $B$ is given by

$$\Delta \bar{W}^{GN} = \lim_{\gamma \to 0} \Big[ \sum_{b=1}^B J_{\bar{W}}^{(b)T} J_{\bar{W}}^{(b)} + \gamma I \Big]^{-1} \Big[ \sum_{b=1}^B J_{\bar{W}}^{(b)T} \boldsymbol{\delta}_L^{(b)} \Big], \tag{126}$$

with $J_{\bar{W}}^{(b)}$ the Jacobian of the output w.r.t. the concatenated weights $\bar{W}$ for batch sample $b$, and $\gamma$ a damping parameter. Note that we accumulate the GN curvature $J_{\bar{W}}^{(b)T} J_{\bar{W}}^{(b)}$ over all minibatch samples before taking the inverse.

When the assumptions of Theorem 2 hold, the DFC updates with a minibatch size $B$ can be written by

$$\Delta\bar{W} = \lim_{\gamma \to 0} \sum_{b=1}^{B} \left[ \left( J_{\bar{W}}^{(b)T} J_{\bar{W}}^{(b)} + \gamma I \right)^{-1} J_{\bar{W}}^{(b)T} \boldsymbol{\delta}_L^{(b)} \right] \tag{127}$$

$$= \sum_{b=1}^{B} \left[ J_{\bar{W}}^{(b)\dagger} \boldsymbol{\delta}_L^{(b)} \right]. \tag{128}$$

For $B = 1$, the DFC update (127) overlaps with the GN update (126). However, for $B > 1$ these are not equal anymore, due to the order of summation and inversion being reversed.

### A.6 Effects of the nonlinearity $\phi$ in the weight update

In this section, we study in detail the experimental consequences of using the nonlinear learning rule (2.3) instead of the linear learning rule (9). First, we investigate the case where the assumptions in Theorem 3 are perfectly satisfied and then we investigate the more realistic case where the assumptions are not perfectly satisfied.

When considering the ideal case where Condition 2 is perfectly satisfied and in the limit of $\lambda$ and $\alpha$ to zero, MN updates (214) are obtained if the linear learning rule is used, and the following updates are obtained when the nonlinear learning rule is used:

$$\Delta\bar{W} = RDJ^T (JJ^T)^{-1} \boldsymbol{\delta}_L, \tag{129}$$

with $D$ a diagonal matrix with $\partial\phi(v_j)/\partial(v_j)$ for each neuron in the network on its diagonal and $R$ as defined in eq. (214). For this ideal case, we performed experiments on MNIST comparing the linear to the nonlinear learning rules, and obtained a test error of $2.18^{\pm 0.14}\%$ and $2.11^{\pm 0.10}\%$, respectively. These experiments demonstrate that for this ideal case the nonlinear learning rule (2.3) has no significant benefit over the linear learning rule (9).

On the other hand, to investigate the influence of the nonlinear learning rule for the practical case where Condition 2 is not perfectly satisfied, we performed a new hyperparameter search on MNIST for DFC-SSA with the linear learning rule (9). This resulted in a test error of $5.28^{\pm 0.14}\%$. Comparing this result with the corresponding test performance in Table 1 ($2.29^{\pm 0.097}\%$ test error), we conclude that DFC benefits from the introduction of the chosen nonlinearities in the learning rule (2.3), as the results improve significantly. Hence, we can infer that this increase in performance is due to the way the introduction of the nonlinearity in the learning rule compensates for when the feedback weights do not perfectly satisfy Condition 2.

Lastly, to investigate where this performance gap originates from, we performed another toy experiment similar to Fig. 3 (see Fig. S1) for the linear versus nonlinear learning rule in DFC. The new results show that the updates resulting from the nonlinear learning rule are much better aligned with the MN and GN updates, compared to the linear learning rule, explaining its better performance. Overall, we conclude that introducing the nonlinearity in the learning rule, which prevents saturated neurons from updating their weights, is a useful heuristic to improve the alignment of DFC with the MN and GN updates and consequently improve its performance, when Condition 2 is not perfectly satisfied.

### A.7 Relation between continuous DFC weight updates and steady-state DFC weight updates

All developed learning theory in section 3 considers an update $\Delta W_i$ at the steady-state of the network (1) and controller (4) dynamics instead of a continuous update as defined in (5). Fig. 3F shows that the accumulated continuous updates (5) of DFC align well with the analytical steady-state updates. Here, we indicate why this steady-state update is a good approximation of the accumulated continuous updates (5). We consider two main reasons: (i) the network and controller dynamics settle quickly to their steady-state and (ii) when the dynamics are not settled yet, they oscillate around the steady-state, thereby causing oscillations to cancel each other out approximately.

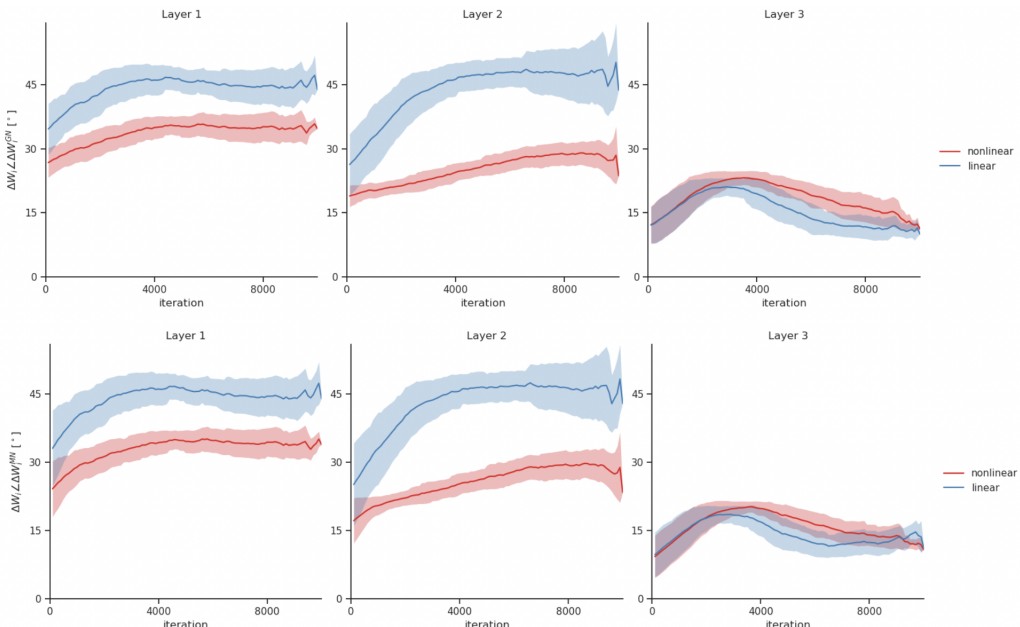

Figure S1: Layer-wise comparison of the angle alignment between the DFC updates and the GN and MN updates, for the linear and nonlinear learning rule variants, when performing nonlinear student-teacher regression task.

Addressing the first reason, consider an input that is presented to the network from time $T_1$ until $T_2$ and that the network and controller dynamics converge at $T_{ss} < T_2$. The change in weight prescribed by (5) is then equal to

$$\int_{T_1}^{T_2} \mathrm{d}W_i = \int_{T_1}^{T_{ss}} \mathrm{d}W_i + \frac{T_2 - T_{ss}}{\tau_W}\big(\phi(\mathbf{v}_{i,\mathrm{ss}}) - \phi(\mathbf{v}_{i,\mathrm{ss}}^{\mathrm{ff}})\big)\mathbf{r}_{i-1,\mathrm{ss}}^T, \tag{130}$$

where we assumed a separation of timescales, i.e., $W_i$ is considered constant over the integration interval. If the dynamics settle quickly, i.e., $T_{ss} - T_1 \ll T_2 - T_{ss}$, the second term in the right-hand-side (RHS) dominates and we have that $\int_{T_1}^{T_2} \mathrm{d}W_i \approx \eta\big(\phi(\mathbf{v}_{i,\mathrm{ss}}) - \phi(\mathbf{v}_{i,\mathrm{ss}}^{\mathrm{ff}})\big)\mathbf{r}_{i-1,\mathrm{ss}}^T$ with $\eta = (T_2 - T_{ss})/\tau_W$.

Addressing the second reason, note that Fig. 1D shows qualitatively that the dynamics oscillate briefly around the steady-state before settling. The first integral term in the RHS of eq. (130) hence integrates over oscillating variables around the steady-state value. These oscillations will partially cancel each other out in the integration, causing $\int_{T_1}^{T_{ss}} \mathrm{d}W_i$ to be approximately equal to $\eta\big(\phi(\mathbf{v}_{i,\mathrm{ss}}) - \phi(\mathbf{v}_{i,\mathrm{ss}}^{\mathrm{ff}})\big)\mathbf{r}_{i-1,\mathrm{ss}}^T$ for some $\eta$.

## A.8 DFC is compatible with various controller types

Throughout the main manuscript, we focused on a proportional-integral (PI) controller. However, the DFC framework is compatible with various other controller types. In the following, we show that the results on learning theory (Section 3 can be generalized to pure integral control, pure proportional control or any combination thereof with derivative control added. Note that for each new controller type, a new stability analysis is needed and whether the feedback learning rule is still compatible with the controller also needs to be checked, which we leave to future work.

### A.8.1 Pure integral control

For pure integral control, the steady-state solutions of Lemma 1 still apply, with $\tilde{\alpha} = \alpha$. Hence, all learning theory results of Section 3 directly apply to this case. Furthermore, Proposition 5 and Theorem 6 are already designed for pure integral control.

### A.8.2 Pure proportional control

By making a first-order Taylor approximation of the network dynamics with only proportional control (putting $K_I = 0$ in eq. (4)), we obtain the following steady-state solution:

$$\mathbf{v}_{\mathrm{ss}} = \mathbf{v}_{\mathrm{ss}}^{\mathrm{ff}} + (QJ + \frac{1}{k_p}I)^{-1}Q\boldsymbol{\delta}_L + \mathcal{O}(\lambda^2), \tag{131}$$

with $k_p$ the proportional gain, i.e., $\mathbf{u} = k_p\mathbf{e}$. When comparing with eq. (6), we see that the steady-state solution only with proportional control has a similar structure to the one with PI control, with $\tilde{\alpha}$ replaced by $\frac{1}{k_p}$. Note, however, that the damped inverse of $QJ$ is taken instead of $JQ$. By using a similar proof technique as for Lemma S2, we can show that $\lim_{k_p \to \infty}(QJ + \frac{1}{k_p}I)^{-1}Q = J^\dagger$ iff Condition 2 holds.[7] Consequently, Theorems 2 and 3 and Proposition 4 hold also for proportional control, if the limit of $\alpha$ to zero is replaced by the limit of $k_p$ to infinity. Furthermore, the main intuitions of Theorem 6 for training the feedback can be applied to proportional control, given that one finds a way to keep the network stable during the initial feedback weights training phase.

Despite these theoretical similarities between proportional and PI control in DFC, there are some significant practical differences. First, for finite $k_p$ in proportional control, there is always a residual error that remains and hence the output target will never be exactly reached. Second, if noise is present in the network, it gets amplified by the same factor $k_p$. Hence, using a high $k_p$ in proportional control makes the controlled network sensitive to noise. Adding an integral control component can alleviate these issues by replacing the need for a large gain, $k_p$, with the need for a good integrator circuit (i.e., low $\alpha$) [34], for which a rich neuroscience literature exists [58, 59, 60, 61, 62]. This way, we can use a smaller gain, $k_p$, without increasing the residual error and consequently make the network less sensitive to noise. This is also interesting from a biological point of view since biological networks are considered to be substantially noisy.

### A.8.3 Adding derivative control

Proportional, integral or proportional-integral control can be combined with derivative control. As the derivative term disappears at the steady state, the steady-state solutions of Lemma 1 remain unaltered and the learning theory results can be directly applied. However, note that the derivative control term can significantly impact the stability and feedback learning of the network.

## B  Proofs and extra information for Section 4: Stability of DFC

### B.1  Stability analysis with instantaneous system dynamics

In this section, we first derive eq. (11), which corresponds to the dynamics of the controller obtained when assuming a separation of timescales between the controller and the network ($\tau_u \gg \tau_v$), and only having integrative control ($k_p = 0$).

Let us recall that $\mathbf{v}_{\mathrm{ss}}$ and $\mathbf{v}^-$ are the steady-state solutions of the dynamical system (1) with and without control, respectively. Now, by linearizing the network dynamics (1) around the feedforward steady-state, $\mathbf{v}^-$, we can write

$$\mathbf{r}_L = \mathbf{r}_L^- + JQ\mathbf{u} + \mathcal{O}(\lambda^2), \tag{132}$$

with $J \triangleq \left[\frac{\partial \mathbf{r}_L^-}{\partial \mathbf{v}_1}, ..., \frac{\partial \mathbf{r}_L^-}{\partial \mathbf{v}_L}\right]\Big|_{\mathbf{v}=\mathbf{v}^-}$ the network Jacobian evaluated at the steady state, and where we dropped the time dependence $(t)$ for conciseness.

Taking into account the results of equations (3) and (132), the control error can then be rewritten as

$$\mathbf{e} = \mathbf{r}_L^* - \mathbf{r}_L = (\mathbf{r}_L^* - \mathbf{r}_L^-) - (\mathbf{r}_L - \mathbf{r}_L^-) = \boldsymbol{\delta}_L - JQ\mathbf{u} + \mathcal{O}(\lambda^2). \tag{133}$$

Consequently, eq. (11) follows:

$$\tau_u\dot{\mathbf{u}} = \mathbf{e} - \alpha\mathbf{u} = \boldsymbol{\delta}_L - JQ\mathbf{u} - \alpha\mathbf{u} + \mathcal{O}(\lambda^2) = \boldsymbol{\delta}_L - (JQ + \alpha I)\mathbf{u} + \mathcal{O}(\lambda^2), \tag{134}$$

where we changed the notation $\frac{\mathrm{d}}{\mathrm{d}t}\mathbf{u}$ to $\dot{\mathbf{u}}$ for conciseness. Now, we continue by proving Proposition 5, restated below for convenience.

---

[7]We leave the proof as an exercise for the interested reader. The proof follows the same approach as Lemma S2 and uses l'Hôpital's rule for taking the correct limit of $k \to \infty$.

**Proposition S9** (Proposition 5 in main manuscript). *Assuming $\tau_u \gg \tau_v$ and $k_p = 0$, the network and controller dynamics are locally asymptotically stable around its equilibrium iff Condition 3 holds.*

*Proof.* Assuming instantaneous system dynamics ($\tau_u \gg \tau_v$), then the stability of the system is entirely up to the controller dynamics. To prove that the system's equilibrium is locally asymptotically stable, we need to guarantee that the Jacobian associated to the controller dynamics evaluated at its steady-state solution, $\mathbf{v}_{ss}$, has only eigenvalues with a strictly negative real part [38]. This Jacobian can be obtained in a similar fashion to that of eq. (11), and is given by

$$J_u = -(J_{ss}Q + \alpha I), \quad \text{with} \quad J_{ss} \triangleq \left[ \frac{\partial \mathbf{r}_L^-}{\partial \mathbf{v}_1}, ..., \frac{\partial \mathbf{r}_L^-}{\partial \mathbf{v}_L} \right]\Bigg|_{\mathbf{v}=\mathbf{v}_{ss}}. \tag{135}$$

To fulfill the local asymptotic stability condition, $J_{ss}Q + \alpha I$ can only have eigenvalues with strictly positive real parts. As adding $\alpha I$ to $J_{ss}Q$ results in adding $\alpha$ to the eigenvalues of $J_{ss}Q$, the local asymptotic stability condition requires that the real parts of the eigenvalues of $J_{ss}Q$ are all greater than $-\alpha$, corresponding to Condition 3. $\qquad\square$

### B.2 Stability of the full system

In this section, we derive a concise representation of the full dynamics of the network (1) and controller dynamics (4) in the general case where the timescale of the neuronal dynamics, $\tau_v$, is not negligible and we have proportional control ($k_p > 0$). Proposition S10 provides the abstract conditions that guarantee local asymptotic stability of the steady states of the full dynamical system.

**Proposition S10.** *The network and controller dynamics are locally asymptotically stable around its equilibrium iff the following matrix has strictly negative eigenvalues:*

$$A_{PI} = \begin{bmatrix} -\frac{1}{\tau_v}(I - \hat{J}_{ss}) & \frac{1}{\tau_v}(I - \hat{J}_{ss})Q \\ J_{ss}\left((\frac{k_p}{\tau_v} - \frac{1}{\tilde{\tau}_u})I - \frac{k_p}{\tau_v}\hat{J}_{ss}\right) & -\frac{k_p}{\tau_v}J_{ss}(I - \hat{J}_{ss})Q - \frac{\tilde{\alpha}}{\tilde{\tau}_u}I \end{bmatrix} \tag{136}$$

*with $\tilde{\alpha} = \frac{\alpha}{1+k_p\alpha}$, $\tilde{\tau}_u = \frac{\alpha}{1+k_p\alpha}$, $J_{ss} = \frac{\partial \mathbf{r}_L}{\partial \mathbf{v}}\big|_{\mathbf{v}=\mathbf{v}_{ss}}$ and $\hat{J}_{ss}$ defined in equations (143) and (148).*

*Proof.* Recall that the controller is given by (4)

$$\mathbf{u} = \mathbf{u}^{\text{int}} + k_p\mathbf{e}, \tag{137}$$

where $\tau_u\dot{\mathbf{u}}^{\text{int}} = \mathbf{e} - \alpha\mathbf{u}^{\text{int}}$. Then, the controller dynamics can be written as

$$\begin{aligned} \dot{\mathbf{u}} &= \frac{1}{\tau_u}\big(\mathbf{e} - \alpha(\mathbf{u} - k_p\mathbf{e})\big) + k_p\dot{\mathbf{e}} \\ \Leftrightarrow \tau_u\dot{\mathbf{u}} &= (1 + \alpha k_p)\mathbf{e} + k_p\tau_u\dot{\mathbf{e}} - \alpha\mathbf{u} \\ \Leftrightarrow \tilde{\tau}_u\dot{\mathbf{u}} &= \mathbf{e} + k_p\tilde{\tau}_u\dot{\mathbf{e}} - \tilde{\alpha}\mathbf{u}, \end{aligned} \tag{138}$$

with $\tilde{\tau}_u = \tau_u/(1 + \alpha k_p)$ and $\tilde{\alpha} = \alpha/(1 + \alpha k_p)$.

Recall that the network dynamics are given by (1)

$$\tau_v\dot{\mathbf{v}}_i = -\mathbf{v}_i + W_i\phi(\mathbf{v}_{i-1}) + Q_i\mathbf{u} = -\Delta\mathbf{v}_i + Q_i\mathbf{u}, \tag{139}$$

with $\Delta\mathbf{v}_i = \mathbf{v}_i - W_i\phi(\mathbf{v}_{i-1})$. Which allows us to write

$$\Delta\dot{\mathbf{v}}_i = \dot{\mathbf{v}}_i - W_iD(\mathbf{v}_{i-1})\dot{\mathbf{v}}_{i-1}, \quad \text{with} \quad D(\mathbf{v}_{i-1}) \triangleq \frac{\partial\phi(\mathbf{v}_{i-1})}{\partial\mathbf{v}_{i-1}}\Big|_{\mathbf{v}_{i-1}=\mathbf{v}_{i-1}(t)}. \tag{140}$$

We can now obtain the network dynamics in terms of $\Delta\dot{\mathbf{v}}$ as

$$\begin{aligned} \tau_v\Delta\dot{\mathbf{v}}_i &= -\Delta\mathbf{v}_i + Q_i\mathbf{u} - W_iD(\mathbf{v}_{i-1})(\Delta\mathbf{v}_{i-1} + Q_{i-1}\mathbf{u}) \\ &= -\Delta\mathbf{v}_i + W_iD(\mathbf{v}_{i-1})\Delta\mathbf{v}_{i-1} + (Q_i - W_i\Delta\mathbf{v}_{i-1}Q_{i-1})\mathbf{u}, \end{aligned} \tag{141}$$

which for the entire system is

$$\tau_v\Delta\dot{\mathbf{v}} = -(I - \hat{J}(\mathbf{v}))\Delta\mathbf{v} + (I - \hat{J}(\mathbf{v}))Q\mathbf{u}, \tag{142}$$

with

$$\hat{J}(\mathbf{v}) \triangleq \begin{bmatrix} 0 & 0 & 0 & \dots & 0 \\ W_2 D(\mathbf{v}_1) & 0 & 0 & \dots & 0 \\ 0 & W_3 D(\mathbf{v}_2) & 0 & \dots & 0 \\ \vdots & \ddots & \ddots & \ddots & \vdots \\ 0 & \dots & 0 & W_L D(\mathbf{v}_{L-1}) & 0 \end{bmatrix} \tag{143}$$

Let us now proceed to linearize the network and controller dynamical systems by defining

$$\tilde{\Delta}\mathbf{v} = \Delta\mathbf{v} - \Delta\mathbf{v}_{\text{ss}} \quad \text{and} \quad \tilde{\Delta}\mathbf{u} = \mathbf{u} - \mathbf{u}_{\text{ss}}, \tag{144}$$

with $\mathbf{u}_{\text{ss}}$ and $\Delta\mathbf{v}_{\text{ss}}$ the steady states of the network and controller (c.f. Lemma 1). With a first order Taylor approximation, we can write $\mathbf{r}_L \approx \mathbf{r}_{L,\text{ss}} + J_{\text{ss}}\tilde{\Delta}\mathbf{v}$, where $J_{\text{ss}}$ is as in eq. (135).

The controller dynamics (138) can now be rewritten as

$$\tilde{\tau}_u \tilde{\Delta}\dot{\mathbf{u}} = \mathbf{r}_L^* - \mathbf{r}_{L,\text{ss}} - J_{\text{ss}}\tilde{\Delta}\mathbf{v} - \tilde{\tau}_u k_p J_{\text{ss}}\tilde{\Delta}\dot{\mathbf{v}} - \tilde{\alpha}\mathbf{u}_{\text{ss}} - \tilde{\alpha}\tilde{\Delta}\mathbf{u}. \tag{145}$$

When the network and the controller are at equilibrium, eq. (138) yields

$$0 = \mathbf{e}_{\text{ss}} - \tilde{\alpha}\mathbf{u}_{\text{ss}} = \mathbf{r}_L^* - \mathbf{r}_{L,\text{ss}} - \tilde{\alpha}\mathbf{u}_{\text{ss}}, \tag{146}$$

and we can rewrite eq. (145) as

$$\tilde{\tau}_u \tilde{\Delta}\dot{\mathbf{u}} = -J_{\text{ss}}\tilde{\Delta}\mathbf{v} - \tilde{\tau}_u k_p J_{\text{ss}}\tilde{\Delta}\dot{\mathbf{v}} - \tilde{\alpha}\tilde{\Delta}\mathbf{u}. \tag{147}$$

Once again, when the network and the controller are at equilibrium, incorporating the definitions in (144) into eq. (142), it follows that

$$\tau_v \tilde{\Delta}\dot{\mathbf{v}} = -(I - \hat{J}_{\text{ss}})(\Delta\mathbf{v}_{\text{ss}} + \tilde{\Delta}\mathbf{v}) + (I - \hat{J}_{\text{ss}})Q(\mathbf{u}_{\text{ss}} + \tilde{\Delta}\mathbf{u}), \quad \hat{J}_{\text{ss}} \triangleq \hat{J}(\mathbf{v}_{\text{ss}}). \tag{148}$$

At steady-state, eq. (142) yields

$$0 = -(I - \hat{J}_{\text{ss}})\Delta\mathbf{v}_{\text{ss}} + (I - \hat{J}_{\text{ss}})Q\mathbf{u}_{\text{ss}}, \tag{149}$$

which allows us to rewrite eq. (148) as

$$\tau_v \tilde{\Delta}\dot{\mathbf{v}} = -(I - \hat{J}_{\text{ss}})\tilde{\Delta}\mathbf{v} + (I - \hat{J}_{\text{ss}})Q\tilde{\Delta}\mathbf{u}. \tag{150}$$

Using the results from eq. (150), we can write eq. (147) as

$$\tilde{\tau}_u \tilde{\Delta}\dot{\mathbf{u}} = -J_{\text{ss}}\tilde{\Delta}\mathbf{v} - \frac{\tilde{\tau}_u}{\tau_v} k_p J_{\text{ss}}\Big( -(I - \hat{J}_{\text{ss}})\tilde{\Delta}\mathbf{v} + (I - \hat{J}_{\text{ss}})Q\tilde{\Delta}\mathbf{u} \Big) - \tilde{\alpha}\tilde{\Delta}\mathbf{u}. \tag{151}$$

Finally, as $\tilde{\Delta}\dot{\mathbf{v}} = \Delta\dot{\mathbf{v}} = \dot{\mathbf{v}}$ and $\tilde{\Delta}\dot{\mathbf{u}} = \dot{\mathbf{u}}$ (144), this allows us to to infer local stability results for the full system dynamics by looking into the dynamics of $\tilde{\Delta}\dot{\mathbf{v}}$ and $\tilde{\Delta}\dot{\mathbf{u}}$ around the steady state:

$$\begin{bmatrix} \tilde{\Delta}\dot{\mathbf{v}} \\ \tilde{\Delta}\dot{\mathbf{u}} \end{bmatrix} = \begin{bmatrix} -\frac{1}{\tau_v}(I - \hat{J}_{\text{ss}}) & \frac{1}{\tau_v}(I - \hat{J}_{\text{ss}})Q \\ J_{\text{ss}}\big((\frac{k_p}{\tau_v} - \frac{1}{\tilde{\tau}_u})I - \frac{k_p}{\tau_v}\hat{J}_{\text{ss}}\big) & -\frac{k_p}{\tau_v}J_{\text{ss}}(I - \hat{J}_{\text{ss}})Q - \frac{\tilde{\alpha}}{\tilde{\tau}_u}I \end{bmatrix} \begin{bmatrix} \tilde{\Delta}\mathbf{v} \\ \tilde{\Delta}\mathbf{u} \end{bmatrix} \triangleq A_{PI} \begin{bmatrix} \tilde{\Delta}\mathbf{v} \\ \tilde{\Delta}\mathbf{u} \end{bmatrix} \tag{152}$$

Now, to guarantee local asymptotic stability of the system's equilibrium, then the eigenvalues of $A_{PI}$ must have strictly negative real parts [38]. $\qquad \square$

The current form of the system matrix $A_{PI}$ provides no straightforward intuition on finding interpretable conditions for the feedback weights $Q$ such that local stability is reached. One can apply Gershgoring's circle theorem to infer sufficient restrictions on $J$ and $Q$ to ensure local asymptotic stability [63]. However, the resulting conditions are too conservative and do not provide intuition in which types of feedback learning rules are needed to ensure stability.

### B.3 Toy experiments for relation of Condition 3 and full system dynamics

To investigate whether Condition 3 is a good proxy for the local stability of the actual dynamics, we plotted the maximum real parts of the eigenvalues of $JQ + \alpha I$ (Condition 3, see Fig. S2.a) and of $A_{PI}$ (the actual dynamics, see eq. (136) and Fig. S2.b). We used the same student-teacher regression setting and configuration as in the toy experiments of Fig. 3.

Fig. S2 shows that the maximum real part of the eigenvalues of $A_{PI}$ follow the same trend as the eigenvalues of $JQ + \alpha I$. Although they differ in exact value, both eigenvalue trajectories are slowly decreasing during training and are strictly negative, thereby indicating that Condition 3 is a good proxy for the local stability of the actual dynamics.

When we only consider leaky integral control ($k_p = 0$, see Fig. S2.c), the dynamics become unstable during late training, highlighting that adding proportional control is crucial for the stability of the dynamics. Interestingly, training the feedback weights (blue curve) does not help in this case for making the system stable, on the contrary, it pushes the network to become unstable more quickly. These leaky integral control dynamics are equal to the simplified dynamics used in Condition 3 in the limit of $\tau_v/\tau_u \to 0$, which are stable (see Fig. S2.a). Hence, slower network dynamics (finite time constant $\tau_v$) cause the leaky integral control to become unstable, due to a communication delay between controller and network, causing unstable oscillations. For this toy experiment, we used $\tau_v/\tau_u = 0.2$.

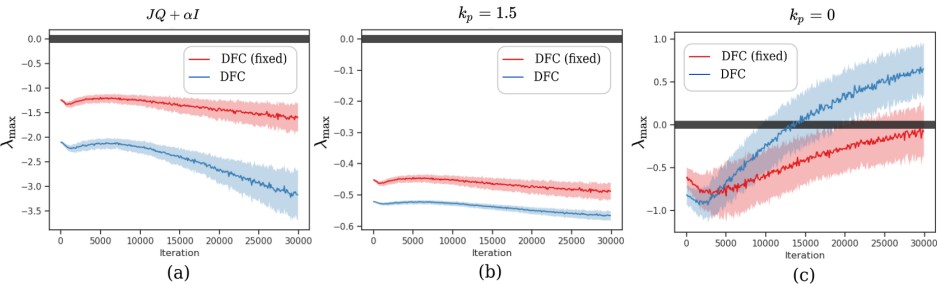

Figure S2: Visualization of the maximum real parts of the eigenvalues of the system matrices during the student teacher regression for DFC with the same configuration as in Fig. 3. (a) The maximum real part of the eigenvalues of $JQ + \alpha I$, hence representing Condition 3. (b) The maximum real part of the eigenvalues of the system dynamics matrix $A_{PI}$ (136), representing the actual local stability of DFC. (c) The maximum real part of the eigenvalues of the system dynamics matrix $A_{PI}$ (136) without proportional control ($k_p = 0$), representing the local stability of DFC with only integral control. The black bar represents the local stability threshold.

## C  Proofs and extra information for Section 5: Learning the feedback weights

### C.1  Learning the feedback weights in a sleep phase

In this section, we show that the plasticity rule for the apical synapses (13) drives the feedback weights to fulfill Conditions 2 and 3. We first sketch an intuitive argument on why the feedback learning rule works. Next, we state the full Theorem and give its proof.

#### C.1.1  Intuition behind the feedback learning rule

Inspired by the Weight Mirroring Method [14] we use white noise in the network to carry information from the network Jacobian $J$ into the output $\mathbf{r}_L$. To gain intuition, we first consider a normal feedforward neural network

$$\mathbf{r}_i^- = \phi(\mathbf{v}_i^-) = \phi(W_i \mathbf{r}_{i-1}^-), \quad 1 \le i \le L. \tag{153}$$

Now, we perturb each layer's pre-nonlinearity activation with white noise $\boldsymbol{\xi}_i$ and propagate the perturbations forward:

$$\tilde{\mathbf{v}}_i^- = W_i \phi(\tilde{\mathbf{v}}_{i-1}^-) + \sigma \boldsymbol{\xi}_i, \quad 1 \le i \le L, \tag{154}$$

with $\tilde{\mathbf{r}}_0^- = \mathbf{r}_0^-$. For small $\sigma$, a first-order Taylor approximation of the perturbed output gives

$$\tilde{\mathbf{r}}_L^- = \mathbf{r}_L^- + \sigma J\boldsymbol{\xi} + \mathcal{O}(\sigma^2), \tag{155}$$

with $\boldsymbol{\xi}$ the concatenated vector of all $\boldsymbol{\xi}_i$. If we now take as output target $\mathbf{r}_L^* = \mathbf{r}_L^-$, the output error is equal to

$$\mathbf{e} = \mathbf{r}_L^* - \tilde{\mathbf{r}}_L^- = -\sigma J\boldsymbol{\xi} + \mathcal{O}(\sigma^2). \tag{156}$$

We now define a simple learning rule $\Delta Q = -\sigma \boldsymbol{\xi}\mathbf{e}^T - \beta Q$, which is a simple anti-Hebbian rule with as presynaptic signal the output error $\mathbf{e}$ and as postsynaptic signal the noise inside the neuron $\sigma\boldsymbol{\xi}$, combined with weight decay. If $\boldsymbol{\xi}$ is uncorrelated white noise with correlation matrix equal to the identity matrix, the expectation of this learning rule is

$$\mathbb{E}[\Delta Q] = \sigma^2 J^T - \beta Q. \tag{157}$$

We see that this learning rule lets the feedback weights $Q$ align with the transpose of the networks Jacobian $J$ and has a weight decay term to prevent $Q$ from diverging.

There are three important differences between this simplified intuitive argumentation for the feedback learning rule and the actual feedback learning rule (13) used by DFC, which we will address in the next section.

1. DFC considers continuous dynamics, hence, the incorporation of noise leads to stochastic differential equations (SDEs) instead of a discrete perturbation of the network layers. The handling of SDEs needs special care, leading to the use of exponentially filtered white noise instead of purely white noise (see next section).

2. The postsynaptic part of the feedback learning rule (13) for DFC is the control signal $\mathbf{u}$ instead of the output error $\mathbf{e}$. The control signal integrates the output error over time, causing correlations over time to arise in the feedback learning rule.

3. The presynaptic part of the feedback learning rule (13) for DFC is the feedback compartment $\mathbf{v}^{\text{fb}}$ which consists of both the controller input $Q\mathbf{u}$ and the noise instead of only the noise. This will lead to an extra term in the expectation $\mathbb{E}[\Delta Q]$, which results in $Q$ aligning with $J^T(JJ^T + \gamma I)^{-1}, \gamma > 0$ instead of $J^T$.

### C.1.2 Theorem and proof

**Noise dynamics.** For simplicity of the argument and proof, we assume that the noise only enters through the feedback compartment of the neuron. The proposed theorem below and its proof also holds when extra noise is added in the feedforward and central compartment, as long as it is independent from the noise in the feedback compartment. Throughout the main manuscript, we assumed instantaneous dynamics of the feedback compartment, i.e., $\mathbf{v}_i^{\text{fb}}(t) = Q_i\mathbf{u}(t)$. If we now add white noise to the feedback compartment, the limit to instantaneous dynamics is not well defined anymore ($\lim_{\tau_{v^{\text{fb}}} \to 0}$ in eq. (158)). Hence, we introduce the following dynamics for the feedback compartment

$$\tau_{v^{\text{fb}}} \frac{\mathrm{d}}{\mathrm{d}t}\mathbf{v}_i^{\text{fb}}(t) = -\mathbf{v}_i^{\text{fb}}(t) + Q_i\mathbf{u}(t) + \sigma\boldsymbol{\xi}_i(t). \tag{158}$$

The network dynamics (1) are now given by

$$\tau_v \frac{\mathrm{d}\mathbf{v}_i(t)}{\mathrm{d}t} = -\mathbf{v}_i(t) + W_i\phi(\mathbf{v}_{i-1}(t)) + \mathbf{v}_i^{\text{fb}}(t). \tag{159}$$

When we remove the noise $\boldsymbol{\xi}_i$ and take $\lim_{\tau_{v^{\text{fb}}} \to 0}$, we recover the original network dynamics (1) of the main manuscript. If we assume that $\mathbf{u}$ is independent from $\mathbf{v}_i^{\text{fb}}$, eq. (158) is a linear time-invariant stochastic differential equation that can be solved with the Variation of Constants method, without a special treatment for the white noise [43], leading to the following solution for $\mathbf{v}_i^{\text{fb}}$:

$$\mathbf{v}_i^{\text{fb}}(t) = \frac{1}{\tau_{v^{\text{fb}}}} \int_{-\infty}^{t} \exp\left(-\frac{1}{\tau_{v^{\text{fb}}}}(t-\tau)\right) Q_i\mathbf{u}(\tau)\mathrm{d}\tau + \frac{\sigma}{\tau_{v^{\text{fb}}}} \int_{-\infty}^{t} \exp\left(-\frac{1}{\tau_{v^{\text{fb}}}}(t-\tau)\right)\boldsymbol{\xi}_i(\tau)\mathrm{d}\tau. \tag{160}$$

If we now assume that $\tau_{v^{\text{fb}}} \ll \tau_u$, and hence the dynamics of the feedback compartment is much faster than $\mathbf{u}$, $\mathbf{v}^{\text{fb}}$ can be approximated by

$$\mathbf{v}_i^{\text{fb}}(t) \approx Q_i \mathbf{u}(t) + \frac{\sigma}{\tau_{v^{\text{fb}}}} \int_{-\infty}^t \exp\left(-\frac{1}{\tau_{v^{\text{fb}}}}(t - \tau)\right) \boldsymbol{\xi}_i(\tau) \mathrm{d}\tau \tag{161}$$

$$\triangleq Q_i \mathbf{u}(t) + \sigma \boldsymbol{\epsilon}_i(t), \tag{162}$$

with $\boldsymbol{\epsilon}_i(t)$ exponentially filtered white noise, i.e., an Ornstein-Uhlenbeck process [43] with zero mean and covariance equal to [8]

$$\mathbb{E}\left[\boldsymbol{\epsilon}_i(t)\boldsymbol{\epsilon}_i(t + \Delta t)^T\right] = \frac{1}{2\tau_{v^{\text{fb}}}} \exp\left(-\frac{1}{\tau_{v^{\text{fb}}}}|\Delta t|\right). \tag{163}$$

In the remainder of the section, we assume this approximation to be exact. The network dynamics (159) can then be written as

$$\tau_v \frac{\mathrm{d}\mathbf{v}_i(t)}{\mathrm{d}t} = -\mathbf{v}_i(t) + W_i \phi(\mathbf{v}_{i-1}(t)) + Q_i \mathbf{u}(t) + \sigma \boldsymbol{\epsilon}_i(t). \tag{164}$$

Now, we are ready to state and prove the main theorem of this section, which shows that the feedback weight plasticity rule (13) pushes the feedback weights to align with a damped pseudoinverse of the forward Jacobian $J$ of the network.

**Theorem S11.** *Assuming stable dynamics, a separation of timescales $\tau_{v^{\text{fb}}}, \tau_v \ll \tau_u \ll \tau_Q$, $k_p = 0$, $\alpha \gg |\lambda_{max}(JQ_M)|$ and $J$ is of full rank, and given $\mathbf{r}_L^* = \mathbf{r}_L^-$ and uncorrelated white noise $\boldsymbol{\xi}_i \sim \mathcal{N}(\mathbf{0}, I)$ entering the feedback compartment, the dynamics of the first moment $Q_M$ of the apical weight plasticity (13) has the following approximate solution in the limit of $\sigma \to 0$:*

$$\lim_{\sigma \to 0} \frac{d}{dt} Q_M \approx -\frac{1}{\tau_u \alpha} Q_M J J^T + \frac{1}{2\tau_u} J^T - \beta Q_M, \tag{165}$$

*and the first moment converges to:*

$$Q_M^{\text{ss}} \approx \frac{\alpha}{2} J^T (JJ^T + \gamma I)^{-1}, \tag{166}$$

*with $\gamma = \alpha\beta\tau_u$. Furthermore, $Q_M^{\text{ss}}$ satisfies Conditions 2 and 3, even if $\alpha = 0$ in the latter.*

*Proof.* Linearizing the system dynamics (which becomes exact in the limit of $\sigma \to 0$ and assuming stable dynamics), results in the following dynamical equation for the controller, recalling that $\mathbf{r}_L^* = \mathbf{r}_L^-$ (c.f. App. A.1):

$$\tau_u \frac{\mathrm{d}}{\mathrm{d}t} \mathbf{u}(t) = -J\Delta\mathbf{v}(t) - \alpha\mathbf{u}(t), \tag{167}$$

with $\Delta\mathbf{v}_i \triangleq \mathbf{v}_i - W_i\phi(\mathbf{v}_{i-1})$ and $\Delta\mathbf{v}$ the concatenation of all $\Delta\mathbf{v}_i$. When we have a separation of timescales between the network and controller, i.e., $\tau_v \ll \tau_u$, which corresponds with *instant system dynamics* of the network (164), we get

$$\Delta\mathbf{v}_i = Q_i\mathbf{u} + \sigma\boldsymbol{\epsilon}_i \tag{168}$$

$$\Delta\mathbf{v} = Q\mathbf{u} + \sigma\boldsymbol{\epsilon} \tag{169}$$

where the latter is the concatenated version of the former. Combining this with eq. (167) gives the following stochastic differential equation for the controller dynamics:

$$\tau_u \frac{\mathrm{d}}{\mathrm{d}t} \mathbf{u}(t) = -(JQ + \alpha I)\mathbf{u}(t) - \sigma J\boldsymbol{\epsilon}(t). \tag{170}$$

When we have a separation of timescales between the synaptic plasticity and controller dynamics, i.e., $\tau_u \ll \tau_Q$, we can treat $Q$ as constant and therefore eq. (170) represents a linear time-invariant stochastic differential equation, which has as solution [43]

$$\mathbf{u}(t) = -\frac{\sigma}{\tau_u} \int_{-\infty}^t e^{-\frac{1}{\tau_u} A(t - \tau)} J\boldsymbol{\epsilon}(\tau) \mathrm{d}\tau \tag{171}$$

$$A \triangleq JQ + \alpha I. \tag{172}$$

---

[8]Note that in $\lim_{\tau_{v^{\text{fb}}} \to 0}$, $\boldsymbol{\epsilon}_i(t)$ has infinite variance, and is hence not well defined, explaining why we need to make this detour of assuming non-instantaneous dynamics for the feedback compartment.

Using the approximate solution of the feedback compartment (161) (which we consider exact due to the separation of timescales $\tau_{v^{\text{fb}}} \ll \tau_u$), we can write the expectation of the first part of the feedback learning rule (13) as

$$\mathbb{E}\big[-\mathbf{v}^{\text{fb}}(t)\mathbf{u}(t)^T\big] = \mathbb{E}\Big[-\big(Q\mathbf{u}(t)+\sigma\boldsymbol{\epsilon}(t)\big)\mathbf{u}(t)^T\Big] \tag{173}$$

$$= \mathbb{E}\Big[\underbrace{-\frac{\sigma^2}{\tau_u^2}Q\int_{-\infty}^t \exp\big(-\frac{1}{\tau_u}A(t-\tau)\big)J\boldsymbol{\epsilon}(\tau)\mathrm{d}\tau \int_{-\infty}^t \boldsymbol{\epsilon}(\tau)^T J^T \exp\big(-\frac{1}{\tau_u}A^T(t-\tau)\big)\mathrm{d}\tau}_{(a)}$$
$$\tag{174}$$

$$\underbrace{\ldots + \frac{\sigma^2}{\tau_u}\boldsymbol{\epsilon}(t)\int_{-\infty}^t \boldsymbol{\epsilon}(\tau)^T J^T \exp\big(-\frac{1}{\tau_u}A^T(t-\tau)\big)\mathrm{d}\tau}_{(b)}\Big]. \tag{175}$$

Focusing on (a) and using the covariance of $\boldsymbol{\epsilon}$ (163), we get:

$$\mathbb{E}[(a)] = -\frac{\sigma^2}{\tau_u^2}Q\int_{-\infty}^t e^{\frac{1}{\tau_u}A(t-\tau_1)}J\int_{-\infty}^t \frac{1}{2\tau_{v^{\text{fb}}}}e^{-\frac{1}{\tau_{v^{\text{fb}}}}|\tau_1-\tau_2|}J^T e^{-\frac{1}{\tau_u}A^T(t-\tau_2)}\mathrm{d}\tau_2\mathrm{d}\tau_1 \tag{176}$$

$$= -\frac{\sigma^2}{\tau_u^2}Q\int_{-\infty}^t e^{-\frac{1}{\tau_u}A(t-\tau_1)}J\Bigg(\int_{-\infty}^{\tau_1} \frac{1}{2\tau_{v^{\text{fb}}}}J^T e^{-\big(\frac{1}{\tau_{v^{\text{fb}}}}I+\frac{1}{\tau_u}A^T\big)(\tau_1-\tau_2)}e^{-\frac{1}{\tau_u}A^T(t-\tau_1)}\mathrm{d}\tau_2 \tag{177}$$

$$\ldots + \int_{\tau_1}^t \frac{1}{2\tau_{v^{\text{fb}}}}J^T e^{-\big(\frac{1}{\tau_{v^{\text{fb}}}}I-\frac{1}{\tau_u}A^T\big)(\tau_2-\tau_1)}e^{-\frac{1}{\tau_u}A^T(t-\tau_1)}\mathrm{d}\tau_2\Bigg)\mathrm{d}\tau_1 \tag{178}$$

$$\approx -\frac{\sigma^2}{\tau_u^2}Q\int_{-\infty}^t e^{-\frac{1}{\tau_u}A(t-\tau_1)}JJ^T e^{-\frac{1}{\tau_u}A^T(t-\tau_1)}\mathrm{d}\tau_1, \tag{179}$$

where we used in the last step that $\tau_{v^{\text{fb}}} \ll \tau_u$, hence $\frac{1}{\tau_{v^{\text{fb}}}}I - \frac{1}{\tau_u}A^T \approx \frac{1}{\tau_{v^{\text{fb}}}}I$ and $\frac{1}{\tau_{v^{\text{fb}}}}\int_{-t_1}^0 e^{-\frac{1}{\tau_{v^{\text{fb}}}}\tau}\mathrm{d}\tau \approx 1$ when $\tau_{v^{\text{fb}}} \ll t_1$ for $t_1 > 0$. If we further assume that $\alpha \gg \max\big(\{|\lambda_i(JQ)|\}\big)$ with $\lambda_i(JQ)$ the eigenvalues of $JQ$, we have that

$$e^A = e^{JQ+\alpha I} \approx e^\alpha, \tag{180}$$

and hence

$$\mathbb{E}[(a)] \approx -\frac{\sigma^2}{\tau_u^2}Q\int_{-\infty}^t e^{-\frac{2}{\tau_u}\alpha(t-\tau_1)}JJ^T\mathrm{d}\tau_1 \tag{181}$$

$$= -\frac{\sigma^2}{\tau_u\alpha}QJJ^T. \tag{182}$$

Focusing on part (b), we get

$$\mathbb{E}[(b)] = \frac{\sigma^2}{\tau_u}\int_{-\infty}^t \frac{1}{2\tau_{v^{\text{fb}}}}e^{-\frac{1}{\tau_{v^{\text{fb}}}}(t-\tau)}J^T \exp\big(-\frac{1}{\tau_u}A^T(t-\tau)\big)\mathrm{d}\tau \tag{183}$$

$$= \frac{\sigma^2}{\tau_u}J^T\int_{-\infty}^t \frac{1}{2\tau_{v^{\text{fb}}}}e^{-\big(\frac{1}{\tau_{v^{\text{fb}}}}I+\frac{1}{\tau_u}A^T\big)(t-\tau)}\mathrm{d}\tau \tag{184}$$

$$\approx \frac{\sigma^2}{2\tau_u}J^T. \tag{185}$$

Taking everything together, we get the following approximate dynamics for the first moment of $Q$:

$$\lim_{\sigma\to 0}\frac{\mathrm{d}}{\mathrm{d}t}\mathbb{E}[Q] = \lim_{\sigma\to 0}\frac{\mathrm{d}}{\mathrm{d}t}Q_M = \lim_{\sigma\to 0}\mathbb{E}\big[-\frac{1}{\sigma^2}\mathbf{v}^{\text{fb}}\mathbf{u}^T - \beta Q\big] \tag{186}$$

$$\approx -\frac{1}{\tau_u\alpha}Q_M JJ^T + \frac{1}{2\tau_u}J^T - \beta Q_M. \tag{187}$$

Assuming the approximation exact and solving for the steady state, we get:

$$0 = -\frac{1}{\tau_u \alpha} Q_M^{\text{ss}} J J^T + \frac{1}{2\tau_u} J^T - \beta Q_M^{\text{ss}} \tag{188}$$

$$\Rightarrow \quad Q_M^{\text{ss}} = \frac{\alpha}{2} J^T (J J^T + \alpha \beta \tau_u I)^{-1}. \tag{189}$$

The only thing remaining to show is that the dynamics of $Q_M$ are convergent. By vectorizing eq. (187), we get

$$\tau_Q \frac{\mathrm{d}}{\mathrm{d}t} \text{vec}(Q_M) = -\frac{1}{\tau_u \alpha} (J J^T \otimes I + \alpha \beta \tau_u I) \text{vec}(Q_M) + \frac{1}{2\tau_u} \text{vec}(J^T). \tag{190}$$

As the eigenvalues of a Kronecker product $A \otimes B$ are equal to the products $\lambda_i^A \lambda_j^B$, the eigenvalues of $J J^T \otimes I$ are equal to the eigenvalues of $J J^T$ (in higher multiplicity) and hence all positive. This makes the above dynamical system convergent, thereby concluding the main part of the proof. Finally, Lemma S12 shows that, if $J$ is full rank, $Q_M^{\text{ss}} = \frac{\alpha}{2} J^T (J J^T + \alpha \beta \tau_u I)^{-1}$ satisfies Conditions 2 and 3, even if $\alpha = 0$ in the latter.

$\square$

**Lemma S12.** $Q = J^T (J J^T + \gamma I)^{-1}$ with $\gamma \geq 0$ satisfies Condition 2 and the product $J J^T (J J^T + \gamma I)^{-1}$ with $\gamma \geq 0$ has strictly positive eigenvalues if $J$ is of full rank.

*Proof.* When $J$ is of full rank, $J^T (J J^T + \gamma I)^{-1}$ can be written as $J^T M$, with $M$ a square full rank matrix. As $M$ is full rank, $J^T M$ has the same column space as $J^T$, thereby proving that $Q = J^T (J J^T + \gamma I)^{-1}$ with $\gamma \geq 0$ satisfies Condition 2.

Next, consider the singular value decomposition of $J$:

$$J = U \Sigma V^T. \tag{191}$$

Now, $J J^T (J J^T + \gamma I)^{-1}$ can be written as

$$J J^T (J J^T + \gamma I)^{-1} = U \Sigma \Sigma^T (\Sigma \Sigma^T + \gamma I)^{-1} U^T, \tag{192}$$

with $\Sigma \Sigma^T (\Sigma \Sigma^T + \gamma I)^{-1}$ a diagonal matrix with $\frac{\sigma_i^2}{\sigma_i^2 + \gamma} > 0$ on its diagonal, and $\sigma_i$ the singular values of $J$. As $U^T = U^{-1}$, and $\Sigma \Sigma^T (\Sigma \Sigma^T + \gamma I)^{-1}$ is diagonal, eq. (192) is the eigenvalue decomposition of $J J^T (J J^T + \gamma I)^{-1}$, with eigenvalues $\frac{\sigma_i^2}{\sigma_i^2 + \gamma} > 0$, thereby concluding the proof. $\square$

### C.2  Toy experiments corroborating the theory

To test whether Theorem S11 can also provide insight into more realistic settings, we conducted a series of student-teacher toy regression experiments with a one-hidden-layer network of size $20 - 10 - 5$ for more realistic values of $\tau_{v^{\text{fb}}}$, $\tau_v$, $\alpha$ and $k_p > 0$. For details about the simulation implementation, see App. E. We investigate the learning of $Q$ during pre-training, hence, when the forward weights $W_i$ are fixed. In contrast to Theorem S11, we use multiple batch samples for training the feedback weights. When the network is linear, $J$ remains the same for each batch sample, hence mimicking the situation of Theorem S11 where $Q$ is trained on only one sample to convergence. When the network is nonlinear, however, $J$ will be different for each sample, causing $Q$ to align with an average configuration over the batch samples.

We start by investigating which damping value $\gamma$ accurately describes the alignment of $Q$ with $J^T (J J^T + \gamma I)^{-1}$ in this more realistic case. Fig. S3.a shows the alignment of $Q$ with $J^T (J J^T + \gamma I)^{-1}$ for different damping values $\gamma$ in a linear network. Interestingly, the damping value that optimally describes the alignment of $Q$ is $\gamma = 5$, which is much larger than would be predicted by Theorem S11 which uses simplified conditions. Hence, the more realistic settings used in the simulation of these toy experiments result in a larger damping value $\gamma$. For nonlinear networks, similar conclusions can be drawn (see Fig. S3.b), however, with slightly worse alignment due to $J$ changing for each batch sample. Note that almost perfect compliance to Condition 2 is reached for both the linear and nonlinear case (not shown here).

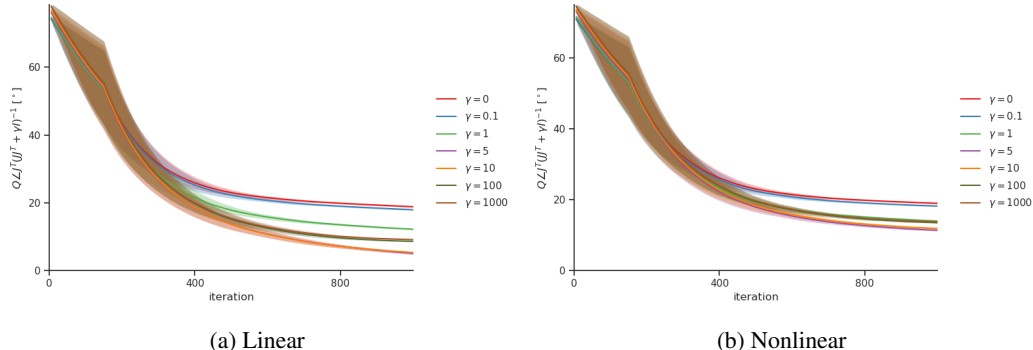

| | |
|---|---|
| (a) Linear | (b) Nonlinear |

Figure S3: Alignment of the feedback weights $Q$ with the damped pseudoinverse $J^T(JJ^T + \gamma I)^{-1}$ for various values of $\gamma$. We used a one-hidden-layer network of size 20-10-5 with a linear output layer and (a) a linear hidden layer or (b) a $\texttt{tanh}$ hidden layer. Hyperparameters: $k_p = 0$, $\alpha = 0.5$, $\tau_u = 1.$, $\tau_{v^{\text{fb}}} = 0.3$, $\tau_v = 0.005$, $\beta = 0.01$ and $\sigma = 0.01$. We used 300 Euler-Maruyama simulation steps of size $\Delta t = 0.001$. A window-average is plotted together with the window-std (shade).

Next, we investigate how big $\alpha$ needs to be for good alignment. Surprisingly, Fig. S4 shows that $Q$ reaches almost perfect alignment for all values of $\alpha \in [0, 1]$, both for linear and nonlinear networks. We hypothesize that this is due to the short simulation window (300 steps of $\Delta t = 0.001$) that we used to reduce computational costs, preventing the dynamics from diverging, even when they are unstable. Interestingly, this hypothesis leads to another case where the feedback learning rule (13) can be used besides for big $\alpha$: when the network activations can be 'reset' when they start diverging, e.g., by inhibition from other brain areas, the feedback weights can be learned properly, even with unstable dynamics.

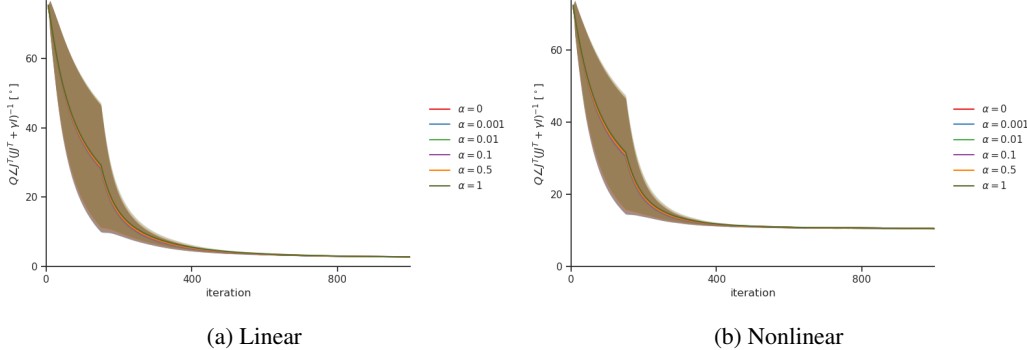

| | |
|---|---|
| (a) Linear | (b) Nonlinear |

Figure S4: Alignment of the feedback weights $Q$ with the damped pseudoinverse $J^T(JJ^T + \gamma I)^{-1}$ for various values of $\alpha$. We used a one-hidden-layer network of size 20-10-5 with a linear output layer and (a) a linear hidden layer or (b) a $\texttt{tanh}$ hidden layer. Hyperparameters: $k_p = 0.3$, $\gamma = 5$, $\tau_u = 1.$, $\tau_{v^{\text{fb}}} = 0.3$, $\tau_v = 0.005$, $\beta = 0.01$ and $\sigma = 0.01$. We used 300 Euler-Maruyama simulation steps of size $\Delta t = 0.001$. A window-average is plotted together with the window-std (shade).

Finally, we investigate how $k_p$ influences the feedback learning. Fig. S5 shows that bigger $k_p$ increase the speed of alignment with $J^T(JJ^T + \gamma I)^{-1}$.

## C.3 Learning the forward and feedback weights simultaneously

In this section, we show that the forward and feedback weights can be learned simultaneously, when noise is added to the feedback compartment, resulting in the noisy dynamics of eq. (164), and when the feedback plasticity rule (13) uses a high-pass filtered version of $\mathbf{u}$ as presynaptic plasticity signal.

We make the same assumptions as in Theorem S11, except now the output target $\mathbf{r}_L^*$ is the one for learning the forward weights, hence given by eq. (3). Linearizing the network dynamics, gives us the

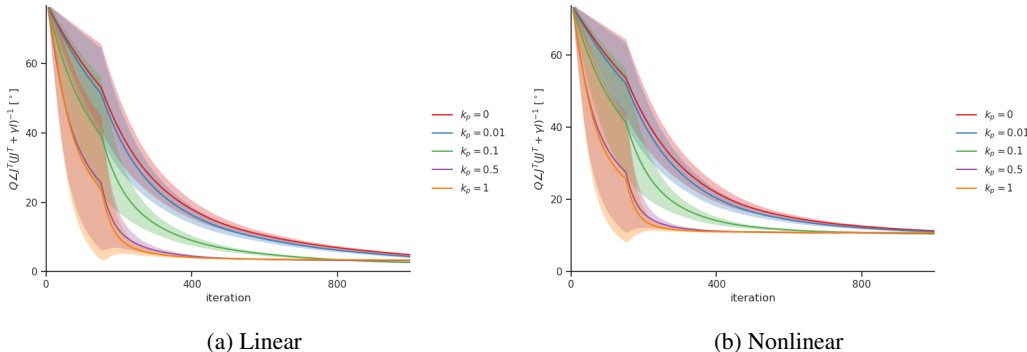

(a) Linear            (b) Nonlinear

Figure S5: Alignment of the feedback weights $Q$ with the damped pseudoinverse $J^T(JJ^T + \gamma I)^{-1}$ for various values of $k_p$. We used a one-hidden-layer network of size 20-10-5 with a linear output layer and (a) a linear hidden layer or (b) a tanh hidden layer. Hyperparameters: $\alpha = 0.1$, $\gamma = 5$, $\tau_u = 1.$, $\tau_{v^{\text{fb}}} = 0.3$, $\tau_v = 0.005$, $\beta = 0.01$ and $\sigma = 0.01$. We used 300 Euler-Maruyama simulation steps of size $\Delta t = 0.001$. A window-average is plotted together with the window-std (shade).

following expression for the control error

$$\mathbf{e}(t) = \boldsymbol{\delta}_L - J\Delta\mathbf{v}(t), \tag{193}$$

and for the controller dynamics (with $k_p = 0$)

$$\tau_u \frac{\mathrm{d}}{\mathrm{d}t}\mathbf{u}(t) = \boldsymbol{\delta}_L - J\Delta\mathbf{v}(t) - \alpha\mathbf{u}(t). \tag{194}$$

Using instantaneous network dynamics ($\tau_v \ll \tau_u$), we have that $\Delta\mathbf{v}(t) = Q\mathbf{u}(t) + \sigma\boldsymbol{\epsilon}(t)$, giving us:

$$\tau_u \frac{\mathrm{d}}{\mathrm{d}t}\mathbf{u}(t) = \boldsymbol{\delta}_L - (JQ + \alpha I)\mathbf{u}(t) - \sigma J\boldsymbol{\epsilon}(t). \tag{195}$$

We now continue by investigating the dynamics of newly defined signal $\Delta\mathbf{u}(t)$ that subtracts a baseline from the control signal $\mathbf{u}(t)$:

$$\Delta\mathbf{u}(t) \triangleq \mathbf{u}(t) - \mathbf{u}_{\text{ss}}, \quad \mathbf{u}_{\text{ss}} = (JQ + \alpha I)^{-1}\boldsymbol{\delta}_L, \tag{196}$$

with $\mathbf{u}_{\text{ss}}$ being the steady state of $\mathbf{u}$ in the dynamics without noise (see Lemma 1). Rewriting the dynamics (195) for $\Delta\mathbf{u}$ gives us

$$\tau_u \frac{\mathrm{d}}{\mathrm{d}t}\Delta\mathbf{u}(t) = -(JQ + \alpha I)\Delta\mathbf{u}(t) - \sigma J\boldsymbol{\epsilon}(t). \tag{197}$$

We now recovered exactly the same dynamics for $\Delta\mathbf{u}$ as was the case for $\mathbf{u}$ (170) during the sleep phase where $\mathbf{r}_L^* = \mathbf{r}_L^-$ in Theorem S11. Now, we introduce a new plasticity rule for $Q$ using $\Delta\mathbf{u}$ instead of $\mathbf{u}$ as presynaptic plasticity signal:

$$\tau_Q \frac{\mathrm{d}}{\mathrm{d}t}Q(t) = -\mathbf{v}^{\text{fb}}(t)\Delta\mathbf{u}(t)^T - \beta Q(t). \tag{198}$$

Upon noting that $\Delta\mathbf{u}$ (representing the noise fluctuations in $\mathbf{u}$) is independent of $\mathbf{u}_{\text{ss}}$ (representing the control input needed to drive the network to $\mathbf{r}_L^*$), the approximate first moment dynamics described in Theorem S11 also hold for the new plasticity rule (198). Furthermore, when the controller dynamics (195) have settled, $\mathbf{u}_{\text{ss}}$ is the average of $\mathbf{u}(t)$ (which has zero-mean noise fluctuations on top of $\mathbf{u}_{\text{ss}}$), hence, $\Delta\mathbf{u}$ can be seen as a high-pass filtered version of $\mathbf{u}(t)$.

To conclude, we have shown that the sleep phase for training the feedback weights $Q$ can be merged with the phase for training the forward weights with $\mathbf{r}_L^*$ as defined in eq. (3), if the plasticity rule for $Q$ (198) uses a high-pass filtered version $\Delta\mathbf{u}$ of $\mathbf{u}$ as presynaptic plasticity signal and when the network and controller are fluctuating around their equilibrium, as we did not take initial conditions into account. We hypothesize that even with initial dynamics that have not yet converged to the steady-state, the plasticity rule for Q (198) with $\Delta\mathbf{u}$ a high-pass filtered version of $\mathbf{u}$ will result in

proper feedback learning, as high-pass filtering $\mathbf{u}(t)$ will extract high-frequency noise fluctuations[9] out of it which are correlated with $\mathbf{v}^{\text{fb}}$ and can hence be used for learning $Q$. We leave it to future work to experimentally verify this hypothesis. Merging the two phases into one has as a consequence that there is also noise present during the learning of the forward weights (5), which we investigate in the next subsection.

## C.4 Influence of noisy dynamics on learning the forward weights

When there is noise present in the dynamics during learning the forward weights, this will have an influence on the updates of $W_i$. It turns out that the same noise correlations that we used in the previous sections to learn the feedback weights will cause bias terms to appear in the updates of the forward weights $W_i$ (5). This issue is not unique to our DFC setting with a feedback controller but appears in general in methods that use error feedback and have realistic noise dynamics in their hidden layers. In this section, we lay down the issues caused by noise dynamics for learning forward weights for general methods that use error feedback. At the end of the section, we comment on the implications of these issues for DFC.

For simplicity, we consider a normal feedforward neural network

$$\mathbf{r}_i^- = \phi(\mathbf{v}_i^-) = \phi(W_i\mathbf{r}_{i-1}^-), \quad 1 \le i \le L. \tag{199}$$

To incorporate the notion of noisy dynamics, we perturb each layer's pre-nonlinearity activation with zero-mean noise $\boldsymbol{\epsilon}_i$ and propagate the perturbations forward:

$$\tilde{\mathbf{v}}_i^- = W_i\phi(\tilde{\mathbf{v}}_{i-1}^-) + \sigma\boldsymbol{\epsilon}_i, \quad 1 \le i \le L, \tag{200}$$

with $\tilde{\mathbf{r}}_0^- = \mathbf{r}_0^-$. For small $\sigma$, a first-order Taylor approximation of the perturbed output gives

$$\tilde{\mathbf{r}}_L^- = \mathbf{r}_L^- + \sigma J\boldsymbol{\epsilon} + \mathcal{O}(\sigma^2), \tag{201}$$

with $\boldsymbol{\epsilon}$ the concatenated vector of all $\boldsymbol{\epsilon}_i$. If the task loss is an $L^2$ loss and we have the training label $\mathbf{r}_L^*$, the output error is equal to

$$\mathbf{e}_L = \mathbf{r}_L^* - \tilde{\mathbf{r}}_L^- = \boldsymbol{\delta}_L - \sigma J\boldsymbol{\epsilon} + \mathcal{O}(\sigma^2), \tag{202}$$

with $\boldsymbol{\delta}_L = \mathbf{r}_L^* - \mathbf{r}_L^-$, the output error without noise perturbations. To remain general, we define the feedback path $\mathbf{e}_i = g_i(\mathbf{e}_L)$ that transports the output error $\mathbf{e}_L$ to the hidden layer $i$, at the level of the pre-nonlinearity activations. E.g., for BP, $\mathbf{e}_i = g_i(\mathbf{e}_L) = J_i^T\mathbf{e}_L$, and for direct linear feedback mappings such as DFA, $\mathbf{e}_i = g_i(\mathbf{e}_L) = Q_i\mathbf{e}_L$. Now, the commonly used update rule of postsynaptic error signal multiplied with presynaptic input gives (after a first-order Taylor expansion of all terms)

$$\Delta W_i = \eta\mathbf{e}_i\tilde{\mathbf{r}}_{i-1}^{-T} \tag{203}$$

$$= \eta\big(\boldsymbol{\delta}_i - \sigma J_{g_i}J\boldsymbol{\epsilon} + \mathcal{O}(\sigma^2)\big)\big(\mathbf{r}_{i-1}^- + \sigma D_{i-1}\boldsymbol{\epsilon}_{i-1} + \mathcal{O}(\sigma^2)\big)^T, \tag{204}$$

with $\boldsymbol{\delta}_i = g_i(\boldsymbol{\delta}_L)$, $J_{g_i} = \frac{\partial g_i(\mathbf{e}_L)}{\partial \mathbf{e}_L}\big|_{\mathbf{e}_L=\boldsymbol{\delta}_L}$ and $D_i = \frac{\partial \mathbf{r}_i^-}{\partial \mathbf{v}_i}\big|_{\mathbf{v}_i=\mathbf{v}_i^-}$. Taking the expectation of $\Delta W_i$, we get

$$\mathbb{E}[\Delta W_i] = \eta\boldsymbol{\delta}_i\mathbf{r}_{i-1}^{-T} - \eta\sigma^2 J_{g_i}J_{i-1}\Sigma_{i-1}D_{i-1} + \mathcal{O}(\sigma^3), \tag{205}$$

with $\Sigma_{i-1}$ the covariance matrix of $\boldsymbol{\epsilon}_{i-1}$. We see that besides the desired update $\eta\boldsymbol{\delta}_i\mathbf{r}_{i-1}^{-T}$, there also appears a bias term due to the noise, which scales with $\sigma^2$ and cannot be avoided by averaging over weight updates. The noise bias arises from the correlation between the noise in the presynaptic input $\tilde{\mathbf{r}}_{i-1}$ and the postsynaptic error $\mathbf{e}_i$. Note that it is not a valid strategy to assume that the noise in $\mathbf{e}_i$ is uncorrelated from the noise in $\tilde{\mathbf{r}}_{i-1}$ due to a time delay between the two signals, as in more realistic cases, $\boldsymbol{\epsilon}$ originates from stochastic dynamics that integrate noise over time (e.g., one can think of $\boldsymbol{\epsilon}$ as an Ornstein-Uhlenbeck process [43]) and is hence always correlated over time.

In DFC, similar noise biases arise in the average updates of $W_i$. To reduce the relative impact of the noise bias on the weight update, the ratio $\|\boldsymbol{\delta}_i\|_2/\sigma^2$ must be big enough, hence strong error feedback

---

[9]Not all noise fluctuations are high-frequency. However, the important part of the hypothesis is that the high-pass filtering selects noise components that are zero-mean and correlate with $\mathbf{v}^{\text{fb}}$.

is needed. In DFC, $\|\boldsymbol{\delta}_L\|_2$, and hence also the postsynaptic error term in the weight updates for $W_i$, scales with the target stepsize $\lambda$. Interestingly, this causes a trade-off to appear in DFC: on the one hand, $\lambda$ needs to be small such that the weight updates (5) approximate GN and MN optimization (the theorems used Taylor approximations which become exact for $\lambda \to 0$), and on the other hand, $\lambda$ needs to be big to prevent the forward weight updates from being buried in the noise bias.

A possible solution for removing the noise bias from the average forward weight updates is to either buffer the postsynaptic error term or the presynaptic input $\mathbf{r}_{i-1}$, or both (e.g., accumulating them or low-pass filtering them), before they are multiplied with each other to produce the weight update. This procedure would average the noise out in the signals, before they have the chance to correlate with each other in the weight update. Whether this procedure could correspond with biophysical mechanisms in a neuron is an interesting question for future work.

# D  Related work

Our learning theory analysis that connects DFC to Gauss-Newton (GN) optimization was inspired by three independent recent studies that, on the one hand, connect Target Propagation (TP) to GN optimization [21, 22] and, on the other hand, point to a possible connection between Dynamic Inversion (DI) and GN optimization [32]. There are however important distinctions between how DFC approximates GN and how TP and DI approximate GN. In the following subsections, we discuss these related lines of work in detail.

## D.1  Comparison of DFC to TP and variants

Recent work [21, 22] discovered that learning through inverses of the forward pathway can in certain cases lead to an approximation of GN optimization. Although this finding inspired our theoretical results on the CA capabilities of DFC, there are fundamental differences between DFC and TP. The main conceptual difference between DFC and the variants of TP [19, 20, 21, 22] is that DFC uses the combination of network dynamics and a controller to dynamically invert the forward pathway for CA, whereas TP and its variants learn parametric inverses of the forward pathway, encoded in the feedback weights. Although dynamic and parametric inversion seem closely related, they lead to major methodological and theoretical differences.

**Methodological differences between DFC and TP.**  First, for TP and its variants, the task of approximating the inverse of the forward pathway is completely put onto the feedback weights, resulting in the need for a strict relation between the feedforward and feedback pathway at all times during training. DFC, in contrast, reuses the forward pathway to dynamically compute its inverse, resulting in a more flexible relation between the feedforward and feedback pathway, described by Condition 2. To the best of our knowledge, DFC is the first method that approximates a principled optimization method for feedforward neural networks of arbitrary dimensions, compatible with a wide range of feedback connectivity. The recent work of Bengio [22] iteratively improves the inverse and, hence, can compensate for imperfect parametric inverses. However, this method is developed only for invertible networks, which require all layers to have equal dimensions.

Second, DFC drives the hidden neural activations to target values simultaneously, hence letting 'target activations' from upstream layers influence 'target activations' from downstream layers. TP, in contrast, computes each target as a (pseudo)inverse of the output target independently. This is a subtle yet important difference between DFC and TP, which leads to significant theoretical differences, on which we will expand later. To gain intuition, consider the case where we update the weights of both DFC and TP to reach exactly the local layer targets. In TP, if we update the weights of a hidden layer to reach its target, all downstream layers will also reach their target without updating the weights. Hence, if we update all weights simultaneously, the output will overshoot its target. DFC, in contrast, takes the effect of the updated target values of upstream layers already into account, hence, when all weight updates are done simultaneously, the output target is reached exactly (in the linearized dynamics, c.f. Theorem 3).

Third, DFC needs significantly less external coordination compared to the recent TP variants. The new variants of TP with a link to GN [21] need highly coordinated noise phases for computing the Difference Reconstruction Loss (one separated noise phase for each layer). For DTP [20], similar coordination is needed if noisy activations are used for computing the reconstruction loss, as proposed

by the authors. The iterative variant of TP [22] needs coordination in propagating the target values, as the target iterations for a layer can only start when the iterations of the downstream layer have converged. As DFC uses dynamic inversion instead of parametric inversion, possible learning rules for the feedback weights do not need to use the Difference Reconstruction Loss [21] or variants thereof, opening the route to alternative, more biologically realistic learning rules. We propose a first feedback learning rule compatible with DFC, that makes use of noise and Hebbian learning, without the need for extensive external coordination (see also App. C.3 that merges feedforward and feedback weight training in a single-phase).

Finally, DFC uses a multi-compartment neuron model closely corresponding to recent models of cortical pyramidal neurons, to obtain plasticity rules fully local in space and time. Presently, it is unclear whether there exist similar neuron and network models for TP that result in plasticity rules local in time.

**Theoretical differences between DFC and TP.** First, computing layerwise inverses, as is done in TP [19], DTP [20], and iterative TP [22], can only be linked to GN for invertible networks but breaks down for non-invertible networks, as shown by Meulemans et al. [21]. Both DFC and the DRL variants of TP [21] establish a link to GN for both invertible and non-invertible feedforward networks of arbitrary dimensions. However, the DRL variants of TP are linked to a hybrid version of GN and gradient descent, whereas DFC, under appropriate conditions, is linked to pure GN optimization on the parameters. Our Theorems 2 and 3 differ from the theoretical results on the DRL variants of TP [21] due to the fact that: (i) the DRL variants compute targets for the post-nonlinearity activations and the DFC target activations, $\mathbf{v}_i$, are pre-nonlinearity activations; and (ii) the DRL variants compute the targets for each layer independently, whereas DFC dynamically computes the targets while taking into account the changed target activations of other layers. We continue with expanding on this second point.

As explained intuitively before, TP and its variants compute each layer target independently from the other layer targets. Consequently, to link their variants of TP to GN optimization, Meulemans et al. [21] and Bengio [22] need to make a block-diagonal approximation of the GN curvature matrix, with each block corresponding to a single layer. As off-diagonal blocks are put to zero, influences of upstream target values on the downstream targets are ignored. The block-diagonal approximation of the GN curvature matrix was proposed in studies that used GN optimization to train deep neural networks with big minibatch sizes [64, 65]. However, similar to DFC, TP is connected to GN with a minibatch size of 1. In this case, the GN curvature matrix is of low rank, and a block-diagonal approximation of this matrix will change its rank and hence its properties. In the analysis of DFC, in contrast, we do not need to make this block-diagonal approximation, as the target activations, $\mathbf{v}_i$, influence each other. Consequently, DFC has a closer connection to GN optimization than the TP variants [21, 22].

Finally, DFC does not use a reconstruction loss to train the feedback weights but instead uses noise and Hebbian learning.

**Empirical comparison of DFC to TP and variants** Table S1 shows the results for DTP [20], and DDTP-linear [21] (the best performing variant of TP in [21]) on MNIST, Fashion MNIST, MNIST-autoencoder, and MNIST (train), for the same architectures as used for Table 1.

Table S1: The test error (MNIST, Fashion MNIST), test loss (MNIST-autoencoder), and training loss MNIST (train) for DTP and DDTP-linear. Same network architectures and settings as for Table 1.

|  | MNIST | Fashion-MNIST | MNIST-autoencoder | MNIST (train) |
|---|---|---|---|---|
| DTP | $2.61^{\pm 0.13}\%$ | $11.26^{\pm 0.23}\%$ | $22.36^{\pm 0.59} \cdot 10^{-2}$ | $8.36^{\pm 4.09} \cdot 10^{-6}$ |
| DDTP-linear | $2.22^{\pm 0.22}\%$ | $10.84^{\pm 0.22}\%$ | $14.60^{\pm 0.10} \cdot 10^{-2}$ | $1.97^{\pm 0.70} \cdot 10^{-8}$ |

Comparing these results to the ones in Table 1, we see that DFC outperforms DTP on all datasets and DDTP-linear on MNIST-autoencoder, while having similar performance on the other datasets. These encouraging results suggest that the closer connection of DFC to GN, when compared to the one of DDTP-linear to GN (see section D.1), leads to practical improvements in performance in some more challenging datasets.

### D.2 Comparison of DFC to Dynamic Inversion

Recent work introduced DI [32], which, similar to DFC, dynamically inverts the forward pathway through the use of a controller. However, some fundamental differences between DI and DFC lead to various new desirable properties of DFC. First, DFC introduces a principled way to control all layers simultaneously, hence requiring less external control. Second, in contrast to DI, the learning rules in DFC are fully local in time. Third, DFC can train the feedback weights to continuously adapt themselves to the changing forward pathway, leading to more accurate CA. Finally, Podlaski and Machens [32] only explored the link between DI and GN for linear one-hidden layer networks and it requires a block-diagonal approximation of the curvature matrix, similar to TP. Upon closer inspection, the link between DI and GN cannot be generalized to networks with multiple hidden layers of various sizes or nonlinear activation functions, in contrast to DFC. This is because the layerwise dynamical inversion in DI does not result in the pseudoinverses of $J_i = \frac{\partial \mathbf{r}_L}{\partial \mathbf{r}_i}$ since: (i) the pseudoinverse cannot be factorized over the layers [21]; and (ii) in nonlinear networks, the Jacobians are evaluated at a wrong value because DI transmits errors instead of controlled layer activations through the forward path of the network during the dynamical inversion phase.

### D.3 The core contributions of DFC

In summary, we see that DFC merges various insights from different fields resulting in a novel biologically plausible CA technique with unique and interesting properties that transcend the sheer sum of its parts. To clarify the novelty of our work, we summarize here again the core contributions of DFC:

- DFC extends the idea of using a feedback controller to adjust network activations to also provide CA to DNNs by using it to track the desired output target, opening a new route for designing principled CA methods for DNNs.

- To the best of our knowledge, DFC is the first method that approximates a principled optimization method for feedforward neural networks of arbitrary dimensions, while allowing for a wide and flexible range of feedback connectivity, in contrast to a single allowed feedback configuration.

- The learning rules of DFC for the forward and feedback weights are fully local both in time and space, in contrast to many other biologically plausible learning rules. Furthermore, DFC does not need highly specific connectivity motives nor tightly coordinated plasticity mechanisms and can have all weights plastic simultaneously, if the adaptations explained in appendix C.3 are used.

- The multi-compartment neuron model needed for DFC naturally corresponds to recent multi-compartment models of pyramidal neurons.

## E Simulations and algorithms of DFC

In this section, we provide details on the simulation and algorithms used for DFC, DFC-SS, DFC-SSA and for training the feedback weights.

### E.1 Simulating DFC and DFC-SS for training the forward weights

For simulating the network dynamics (1) and controller dynamics (4) without noise, we used the forward Euler method with some slight modifications. First, we implemented the controller dynamics (4) as follows:

$$\mathbf{u}(t) = \mathbf{u}^{\text{int}}(t) + k_p \mathbf{e}(t), \quad \tau_u \frac{\mathrm{d}}{\mathrm{d}t} \mathbf{u}^{\text{int}}(t) = \mathbf{e}(t) - \tilde{\alpha} \mathbf{u}(t). \tag{206}$$

Note that we changed the leakage term from $\alpha \mathbf{u}^{\text{int}}$ to $\tilde{\alpha} \mathbf{u}$, such that we have direct control over the hyperparameter $\tilde{\alpha}$ (the damping factor in Lemma 1) that is now independent of $k_p$. Note that both (4) and (206) result in exactly the same dynamics for $\mathbf{u}(t)$, if $\tilde{\alpha} = \frac{\alpha}{1+k_p\alpha}$ and $\tau_u$ scaled by $\frac{\alpha}{1+k_p\alpha}$. Hence, this is just an implementation strategy to gain direct control over $\tilde{\alpha}$ as a hyperparameter independent from $k_p$.

Algorithm 1 provides the pseudo-code of our simulation of the network and controller dynamics during the training of the forward weights $W_i$ and biases $\mathbf{b}_i$. We use the forward Euler method [43] with stepsize $\Delta t$ and make two nuanced modifications. First, to better reflect the layered structure of the network, we use $\mathbf{v}_i^{\text{ff}}[k+1] = W_i\phi(\mathbf{v}_{i-1}[k+1]) + \mathbf{b}_i$ instead of $\mathbf{v}_i^{\text{ff}}[k+1] = W_i\phi(\mathbf{v}_{i-1}[k]) + \mathbf{b}_i$.[10] For small stepsizes $\Delta t$, this modification has almost no effect. However, for larger stepsizes, the modification better reflects the underlying continuous dynamics with its layerwise structure. Second, using insights from discrete control theory, we use $\mathbf{v}_i^{\text{fb}}[k+1] = Q_i\mathbf{u}[k+1]$ instead of $\mathbf{v}_i^{\text{fb}}[k+1] = Q_i\mathbf{u}[k]$, such that the control error $\mathbf{e}[k]$ of the previous timestep is used to provide feedback, instead of the control error $\mathbf{e}[k-1]$ of two timesteps ago.[11] Again, this modification has almost no effect for small stepsizes $\Delta t$, but better reflects the underlying continuous dynamics for bigger stepsizes. In our simulations, the stepsize $\Delta t$ that worked best for the experiments was small, hence, the discussed modifications had only minor effects on the simulation.

---

**Algorithm 1:** Simulation of DFC for training the forward parameters.

---

Initialize layer activations and parameter update buffers:
**for** *i in range(1,L)* **do**
  $\mathbf{v}_i[1] = \mathbf{v}_i^-$
  $\mathbf{r}_i[1] = \mathbf{r}_i^-$
  $\mathbf{u}^{\text{int}}[1] = 0$
  $\Delta W_i = 0$
  $\Delta \mathbf{b}_i = 0$

**for** *k in range(1,$K_{\max}$)* **do**
  Update controller:
  $\mathbf{e}[k] = \mathbf{r}_L^* - \mathbf{r}_L[k]$
  $\mathbf{u}^{\text{int}}[k+1] = \mathbf{u}^{\text{int}}[k] + \frac{\Delta t}{\tau_u}(\mathbf{e}[k] - \tilde{\alpha}\mathbf{u}[k])$
  $\mathbf{u}[k+1] = \mathbf{u}^{\text{int}}[k+1] + k_p\mathbf{e}[k]$
  Update network:
  **for** *i in range(1,L)* **do**
    $\mathbf{v}_i^{\text{ff}}[k+1] = W_i\phi(\mathbf{v}_{i-1}[k+1]) + \mathbf{b}_i$
    $\mathbf{v}_i^{\text{fb}}[k+1] = Q_i\mathbf{u}[k+1]$
    $\mathbf{v}_i[k+1] = \mathbf{v}_i[k] + \frac{\Delta t}{\tau_v}(-\mathbf{v}_i[k] + \mathbf{v}_i^{\text{ff}}[k+1] + \mathbf{v}_i^{\text{fb}}[k+1])$
    $\mathbf{r}_i[k+1] = \phi(\mathbf{v}_i[k+1])$
    Buffer forward parameter updates:
    $\Delta W_i = \Delta W_i + \big(\phi(\mathbf{v}_i[k+1]) - \phi(\mathbf{v}_i^{\text{ff}}[k+1])\big)\mathbf{r}_{i-1}[k+1]^T$
    $\Delta \mathbf{b}_i = \Delta \mathbf{b}_i + \phi(\mathbf{v}_i[k+1]) - \phi(\mathbf{v}_i^{\text{ff}}[k+1])$

Update forward parameters with $\Delta W_i/K_{\max}$ and $\Delta \mathbf{b}_i/K_{\max}$ and an optimizer of choice

---

For DFC-SS, the same simulation strategy is used, with as only difference that the weight updates $\Delta W_i$ only use the network activations of the last simulation step (see Algorithm 2). Finally, for DFC-SSA, we directly compute the steady-state solutions according to Lemma 1 (see Algorithm 3).

### E.2 Simulating DFC with noisy dynamics for training the feedback weights

For simulating the noisy dynamics during the training of the feedback weights, we use the Euler-Maruyama method [43], which is the stochastic version of the forward Euler method. As discussed in App. C, we let white noise $\boldsymbol{\xi}$ enter the dynamics of the feedback compartment and we now take a finite time constant $\tau_{v^{\text{fb}}}$ for the feedback compartment, as the instantaneous form with $\tau_{v^{\text{fb}}} \to 0$ (that we used for simulating the network dynamics without noise) is not well defined when noise enters the dynamics:

$$\tau_{v^{\text{fb}}}\frac{\mathrm{d}}{\mathrm{d}t}\mathbf{v}_i^{\text{fb}}(t) = -\mathbf{v}_i^{\text{fb}}(t) + Q_i\mathbf{u}(t) + \sigma\boldsymbol{\xi}_i. \tag{207}$$

---

[10]In the code repository, this modification to Euler's method is indicated with the command line argument `inst_transmission`

[11]In the code repository, this modification to Euler's method is indicated with the command line argument `proactive_controller`

---

**Algorithm 2:** Simulation of DFC-SS for training the forward parameters.

---

Initialize layer activations and parameter update buffers:

**for** *i in range(1,L)* **do**

$\quad$ $\mathbf{v}_i[1] = \mathbf{v}_i^-$

$\quad$ $\mathbf{r}_i[1] = \mathbf{r}_i^-$

$\quad$ $\mathbf{u}^{\text{int}}[1] = 0$

**for** *k in range(1,$K_{\max}$)* **do**

$\quad$ Update controller:

$\quad$ $\mathbf{e}[k] = \mathbf{r}_L^* - \mathbf{r}_L[k]$

$\quad$ $\mathbf{u}^{\text{int}}[k+1] = \mathbf{u}^{\text{int}}[k] + \frac{\Delta t}{\tau_u}(\mathbf{e}[k] - \tilde{\alpha}\mathbf{u}[k])$

$\quad$ $\mathbf{u}[k+1] = \mathbf{u}^{\text{int}}[k+1] + k_p\mathbf{e}[k]$

$\quad$ Update network:

$\quad$ **for** *i in range(1,L)* **do**

$\quad\quad$ $\mathbf{v}_i^{\text{ff}}[k+1] = W_i\phi(\mathbf{v}_{i-1}[k+1]) + \mathbf{b}_i$

$\quad\quad$ $\mathbf{v}_i^{\text{fb}}[k+1] = Q_i\mathbf{u}[k+1]$

$\quad\quad$ $\mathbf{v}_i[k+1] = \mathbf{v}_i[k] + \frac{\Delta t}{\tau_v}(-\mathbf{v}_i[k] + \mathbf{v}_i^{\text{ff}}[k+1] + \mathbf{v}_i^{\text{fb}}[k+1])$

$\quad\quad$ $\mathbf{r}_i[k+1] = \phi(\mathbf{v}_i[k+1])$

Compute forward parameter updates using the last simulation step:

$\Delta W_i = \big(\phi(\mathbf{v}_i[K_{\max}]) - \phi(\mathbf{v}_i^{\text{ff}}[K_{\max}])\big)\mathbf{r}_{i-1}[K_{\max}]^T$

$\Delta \mathbf{b}_i = \phi(\mathbf{v}_i[K_{\max}]) - \phi(\mathbf{v}_i^{\text{ff}}[K_{\max}])$

Update forward parameters with $\Delta W_i$ and $\Delta \mathbf{b}_i$ and an optimizer of choice

---

---

**Algorithm 3:** DFC-SSA iteration for training the forward parameters.

---

Compute the network Jacobian $J$

$\boldsymbol{\delta}_L = \mathbf{r}_L^* - \mathbf{r}_L^-$

Compute steady-state solution using Lemma 1:

$\mathbf{u}_{\text{ss}} = \big(JQ + \tilde{\alpha}I\big)^{-1}\boldsymbol{\delta}_L$

$\Delta \mathbf{v}_{\text{ss}} = Q\mathbf{u}_{\text{ss}}$

Split $\Delta \mathbf{v}_{\text{ss}}$ over the layers into $\Delta \mathbf{v}_{i,\text{ss}}$

Compute steady-state network activations:

$\mathbf{r}_{0,\text{ss}} = \mathbf{r}_0^-$

**for** *i in range(1,L)* **do**

$\quad$ $\mathbf{v}_{i,\text{ss}} = W_i\mathbf{r}_{i-1,\text{ss}} + \mathbf{b}_i + \Delta \mathbf{v}_{i,\text{ss}}$

$\quad$ $\mathbf{r}_{i,\text{ss}} = \phi(\mathbf{v}_{i,\text{ss}})$

Compute forward parameter updates using the analytical steady-state solutions:

$\Delta W_i = \big(\phi(\mathbf{v}_{i,\text{ss}}) - \phi(\mathbf{v}_{i,\text{ss}}^{\text{ff}})\big)\mathbf{r}_{i-1,\text{ss}}^T$

$\Delta \mathbf{b}_i = \phi(\mathbf{v}_{i,\text{ss}}) - \phi(\mathbf{v}_{i,\text{ss}}^{\text{ff}})$

Update forward parameters with $\Delta W_i$ and $\Delta \mathbf{b}_i$ and an optimizer of choice

---

The dynamics for the network then becomes

$$\tau_v \frac{\mathrm{d}}{\mathrm{d}t}\mathbf{v}_i(t) = -\mathbf{v}_i(t) + W_i\mathbf{r}_{i-1}(t) + \mathbf{v}_i^{\text{fb}}(t), \tag{208}$$

and, as before, eq. (206) is taken for the controller dynamics. Using the Euler-Maruyama method [43], the feedback compartment dynamics (207) can be simulated as

$$\mathbf{v}_i^{\text{fb}}[k+1] = \mathbf{v}_i^{\text{fb}}[k] + \frac{\Delta t}{\tau_{v^{\text{fb}}}}\big(-\mathbf{v}_i^{\text{fb}}[k] + Q_i\mathbf{u}[k+1]\big) + \frac{\sqrt{\Delta t}}{\tau_{v^{\text{fb}}}}\sigma\Delta\boldsymbol{\xi}_i, \quad \Delta\boldsymbol{\xi}_i \sim \mathcal{N}(0, I). \tag{209}$$

As all other dynamical equations do not have noise, their simulation remains equivalent to the simulation with the forward Euler method. Algorithm 4 provides the pseudo code of the simulation of DFC during the feedback weight training phase.

**Algorithm 4:** Simulation of DFC for training the feedback weights.

---

Initialize layer activations and parameter update buffers:

**for** *i in range(1,L)* **do**
$\quad$ $\mathbf{v}_i[1] = \mathbf{v}_i^-$
$\quad$ $\mathbf{r}_i[1] = \mathbf{r}_i^-$
$\quad$ $\mathbf{u}^{\text{int}}[1] = 0$
$\quad$ $\Delta Q_i = 0$

**for** *k in range(1,$K_{\max}$)* **do**
$\quad$ Update controller:
$\quad$ $\mathbf{e}[k] = \mathbf{r}_L^- - \mathbf{r}_L[k]$
$\quad$ $\mathbf{u}^{\text{int}}[k+1] = \mathbf{u}^{\text{int}}[k] + \frac{\Delta t}{\tau_u}(\mathbf{e}[k] - \tilde{\alpha}\mathbf{u}[k])$
$\quad$ $\mathbf{u}[k+1] = \mathbf{u}^{\text{int}}[k+1] + k_p\mathbf{e}[k]$
$\quad$ Update network:
$\quad$ **for** *i in range(1,L)* **do**
$\quad\quad$ $\mathbf{v}_i^{\text{ff}}[k+1] = W_i\phi(\mathbf{v}_{i-1}[k+1]) + \mathbf{b}_i$
$\quad\quad$ Sample noise and let it enter in the feedback compartment with non-instantaneous
$\quad\quad$ dynamics:
$\quad\quad$ $\Delta\boldsymbol{\xi}_i \sim \mathcal{N}(0, I)$
$\quad\quad$ $\mathbf{v}_i^{\text{fb}}[k+1] = \mathbf{v}_i^{\text{fb}}[k] + \frac{\Delta t}{\tau_{v\text{fb}}}\big(-\mathbf{v}_i^{\text{fb}}[k] + Q_i\mathbf{u}[k+1]\big) + \frac{\sqrt{\Delta t}}{\tau_{v\text{fb}}}\sigma\Delta\boldsymbol{\xi}_i$
$\quad\quad$ $\mathbf{v}_i[k+1] = \mathbf{v}_i[k] + \frac{\Delta t}{\tau_v}(-\mathbf{v}_i[k] + \mathbf{v}_i^{\text{ff}}[k+1] + \mathbf{v}_i^{\text{fb}}[k+1])$
$\quad\quad$ $\mathbf{r}_i[k+1] = \phi(\mathbf{v}_i[k+1])$
$\quad\quad$ Buffer feedback weight updates:
$\quad\quad$ $\Delta Q_i = \Delta Q_i - \mathbf{v}_i^{\text{fb}}[k]\mathbf{u}[k+1] - \beta Q_i$

Update feedback parameters with $\Delta Q_i/K_{\max}$ and an optimizer of choice

---

# F   Experiments

## F.1   Description of the alignment measures

In this section, we describe the alignment measures used in Fig. 3 in detail.

**Condition 2.** Fig. 3A describes how well the network satisfies Condition 2. For this, we project $Q$ onto the column space of $J^T$, for which we use a projection matrix $P_{J^T}$:

$$P_{J^T}Q = J^T(JJ^T)^{-1}JQ. \tag{210}$$

Then, we compare the Frobenius norm of the projection of $Q$ with the norm of $Q$, via its ratio:

$$\text{ratio}_{\text{Con2}} = \frac{\|P_{J^T}Q\|_F}{\|Q\|_F}. \tag{211}$$

Notice that a $\text{ratio}_{\text{Con2}} = 1$ indicates that the column space of $Q$ lies fully inside the column space of $J^T$, hence indicating that Condition 2 is satisfied.[12] At the opposite extreme, $\text{ratio}_{\text{Con2}} = 0$ indicates that the column space of Q is orthogonal on the column space of $J^T$.

**Condition 1.** Fig. 3C describes how well the network satisfies Condition 1. This condition states that all layers (except the output layer) have an equal $L^2$ norm. To measure how well Condition 1 is satisfied, we compute the standard deviation of the layer norms over the layers, and normalize it by

---

[12]Note that in degenerate cases, $Q$ could be lower rank and still have $\text{ratio}_{\text{Con2}} = 1$ if its (reduced) column space lies inside the column space of $J^T$. As $Q$ is a skinny matrix, we assume it is always of full rank and do not consider this degenerate scenario.

the average layer norm:

$$\text{ratio}_{\text{Con1}} = \frac{\frac{1}{L}\sum_{i=0}^{L}\left(\|\mathbf{r}_i\|_2 - \text{mean}(\|\mathbf{r}\|_2)\right)^2}{\text{mean}(\|\mathbf{r}\|_2)} \tag{212}$$

$$\text{mean}(\|\mathbf{r}\|_2) = \frac{1}{L}\sum_{i=0}^{L}\|\mathbf{r}_i\|_2 \tag{213}$$

We take $\mathbf{r}_i = \mathbf{r}_i^-$ to compute this measure, but other values of $\mathbf{r}_i$ during the dynamics would also work, as they remain close together for a small target stepsize $\lambda$. Now, notice that $\text{ratio}_{\text{Con1}} = 0$ indicates perfect compliance with Condition 1, as then all layers have the same norm, and $\text{ratio}_{\text{Con1}} = 1$ indicates that the layer norms vary by $\text{mean}(\|\mathbf{r}\|_2)$ on average, hence indicating that Condition 1 is not at all satisfied.

**Stability measure.** Fig. 3E describes the stability of DFC during training. For this, we plot the maximum real part of the eigenvalues of the total system matrix $A_{PI}$ around the steady state (see eq. (136)), which describes the dynamics of DFC around the steady state (incorporating $k_p$ and the actual time constants, in contrast to Condition 3).

**Alignment with MN updates.** Fig. 3B describes the alignment of the DFC updates with the ideal weighted MN updates. The MN updates are computed as follows:

$$\Delta\bar{W}^{\text{MN}} = RJ^\dagger\boldsymbol{\delta}_L, \tag{214}$$

with $R$ defined in eq. (70) and $\bar{W}$ the concatenated vectorized form of all weights $W_i$. For the alignment measurements in the computer vision experiments (see Section F.5.3) we use a damped variant of the MN updates:

$$\Delta\bar{W}^{\text{MN}} = RJ^T(JJ^T + \gamma I)^{-1}\boldsymbol{\delta}_L, \tag{215}$$

with $\gamma$ some positive damping constant. The damping constant is needed to incorporate the damping effect of the leakage constant, $\alpha$, into the dynamical inversion, but also to reflect an *implicit damping* effect. Meulemans et al. [21] showed that introducing a higher damping constant, $\gamma$, in the pseudoinverse (214) reflected better the updates made by TP, which uses learned inverses. We found empirically that a higher damping constant, $\gamma$, also reflects better the updates made by DFC. Using a similar argumentation, we hypothesize that this implicit damping in DFC originates from the fact that, in nonlinear networks, $J$ changes for each batch sample and hence $Q$ cannot satisfy Condition 2 for each batch sample. Consequently, $Q$ tries to satisfy Condition 2 as good as possible for all batch samples, but does not satisfy it perfectly, resulting in a phenomenon that can be partially described by implicit damping.

**Alignment with GN updates.** Fig. 3D describes the alignment of the DFC updates with the ideal GN updates. The GN updates are computed as follows:

$$\Delta\bar{W}^{\text{GN}} = J_{\bar{W}}^\dagger\boldsymbol{\delta}_L, \tag{216}$$

with $J_{\bar{W}} = \frac{\partial\mathbf{r}_L^-}{\partial\bar{W}}$, evaluated at the feedforward activations $\mathbf{r}_i^-$. Similarly to the MN updates, we also introduce a damped variant of the GN updates, which is used in the computer vision alignment experiments (Section F.5.3):

$$\Delta\bar{W}^{\text{GN}} = J_{\bar{W}}^T(J_{\bar{W}}J_{\bar{W}}^T + \gamma I)^{-1}\boldsymbol{\delta}_L, \tag{217}$$

where the damping constants, $\gamma$ and $\alpha$, reflect the leakage constant and the implicit damping effects, respectively.

**Alignment with DFC-SSA updates.** Finally, Fig. 3F describes the alignment of the DFC updates with the DFC-SSA updates which use the linearized analytical steady-state solution of the dynamics. The DFC-SSA updates are computed as follows (see also Algorithm 3):

$$\Delta\bar{W}^{\text{SSA}} = R_{\text{ss}}Q(JQ + \tilde{\alpha}I)^{-1}\boldsymbol{\delta}_L, \tag{218}$$

with $R_{\text{ss}}$ defined in eq. (70) but with the steady-state values $\mathbf{r}_{i,\text{ss}}$ instead of $\mathbf{r}_i^-$.

### F.2 Description of training

**Training phases.** We iterate between one epoch of training the forward weights and $X$ epochs of training the feedback weights, with $X \in [1, 2, 3]$ a hyperparameter. The extra epochs for training the feedback weights enable the feedback weights to better satisfy Conditions 2 and 3 when the forward weights are changing fast (e.g., during early training), and slightly improve the performance of DFC. Before the training starts, we pre-train the feedback weights for 10 epochs, starting from a random configuration, to ensure that the network is stable when the training begins and Condition 2 is approximately satisfied.

**Student-teacher toy regression.** For the toy experiments of Fig. 3, we use the student-teacher regression paradigm. Here, a randomly initialized teacher generates a synthetic regression dataset using random inputs. A separate randomly initialized student is then trained on this synthetic dataset. We used more hidden layers and neurons for the teacher network compared to the student network, such that the student network cannot get 'lucky' by being initialized close to the teacher network.

**Optimizer.** In student-teacher toy regression experiments, we use vanilla SGD without momentum as an optimizer. In the computer vision experiments, we use a separate Adam optimizer [44] for the forward and feedback weights, as this improves training results compared to vanilla SGD. As Adam was designed for BP updates, it will likely not be an optimal optimizer for DFC, which uses MN updates. An interesting future research direction is to design new optimizers that are tailored towards the MN updates of DFC, to further improve its performance. We used gradient clipping for all DFC experiments to prevent too large updates when the inverse of $J$ is poorly conditioned.

**Training length and reported test results.** For the classification experiments, we used 100 epochs of training for the forward weights (and a corresponding amount of feedback training epochs, depending on $X$). As the autoencoder experiment was more resource-intensive, we trained the models for only 25 epochs there, as this was sufficient for getting near-perfect autoencoding performance when visually inspected (see Fig. S14). For all experiments, we split the 60000 training samples into a validation set of 5000 samples and a training set of 55000 samples. The hyperparameter searches are done based on the validation accuracy (validation loss for MNIST-autoencoder and train loss for MNIST-train) and we report the test results corresponding to the epoch with best validation results in Table 1.

**Weight initializations.** All network weights are initialized with the Glorot-Bengio normal initialization [66], except when stated otherwise.

**Initialization of the fixed feedback weights.** For the variants of DFC with fixed feedback weights, we use the following initialization:

$$Q_i = \prod_{k=i+1}^{L} W_k^T, \quad 1 \leq i \leq L - 1 \tag{219}$$

$$Q_L = I \tag{220}$$

For $\tanh$ networks, this initialization approximately satisfies Conditions 2 and 3 at the beginning of training. This is because $Q$ will approximate $J^T$, as the forward weights are initialized by Glorot-Bengio normal initialization [66], and the network will consequently be in the approximate linear regime of the $\tanh$ nonlinearities.

**Freeze $Q_L$.** For the MNIST-autoencoder experiments, we fixed the output feedback weights to $Q_L = I$, i.e., one-to-one connections between $\mathbf{r}_L$ and $\mathbf{u}$. As we did not train $Q_L$, we also did not introduce noise in the output layer during the training of the feedback weights. Freezing $Q_L$ prevents the noise in the high-dimensional output layer from burying the noise information originating from the small bottleneck layer and hence enabling better feedback weight training. This measure modestly improved the performance of DFC on MNIST-autoencoder (without fixing $Q_L$, the performance of all DFC variants was around 0.13 test loss – c.f. Table 1 – which is not a big decrease in performance). Freezing $Q_L$ does not give us any advantages over BP or DFA, as these methods implicitly assume to have direct access to the output error, i.e., also having fixed feedback connections between the error

neurons and output neurons equal to the identity matrix. We provided the option to freeze $Q_L$ into the hyperparameter searches of all experiments but this is not necessary for optimal performance of DFC in general, as this option was not always selected by the hyperparameter searches.

**Double precision.** We noticed that the standard data type `float32` of PyTorch [67] caused numerical errors to appear during the last epochs of training when the output error $\boldsymbol{\delta}_L$ is very small. For small $\boldsymbol{\delta}_L$, the difference $\phi(\mathbf{v}_i) - \phi(\mathbf{v}_i^{\text{ff}})$ in the forward weight updates (5) is very small and can result in numerical underflow. We solved this numerical problem by using `float64` (double precision) as data type.

### F.3 Architecture details

We use fully connected (FC) architectures for all experiments.

- Classification experiments (MNIST, Fashion-MNIST, MNIST-train): 3 FC hidden layers of 256 neurons with `tanh` nonlinearity and 1 softmax output layer of 10 neurons.
- MNIST-autoencoder: 256-32-256 FC hidden layers with tanh-linear-tanh nonlinearities and a linear output layer of 784 neurons.
- Student-teacher regression (Fig. 3): 2 FC hidden layers of 10 neurons and tanh nonlinearities, a linear output layer of 5 neurons, and input dimension 15.

**Absorbing softmax into the cross-entropy loss.** For the classification experiments (MNIST, Fashion-MNIST, and MNIST-train), we used a softmax output nonlinearity in combination with the cross-entropy loss. As the softmax nonlinearity and cross-entropy loss cancel out each others curvatures originating from the exponential and log terms, respectively, it is best to combine them into one output loss:

$$\mathcal{L}^{\text{combined}} = -\sum_{b=1}^{B} \mathbf{y}^{(b)T} \log \left( \text{softmax}(\mathbf{r}_L^{(b)}) \right), \tag{221}$$

with $\mathbf{y}^{(b)}$ the one-hot vector representing the class label of sample $b$, and $\log$ the element-wise logarithm. Now, as the softmax is absorbed into the loss function, the network output $\mathbf{r}_L$ can be taken linear and the output target is computed with eq. (3) using $\mathcal{L}^{\text{combined}}$.

### F.4 Hyperparameter searches

All hyperparameter searches were based on the best validation accuracy (best validation loss for MNIST-autoencoder and last train loss for MNIST-train) over all training epochs, using 5000 validation datasamples extracted from the training set. We use the Tree of Parzen Estimators hyperparameter optimization algorithm [68] based on the Hyperopt [69] and Ray Tune [70] Python libraries.

Due to the heavy computational cost of simulating DFC, we performed only hyperparameter searches for DFC-SSA, DFC-SSA (fixed), BP and DFA (200 hyperparameter samples for all methods). We used the hyperparameters found for DFC-SSA and DFC-SSA (fixed) for DFC and DFC-SS, and DFC (fixed) and DFC-SS (fixed), respectively, together with standard simulation hyperparameters for the forward weight training that proved to work well ($k_p = 2$, $\tau_u = 1$, $\tau_v = 0.2$, forward Euler stepsize $\Delta t = 0.02$ and 1000 simulation steps).

Tables S2 and S3 provide the hyperparameters and search intervals that we used for DFC-SSA in all experiments. We included the simulation hyperparameters for the feedback training phase in the search to prevent us from fine-tuning the simulations by hand. Note that we use different simulation hyperparameters for the forward training phase (see paragraph above) and the feedback training phase (see Table S3). This is because the simulation of the feedback training phase needs a small stepsize, $\Delta t_{\text{fb}}$, and a small network time constant, $\tau_v$, to properly simulate the stochastic dynamics. For the forward phase, however, we need to simulate over a much longer time interval, so taking small $\Delta t$ and $\tau_v$[13] would be too resource-intensive. When using $k_p = 2$, $\tau_u = 1$, and $\tau_v = 0.2$ during the simulation of the forward training phase, much bigger timesteps such as $\Delta t = 0.02$ can

---

[13]The simulation stepsize, $\Delta t$, needs to be smaller than the time constants.

be used. Note that these simulation parameters do not change the steady state of the controller and network, as $\tilde{\alpha}$ is independent from $k_p$ in our implementation. We also differentiated $\tilde{\alpha}$ in the forward training phase from $\tilde{\alpha}_{\text{fb}}$ in the feedback training phase, as the theory predicted that a bigger leakage constant is needed during the feedback training phase in the first epochs. However, toy simulations in Section C suggest that the feedback learning also works for smaller $\tilde{\alpha}$, which we did not explore in the computer vision experiments. Finally, we used $\text{lr} \cdot \lambda$ and $\lambda$ as hyperparameters in the search instead of $\text{lr}$ and $\lambda$ separately, as $\text{lr}$ and $\lambda$ have a similar influence on the magnitude of the forward parameter updates. The specific hyperparameter configurations for all experiments can be found in our codebase.[14]

Table S2: Hyperparameter symbols and meaning.

| Symbol | Hyperparameter |
|---|---|
| $\text{lr}$ | Learning rate of the Adam optimizer for the forward parameters |
| $\epsilon$ | Parameter of the Adam optimizer for the forward parameters |
| $\tilde{\alpha}$ | Leakage term of the controller dynamics (206) during training of the forward weights |
| $\lambda$ | Output target stepsize (see (3)) |
| $\text{lr}_{\text{fb}}$ | Learning rate of the Adam optimizer for the feedback parameters |
| $\text{lr}_{\text{fb, pre-train}}$ | Learning rate of the Adam optimizer for the feedback parameters during pre-training |
| $\epsilon_{\text{fb}}$ | Parameter of the Adam optimizer for the feedback parameters |
| $\tilde{\alpha}_{\text{fb}}$ | Leakage term of the controller dynamics (206) during the training of the feedback weights |
| $\beta$ | Weight decay term for the feedback weights |
| $k_{p,\text{fb}}$ | Proportional control constant during training of the feedback weights |
| $\tau_v$ | Time constant of the network dynamics, during training of the feedback weights |
| $\tau_{v^{\text{fb}}}$ | Time constant of the feedback compartment during feedback weight training |
| $\sigma$ | Standard deviation of the noise perturbation during training of the feedback weights |
| $X$ | Number of feedback training epochs after each forward training epoch |
| $\Delta t_{\text{fb}}$ | Stepsize for simulating the dynamics during feedback weight training |
| $t_{\text{max,fb}}$ | Number of simulation steps during feedback weight training |
| freeze_$Q_L$ | Flag for fixing the feedback weights $Q_L$ to the identity matrix |

Table S3: Hyperparameter search intervals for DFC.

| Hyperparameter | Search interval |
|---|---|
| $\text{lr} \cdot \lambda$ | $[10^{-7} : 10^{-4}]$ |
| $\epsilon$ | $[10^{-8} : 10^{-5}]$ |
| $\tilde{\alpha}$ | $[10^{-5} : 10^{-1}]$ |
| $\lambda$ | $[10^{-3} : 10^{-1}]$ |
| $\text{lr}_{\text{fb}}$ | $[1 \cdot 10^{-6} : 5 \cdot 10^{-4}]$ |
| $\text{lr}_{\text{fb, pre-train}}$ | $[5 \cdot 10^{-5} : 1 \cdot 10^{-3}]$ |
| $\epsilon_{\text{fb}}$ | $[10^{-8} : 10^{-5}]$ |
| $\tilde{\alpha}_{\text{fb}}$ | $[0.2 : 1.]$ |
| $\beta$ | $\{0; 10^{-5}; 10^{-3}; 10^{-1}\}$ |
| $k_{p,\text{fb}}$ | $[0 : 0.2]$ |
| $\tau_v$ | $[5 \cdot 10^{-3} : 1.5 \cdot 10^{-2}]$ |
| $\tau_{v^{\text{fb}}}$ | $[0.1 : 0.5]$ |
| $\sigma$ | $[10^{-3} : 10^{-1}]$ |
| $X$ | $\{1; 2; 3\}$ |
| $\Delta t_{\text{fb}}$ | $[1 \cdot 10^{-3} : 5 \cdot 10^{-3}]$ |
| $t_{\text{max,fb}}$ | $[30 : 60]$ |
| freeze_$Q_L$ | $\{\text{True}; \text{False}\}$ |

Table S4: Hyperparameter search intervals for BP and DFA.

| Hyperparameter | Search interval |
|---|---|
| $\text{lr}$ | $[10^{-8} : 10^{-2}]$ |
| $\epsilon$ | $[10^{-8} : 10^{-5}]$ |

[14]PyTorch implementation of all methods is available at `https://github.com/meulemansalex/deep_feedback_control`.

## F.5 Extended experimental results

In this section, we provide extra experimental results accompanying the results of Section 6.

### F.5.1 Training losses of the computer vision experiments

Table S5 provides the best training loss over all epochs for all the considered computer vision experiments. Comparing the train losses with the test performances in Table 1, shows that good test performance is not only caused by good optimization properties (i.e., low train loss) but also by other mechanisms, such as implicit regularization. The distinction is most pronounced in the results for MNIST. These results highlight the need to disentangle optimization from implicit regularization mechanisms to study the learning properties of DFC, which we do in the MNIST-train experiments provided in Table 1.

Table S5: Best training loss over a training of 100 epochs (classification) or 25 epochs (autoencoder). We use the Adam optimizer [44]. Architectures: 3x256 fully connected (FC) tanh hidden layers and softmax output (classification), 256-32-256 FC hidden layers for autoencoder MNIST with tanh-linear-tanh nonlinearities, and a linear output. Mean $\pm$ std (5 random seeds).

|  | MNIST | Fashion-MNIST | MNIST-autoencoder |
|---|---|---|---|
| BP | $7.51^{\pm 1.06} \cdot 10^{-4}$ | $1.19^{\pm 0.064} \cdot 10^{-2}$ | $9.57^{\pm 0.080} \cdot 10^{-2}$ |
| DFC | $1.93^{\pm 3.83} \cdot 10^{-4}$ | $8.84^{\pm 0.87} \cdot 10^{-2}$ | $1.16^{\pm 0.023} \cdot 10^{-1}$ |
| DFC-SSA | $2.51^{\pm 1.76} \cdot 10^{-5}$ | $1.74^{\pm 0.037} \cdot 10^{-1}$ | $1.16^{\pm 0.009} \cdot 10^{-1}$ |
| DFC-SS | $7.71^{\pm 5.04} \cdot 10^{-6}$ | $1.72^{\pm 0.022} \cdot 10^{-1}$ | $1.14^{\pm 0.006} \cdot 10^{-1}$ |
| DFC (fixed) | $9.04^{\pm 3.87} \cdot 10^{-4}$ | $2.34^{\pm 0.46} \cdot 10^{-1}$ | $3.34^{\pm 0.06} \cdot 10^{-1}$ |
| DFC-SSA (fixed) | $1.32^{\pm 0.22} \cdot 10^{-3}$ | $1.69^{\pm 0.24} \cdot 10^{-1}$ | $3.20^{\pm 0.041} \cdot 10^{-1}$ |
| DFC-SS (fixed) | $1.66^{\pm 0.44} \cdot 10^{-3}$ | $1.30^{\pm 0.11} \cdot 10^{-1}$ | $3.23^{\pm 0.04} \cdot 10^{-1}$ |
| DFA | $3.59^{\pm 0.14} \cdot 10^{-4}$ | $6.43^{\pm 0.37} \cdot 10^{-2}$ | $3.05^{\pm 0.021} \cdot 10^{-1}$ |

### F.5.2 Alignment plots for the toy experiment

Here, we show the alignment of the methods used in the toy experiments of Fig. 3 with MN updates and compare it with the alignment with BP updates. We plot the alignment angles per layer to investigate whether the alignment differs between layers. Fig. S6 shows the alignment of all methods with the damped MN angles and Fig. S7 with the BP angles. We see clearly that the alignment with MN angles is much better for the DFC variants with trained feedback weights compared to the alignment with BP angles, hence indicating that DFC uses a fundamentally different approach to learning, compared to BP, and thereby confirming the theory.

### F.5.3 Alignment plots for computer vision experiments

Figures S8 and S9 show the alignment of all methods with MN and BP updates, respectively. In contrast to the toy experiments in the previous section, now the alignment with BP is much closer to the alignment with MN updates. There are two main reasons for this. First, the classification networks we used have big hidden layers and a small output layer. In this case, the network Jacobian $J$ has many rows and only very few columns, which causes $J^{\dagger}$ to approximately align with $J^{T}$ (see among others Theorem S12 in Meulemans et al. [21]). Hence, the BP updates will also approximately align with the MN updates, explaining the better alignment with BP updates on MNIST compared to the toy experiments. Secondly, due to the nonlinearity of the network, $J$ changes for each datasample and $Q$ cannot satisfy Condition 2 exactly for all datasamples. We try to model this effect by introducing a higher damping constant, $\gamma = 1$, for computing the ideal damped MN updates (see Section F.1). However, this higher damping constant is not a perfect model for the phenomena occurring. Consequently, the alignment of DFC with the damped MN updates is suboptimal and a better alignment could be obtained by introducing other variants of MN updates that more accurately describe the behavior of DFC on nonlinear networks.[15] Note that nonetheless, the alignment with MN updates is better compared to the alignment with BP updates.

---

[15]Now, we perform a small grid-search to find a $\gamma \in \{0, 10^{-5}, 10^{-4}, 10^{-3}, 10^{-2}, 10^{-1}, 1, 10\}$ that best aligns with the DFC and DFA updates after 3 epochs of training. As this is a very coarse-grained approach,

Surprisingly, for Fashion-MNIST and MNIST-autoencoder, the DFC updates in the last and penultimate layer align better with BP than with MN updates (see Figures S11-S12). One notable difference between the configurations used for MNIST on the one hand and Fashion-MNIST and MNIST-autoencoder on the other hand, is that the hyperparameter search selected for the latter two to fix the output feedback weights $Q_L$ to the identity matrix (see Section F.2 for a description and discussion). This freezing of the output feedback weights slightly improved the performance of the DFC methods. Freezing $Q_L$ to the identity matrix explains why the output weight updates align closely with BP, as the postsynaptic plasticity signal is now an integrated plus proportional version of the output error. However, it is surprising that the alignment in the penultimate layer is also changed significantly. We hypothesize that this is due to the fact that the feedback learning rule (13) was designed for learning all feedback weights (leading to Theorem 6) and that freezing $Q_L$ breaks this assumption. However, extra investigation is needed to fully understand the occurring phenomena.

---

better alignment angles with damped MN updates could be obtained by a more fine-tuned approach for finding an optimal $\gamma$.

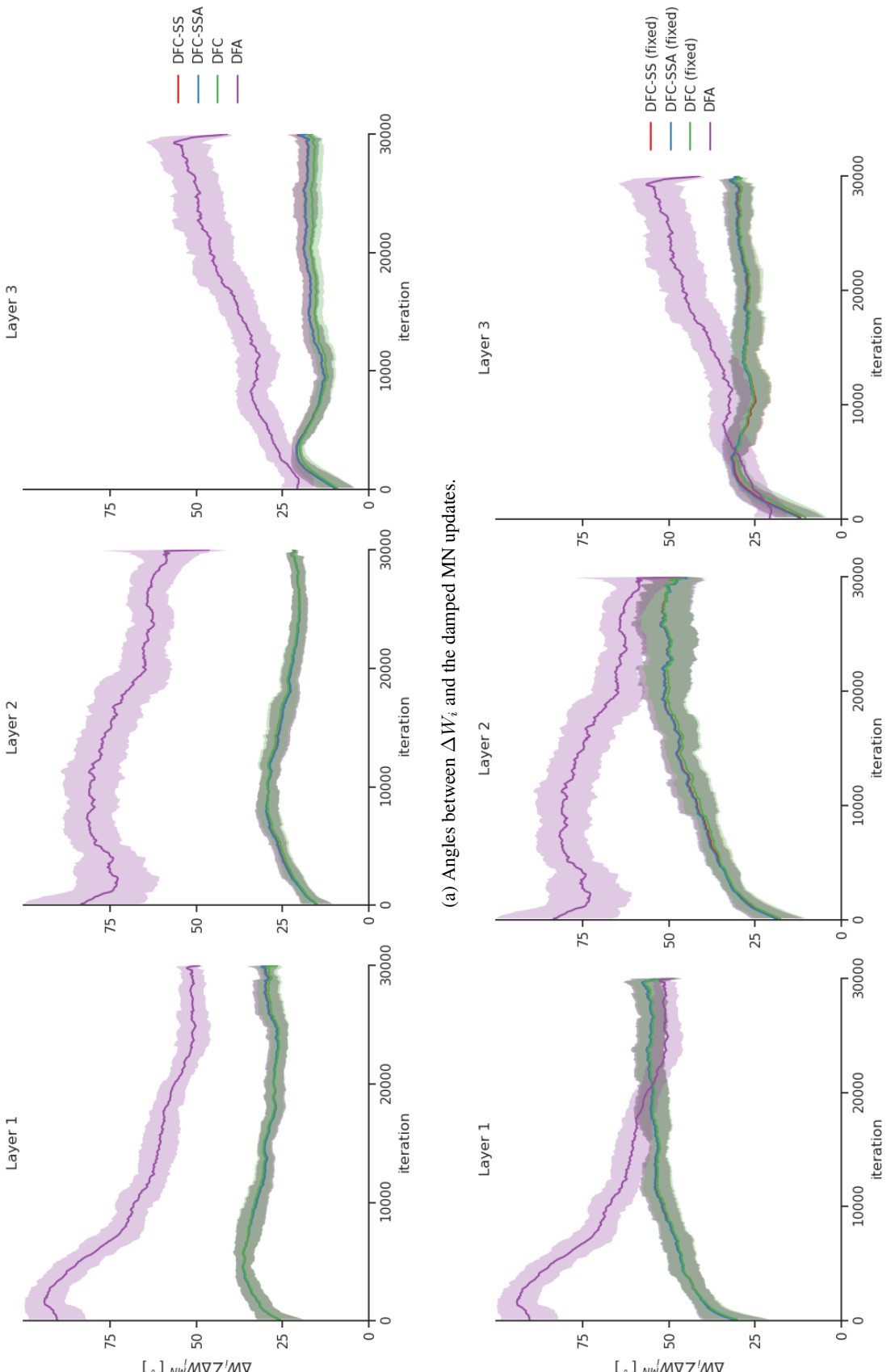

(a) Angles between $\Delta W_i$ and the damped MN updates.

(b) Angles between $\Delta W_i$ with fixed feedback weights and the damped MN updates.

Figure S6: Angles between the damped MN updates with $\gamma = 0.1$ (214) and the ones computed by DFC-SS, DFC-SSA, DFC, and DFA, plotted for all hidden layers with (a) learned feedback weights and (b) fixed feedback weights on the toy experiment explained in Figure 3. A window-average is plotted together with the window-std (shade). The x-axis iterations corresponds to the minibatches processed. The curves of DFC, DFC-SS, and DFC-SSA overlap in some of the plots.

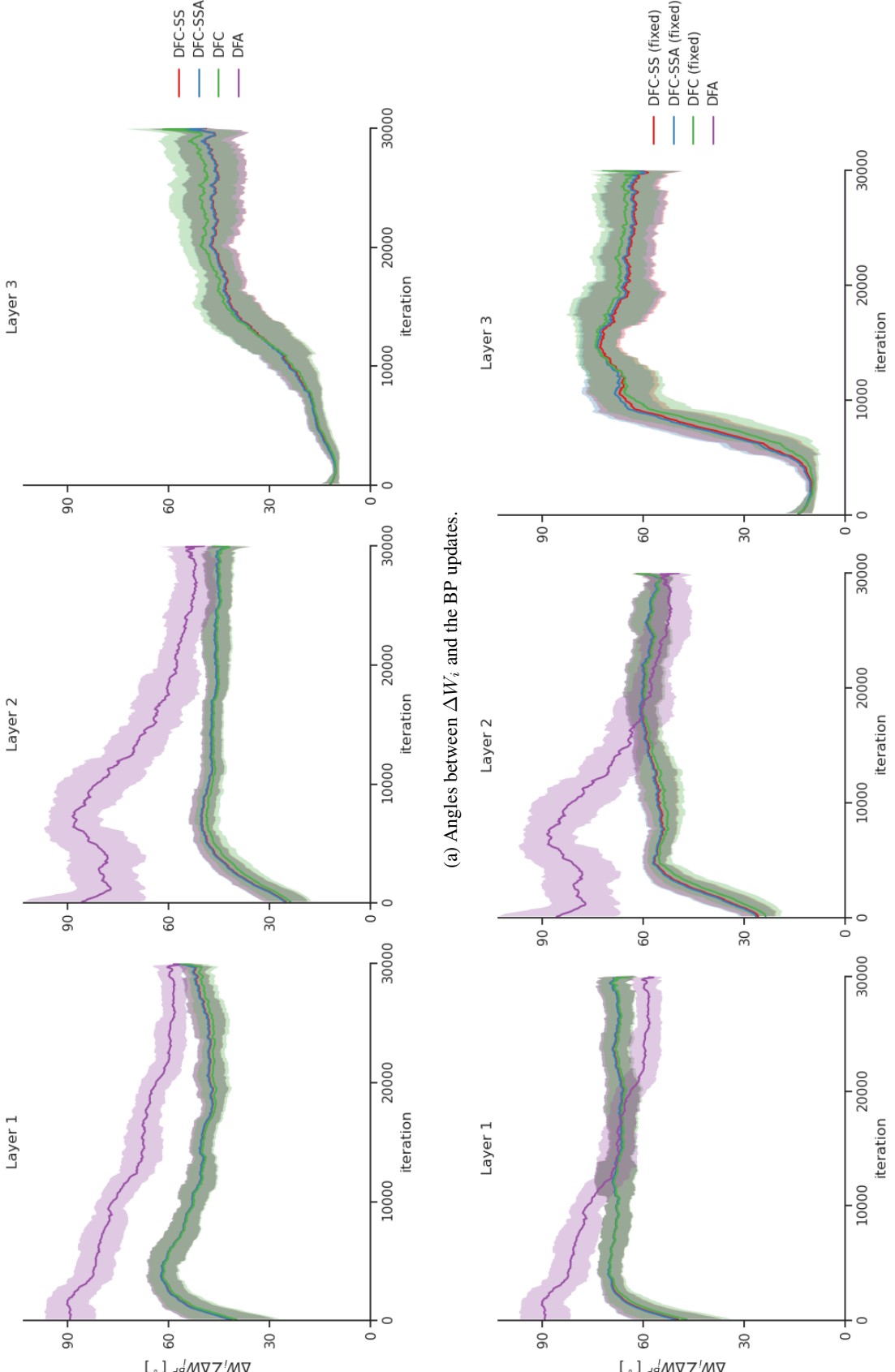

(a) Angles between $\Delta W_i$ and the BP updates.

(b) Angles between the BP updates and $\Delta W_i$ with fixed feedback weights.

Figure S7: Angles between the weight updates $\Delta W_i$ computed by BP and the ones computed by DFC-SS, DFC-SSA, DFC, and DFA, plotted for all hidden layers with (a) learned feedback weights (b) fixed feedback weights on the toy experiment explained in Figure 3. A window-average is plotted together with the window-std (shade). The x-axis iterations corresponds to the minibatches processed. The curves of DFC, DFC-SS and DFC-SSA overlap in some of the plots.

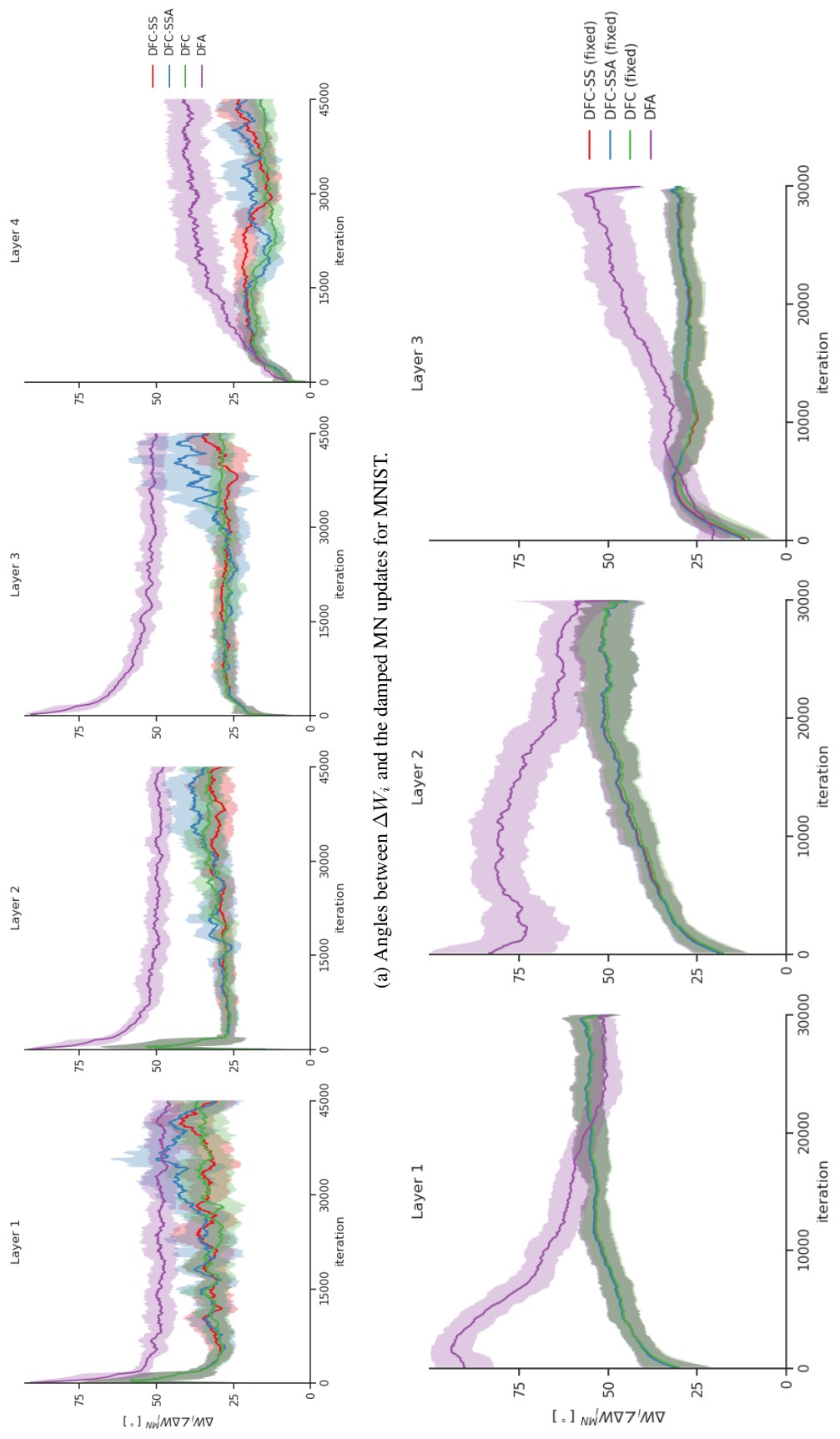

(a) Angles between $\Delta W_i$ and the damped MN updates for MNIST.

(b) Angles between $\Delta W_i$ and the damped MN updates for MNIST with fixed feedback weights.

Figure S8: Angles between the damped MN updates with $\gamma = 1$ (214) and the ones computed by DFC-SS, DFC-SSA, DFC, and DFA, plotted for all hidden layers with (a) learned feedback weights and (b) fixed feedback weights on MNIST. A window-average is plotted together with the window-std (shade). The x-axis iterations corresponds to the minibatches processed. The curves of DFC, DFC-SS, and DFC-SSA overlap in some of the plots.

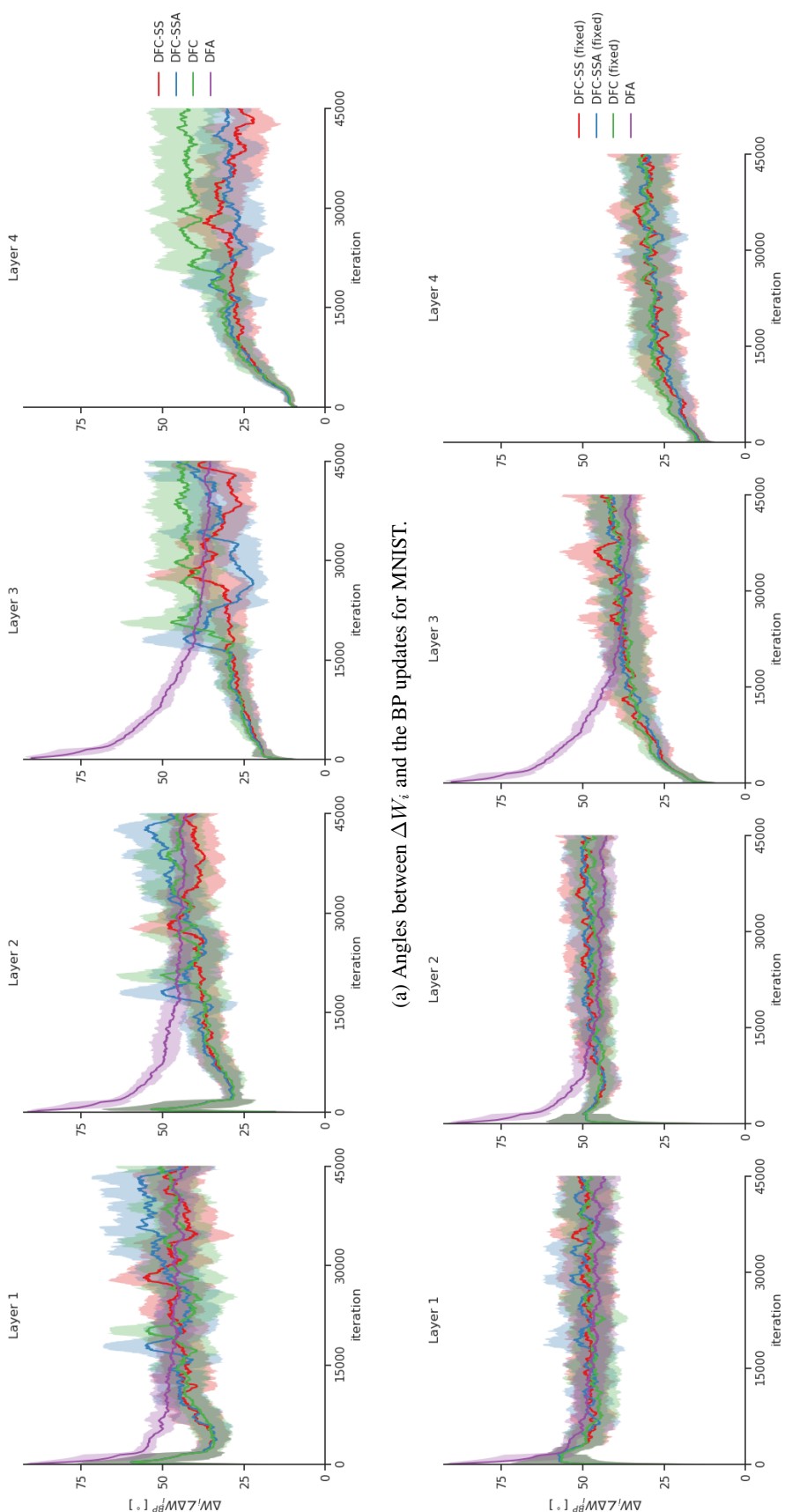

(a) Angles between $\Delta W_i$ and the BP updates for MNIST.

(b) Angles between $\Delta W_i$ and the BP updates for MNIST with fixed feedback weights.

Figure S9: Angles between the weight updates $\Delta W_i$ computed by BP and the ones computed by DFC-SS, DFC-SSA, DFC, and DFA, plotted for all hidden layers with (a) learned feedback weights (b) fixed feedback weights on MNIST. A window-average is plotted together with the window-std (shade). The curves of DFC, DFC-SS, and DFC-SSA overlap in some of the plots. The x-axis iterations corresponds to the minibatches processed.

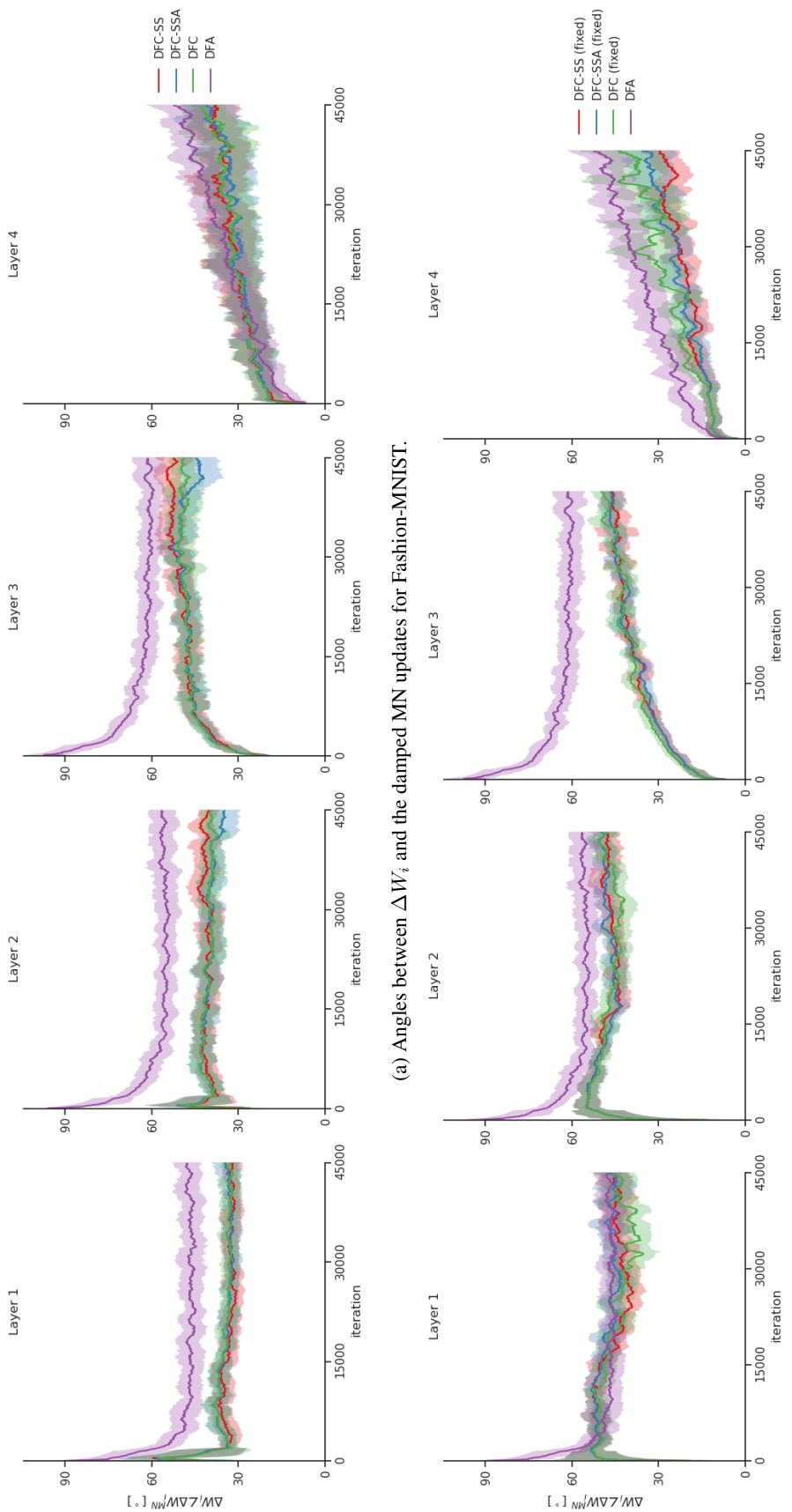

(a) Angles between $\Delta W_i$ and the damped MN updates for Fashion-MNIST.

(b) Angles between $\Delta W_i$ and the damped MN updates for Fashion-MNIST with fixed feedback weights.

Figure S10: Angles between the damped MN updates with $\gamma = 1$ (214) and the ones computed by DFC-SS, DFC-SSA, DFC, and DFA, plotted for all hidden layers with (a) learned feedback weights and (b) fixed feedback weights on Fashion-MNIST. A window-average is plotted together with the window-std (shade). The x-axis iterations corresponds to the minibatches processed. The curves of DFC, DFC-SS and DFC-SSA overlap in some of the plots.

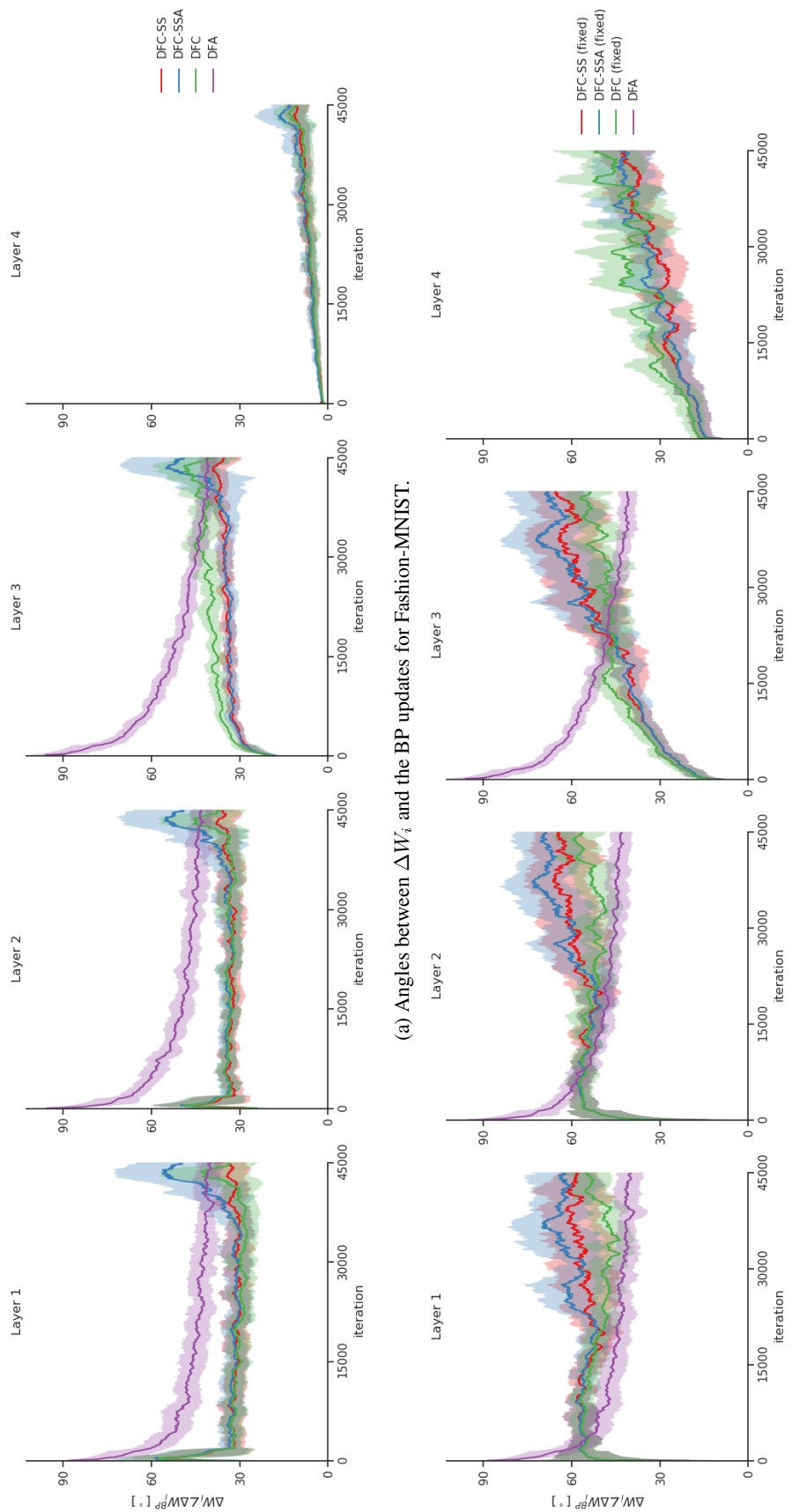

(a) Angles between $\Delta W_i$ and the BP updates for Fashion-MNIST.

(b) Angles between $\Delta W_i$ and the BP updates for Fashion-MNIST with fixed feedback weights.

Figure S11: Angles between the weight updates $\Delta W_i$ computed by BP and the ones computed by DFC-SS, DFC-SSA, DFC, and DFA, plotted for all hidden layers with (a) learned feedback weights (b) fixed feedback weights on Fashion-MNIST. A window-average is plotted together with the window-std (shade). The x-axis iterations corresponds to the minibatches processed. The curves of DFC, DFC-SS and DFC-SSA overlap in some of the plots.

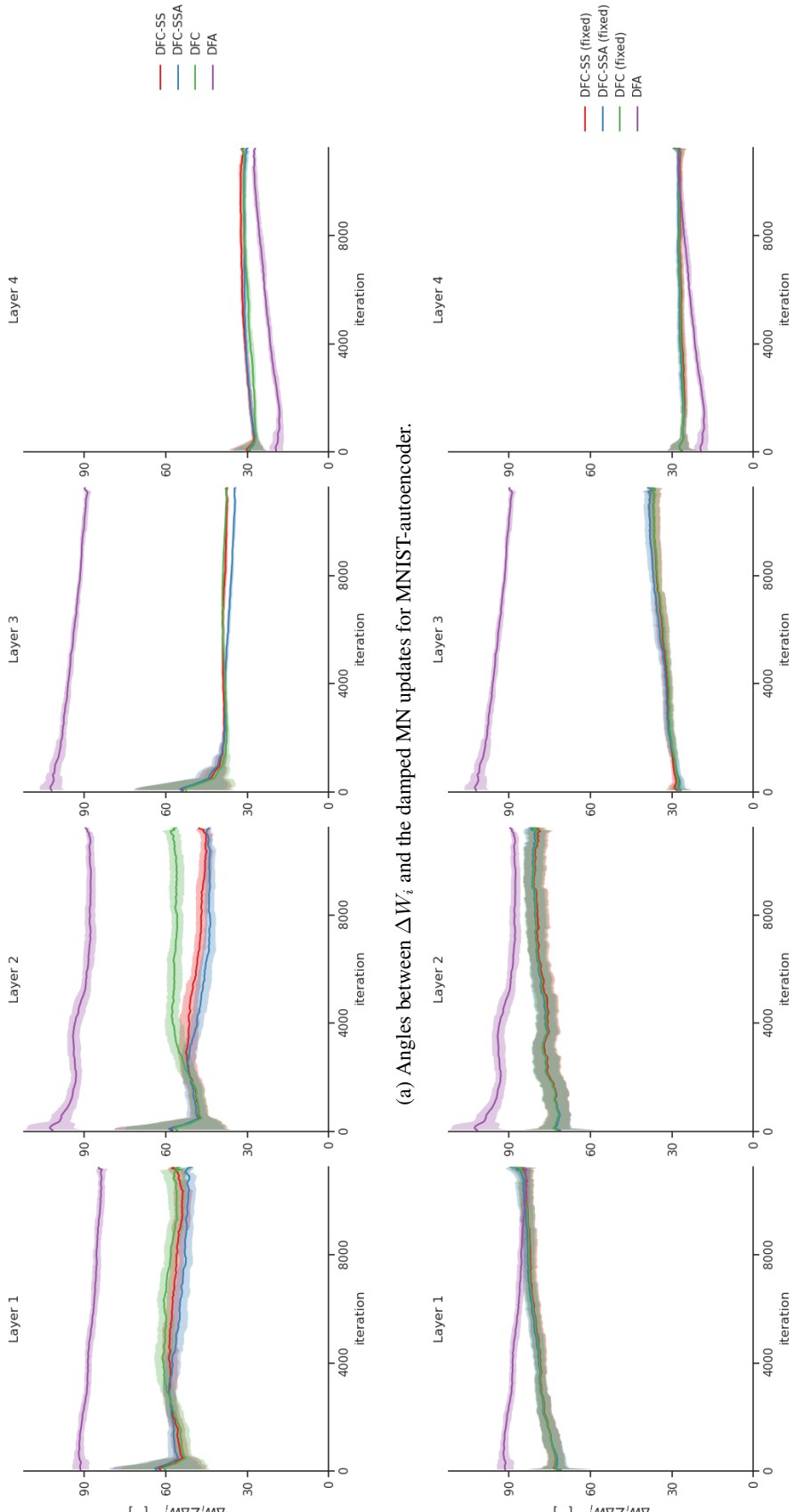

(a) Angles between $\Delta W_i$ and the damped MN updates for MNIST-autoencoder.

(b) Angles between $\Delta W_i$ and the damped MN updates for MNIST-autoencoder with fixed feedback weights.

Figure S12: Angles between the damped MN updates with $\gamma = 1$ (214) and the ones computed by DFC-SS, DFC-SSA, DFC, and DFA, plotted for all hidden layers with (a) learned feedback weights and (b) fixed feedback weights on MNIST-autoencoder. A window-average is plotted together with the window-std (shade). The x-axis iterations corresponds to the minibatches processed. The curves of DFC, DFC-SS and DFC-SSA can overlap in some of the plots.

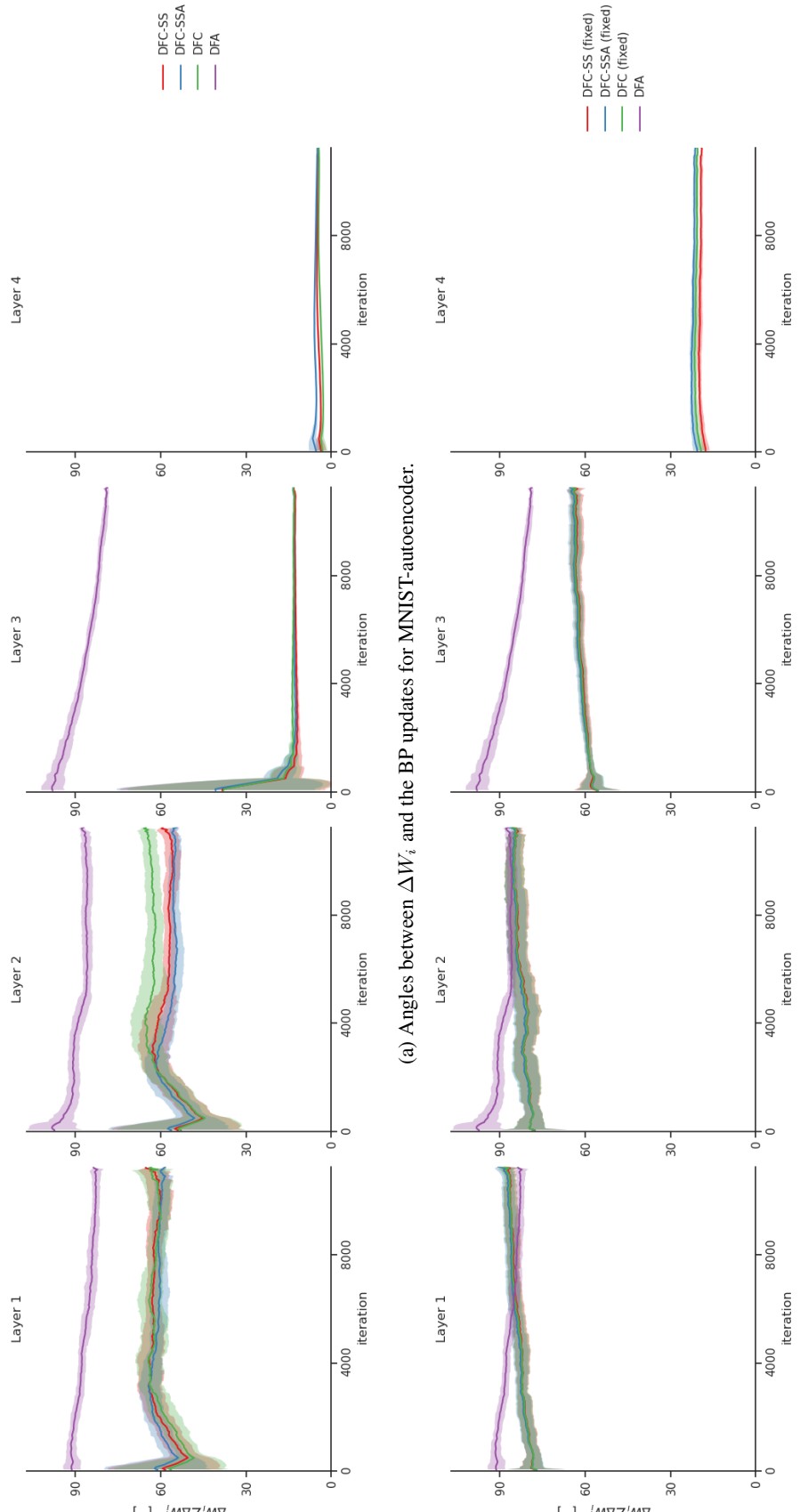

(a) Angles between $\Delta W_i$ and the BP updates for MNIST-autoencoder.

(b) Angles between $\Delta W_i$ and the BP updates for MNIST-autoencoder with fixed feedback weights.

Figure S13: Angles between the weight updates $\Delta W_i$ computed by BP and the ones computed by DFC-SS, DFC-SSA, DFC, and DFA, plotted for all hidden layers with (a) learned feedback weights (b) fixed feedback weights on MNIST-autoencoder. A window-average is plotted together with the window-std (shade). The x-axis iterations corresponds to the minibatches processed. The curves of DFC, DFC-SS and DFC-SSA can overlap in some of the plots.

### F.5.4 Autoencoder images

Fig. S14 shows the autoencoder output for randomly selected samples of BP, DFC-SSA, DFC-SSA (fixed), and DFA, compared with the autoencoder input. As DFC, DFC-SS, and DFC-SSA have very similar test losses and hence autoencoder performance, we only show the plots for DFC-SSA and DFC-SSA (fixed). Fig. S14 shows that BP and the DFC variants with trained weights have almost perfect autoencoding performance when visually inspected, while DFA and the DFC (fixed) variants do not succeed in autoencoding their inputs, which is also reflected in the performance results (see Table 1.

Figure S14: Visual representation of the autoencoder outputs for BP, DFC-SSA, DFC-SSA (fixed), and DFA, compared to the autoencoder input.

### F.6 Resources and compute

For the computer vision experiments, we used GeForce RTX 2080 and GeForce RTX 3090 GPUs. Table S6 provides runtime estimates for 1 epoch of feedforward training and 3 epochs of feedback training (if applicable) for the DFC methods, using a GeForce RTX 2080 GPU. For MNIST and Fashion-MNIST we do 100 training epochs and for MNIST-autoencoder 25 training epochs. We did hyperparameter searches of 200 samples on all datasets for DFC-SSA and DFC-SSA (fixed) and reused the hyperparameter configuration for the other DFC variants. For BP and DFA we also performed hyperparameter searches of 200 samples for all experiments, with computational costs negligible compared to DFC.

Table S6: Estimated run times in seconds per epoch (both feedforward and feedback training included) on GeForce RTX 2080 GPU for the experiments of Table 1

|  | MNIST & Fashion-MNIST | MNIST-autoencoder |
| --- | --- | --- |
| DFC | 500s | 1500s |
| DFC-SSA | 130s | 450s |
| DFC-SS | 500s | 1500s |
| DFC (fixed) | 370s | 1350s |
| DFC-SSA (fixed) | 4s | 300s |
| DFC-SS (fixed) | 370s | 1350s |

### F.7 Dataset and Code licenses

For the computer vision experiments, we used the MNIST dataset [40] and the Fashion-MNIST dataset [41], which have the following licenses:

- MNIST: `https://creativecommons.org/licenses/by-sa/3.0/`

- Fashion-MNIST:`https://opensource.org/licenses/MIT`

For the implementation of the methods, we used PyTorch [71] and built upon the codebase of Meulemans et al. [21], which have the following licenses:

# G   DFC and multi-compartment models of cortical pyramidal neurons

As mentioned in the Discussion, the multi-compartment neuron of DFC (see Fig. 1C) is closely related to recent dendritic compartment models of the cortical pyramidal neuron [23, 25, 26, 47]. In the terminology of these models, our central, feedforward, and feedback compartments, correspond to the somatic, basal dendritic, and apical dendritic compartments of pyramidal neurons. Here, we relate our network dynamics (1) in more detail to the proposed pyramidal neuron dynamics of Sacramento et al. [23]. Rephrasing their dynamics for the somatic membrane potentials of pyramidal neurons (equation (1) of Sacramento et al. [23]) with our own notation, we get

$$\tau_v \frac{\mathrm{d}}{\mathrm{d}t} \mathbf{v}_i(t) = -g_{\mathrm{lk}} \mathbf{v}_i(t) + g_{\mathrm{B}} \big( \mathbf{v}_i^{\mathrm{ff}}(t) - \mathbf{v}_i(t) \big) + g_{\mathrm{A}} \big( \mathbf{v}_i^{\mathrm{fb}}(t) - \mathbf{v}_i(t) \big) + \sigma \boldsymbol{\xi}_i(t). \tag{222}$$

Like DFC, the network is structured in multiple layers, $0 \le i \le L$, where each layer has its own dynamical equation as defined above. Basal and apical dendritic compartments ($\mathbf{v}_i^{\mathrm{ff}}$ and $\mathbf{v}_i^{\mathrm{fb}}$ resp.) of pyramidal cells are coupled towards the somatic compartment ($\mathbf{v}_i$) with fixed conductances $g_{\mathrm{B}}$ and $g_{\mathrm{A}}$, and leakage $g_{\mathrm{lk}}$. Background activity of all compartments is modeled by an independent white noise input $\boldsymbol{\xi}_i \sim \mathcal{N}(0, I)$. The dendritic compartment potentials are given in their instantaneous forms (c.f. equations (3) and (4) in Sacramento et al. [23])

$$\mathbf{v}_i^{\mathrm{ff}}(t) = W_i \phi(\mathbf{v}_{i-1}(t)) \tag{223}$$

$$\mathbf{v}_i^{\mathrm{fb}}(t) = Q_i \mathbf{u}(t) \tag{224}$$

with $W_i$ the synaptic weights of the basal dendrites, $Q_i$ the synaptic weights of the apical dendrites, $\phi$ a nonlinear activation function transforming the voltage levels to firing rates, and $\mathbf{u}$ a feedback input.

Filling the instantaneous forms of $\mathbf{v}^{\mathrm{ff}}$ and $\mathbf{v}^{\mathrm{fb}}$ into the dynamics of the somatic compartment (222), and reworking the equation, we get:

$$\tilde{\tau}_v \frac{\mathrm{d}}{\mathrm{d}t} \mathbf{v}_i(t) = -\mathbf{v}_i + \tilde{g}_{\mathrm{B}} W_i \phi(\mathbf{v}_{i-1}(t)) + \tilde{g}_{\mathrm{A}} Q_i \mathbf{u}(t) + \tilde{\sigma} \boldsymbol{\xi}_i(t), \tag{225}$$

with $\tilde{g}_{\mathrm{B}} = \frac{g_{\mathrm{B}}}{g_{\mathrm{lk}} + g_{\mathrm{B}} + g_{\mathrm{A}}}$, $\tilde{g}_{\mathrm{A}} = \frac{g_{\mathrm{A}}}{g_{\mathrm{lk}} + g_{\mathrm{B}} + g_{\mathrm{A}}}$, $\tilde{\sigma} = \frac{\sigma}{g_{\mathrm{lk}} + g_{\mathrm{B}} + g_{\mathrm{A}}}$ and $\tilde{\tau}_v = \frac{\tau_v}{g_{\mathrm{lk}} + g_{\mathrm{B}} + g_{\mathrm{A}}}$. When we absorb $\tilde{g}_{\mathrm{B}}$ and $\tilde{g}_{\mathrm{A}}$ into $W_i$ and $Q_i$, respectively, we recover the DFC network dynamics (1) with noise added. Hence, we see that not only the multi-compartment neuron model of DFC is closely related to dendritic compartment models of pyramidal neurons, but also the neuron dynamics used in DFC are intimately connected to models of cortical pyramidal neurons. What sets DFC apart from the cortical model of Sacramento et al. [23] is its unique feedback dynamics that make use of a feedback controller and lead to approximate GN optimization.

# H   Feedback pathway designs compatible with DFC

To present DFC in its most simple form, we used direct linear feedback mappings from the output controller towards all hidden layers. However, DFC is also compatible with more general feedback pathways.

Consider $\mathbf{v}_i^{\mathrm{fb}} = g_i(\mathbf{u})$ with $g_i$ a smooth mapping from the control signal $\mathbf{u}$ towards the feedback compartment of layer $i$, leading to the following network dynamics:

$$\tau_v \frac{\mathrm{d}}{\mathrm{d}t} \mathbf{v}_i(t) = -\mathbf{v}_i(t) + W_i \phi \big( \mathbf{v}_{i-1}(t) \big) + g_i \big( \mathbf{u}(t) \big) \quad 1 \le i \le L. \tag{226}$$

The feedback path $g_i$ could be for example a multilayer neural network (see Fig. S15A) and different $g_i$ could share layers (see Fig. S15B). As the output stepsize $\lambda$ is taken small in DFC, the control signal $\mathbf{u}$ will also remain small. Hence, we can take a first-order Taylor approximation of $g_i$ around $\mathbf{u} = 0$:

$$g_i(\mathbf{u}) = J_{g_i} \mathbf{u} + \mathcal{O}(\lambda^2), \tag{227}$$

with $J_{g_i} = \frac{\partial g_i(\mathbf{u})}{\partial \mathbf{u}}\big|_{\mathbf{u}=0}$. With this linear approximation and replacing $Q_i$ by $J_{g_i}$, all previous theoretical results from Section 3 hold, as they consider the limit of $\lambda \to 0$. Furthermore, the local stability results of Section 4 can be recovered by replacing $Q_i$ in Condition 3 with $J_{g_i}$ evaluated at $\mathbf{u} = \mathbf{u}_{\mathrm{ss}}$. Finally, the feedback learning results (Section 5) can be extended to this setting, by learning the synaptic strengths connecting the feedback path $g_i$ to the network layers $\mathbf{v}_i$ according to the proposed feedback learning rule (13). For small $\sigma$, $\mathbf{u}$ will remain small and hence the feedback learning rule will align $J_{g_i}$, correctly evaluated around $\mathbf{u} = 0$, with $J^T(JJ^T + \gamma I)^{-1}$.

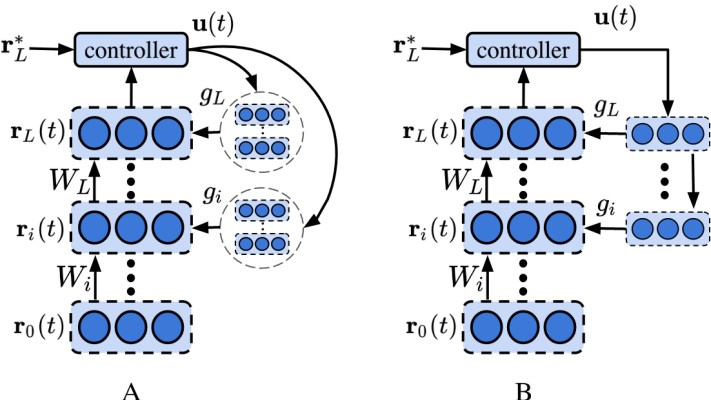

Figure S15: Schematic illustration of more general feedback paths compatible with DFC.

Until now, we considered general feedback paths $g_i$ and linearized them around $\mathbf{u} = 0$, thereby reducing their expressive power to linear mappings. As the forward Jacobian $J$ changes for each datasample in nonlinear networks, it can be helpful to have a feedback path for which $J_{g_i}$ also changes for each datasample. Then, each $J_{g_i}$ can specialize its mapping for a particular cluster of datasamples, thereby enabling a better compliance to Conditions 2 and 3 for each datasample. To let $J_{g_i}$ change depending on the considered datasample and hence activations $\mathbf{v}_i$ of the network, the feedback path $g_i$ needs to be 'influenced' by the network activations $\mathbf{v}_i$.

One interesting direction for future work is to have connections from the network layers $\mathbf{v}_i$ onto the layers of the feedback path $g_i$, that can modulate the nonlinear activation function $\phi_g$ of those layers. By modulating $\phi_g$, the feedback Jacobian $J_{g_i}$ will depend on the network activations $\mathbf{v}_i$ and, hence, will change for each datasample. Interestingly, there are many candidate mechanisms to implement such modulation in biological cortical neurons [72, 73, 74].

Another possible direction is to integrate the feedback path $g_i$ into the forward network (1) and separate forward signals from feedback signals by using neural multiplexed codes [26, 75]. As the feedback path $g_i$ is now integrated into the forward pathway, its Jacobian $J_{g_i}$ can be made dependent on the forward activations $\mathbf{v}_i$. While being a promising direction, merging the forward pathway with the feedback path is not trivial and significant future work would be needed to accomplish it.

# I Discussion on the biological plausibility of the controller

The feedback controller used by DFC (see Fig. 1A and eq. (4)) has three main components. First, it needs to have a way of computing the control error $\mathbf{e}(t)$. Second, it needs to perform a leaky integration ($\mathbf{u}^{\mathrm{int}}$) of the control error. Third, the controller needs to multiply the control error by $k_p$.

Following the majority of biologically plausible learning methods [9, 14, 15, 16, 20, 21, 22, 26, 42], we assume to have access to an output error that the feedback controller can use. As the error is a simple difference between the network output and an output target $\mathbf{r}_L^*$, this should be relatively easily computable. Another interesting aspect of computing the output error is the question of where the output target $\mathbf{r}_L^*$ could originate from in the brain. This is currently an open question in the field [76] which we do not aim to address in this work.

Integrating neural signals over long time horizons is a well-studied subject concerning many application areas, ranging from oculomotor control to maintaining information in working memory

[58, 59, 60, 61, 62]. To provide intuition, a straightforward approach to leaky integration is to use recurrent self-connections with strength $(1 - \alpha)$. Then, the same neural dynamics used in (1) give rise to

$$\tau_u \frac{\mathrm{d}}{\mathrm{d}t} \mathbf{u}^{\mathrm{int}}(t) = -\mathbf{u}^{\mathrm{int}}(t) + W_{\mathrm{in}}\mathbf{e}(t) + (1 - \alpha)\mathbf{u}^{\mathrm{int}}(t) = W_{\mathrm{in}}\mathbf{e}(t) - \alpha\mathbf{u}^{\mathrm{int}}(t). \qquad (228)$$

When we take the input weights $W_{\mathrm{in}}$ equal to the identity matrix, we recover the dynamics for $\mathbf{u}^{\mathrm{int}}(t)$ described in (4).

Finally, a multiplication of the control error by $k_p$ can simply be done by having synaptic weights with strength $k_p$.