# OpenReview forum: "Credit Assignment in Neural Networks through Deep Feedback Control"
_NeurIPS.cc/2021/Conference — NeurIPS 2021 Spotlight_

### Official Review · Reviewer_nh8b · 2021-07-01

**Rating:** 8
**Confidence:** 5

**Summary:**

The authors introduce Deep Feedback Control, an algorithm for training deep neural networks where credit assignment is done with the help of a proportional-integral-derivative (PID or rather PI in this case) controller.

Theoretically, the authors show that a learning rule similar to what they propose (but not the same), under certain assumptions, gives the Gauss-Newton update for an L2 loss, and under milder assumptions gives the minimum norm updates of W. The authors show that the PI controller helps with stability and perform some experiments to show that the assumptions remain more or less valid during training.

### After rebuttal

Because of the increased quality of the work stemming from more theoretical arguments and more experiments, I have increased my score from 5 to 8.

**Limitations And Societal Impact:**

Yes

**Main Review:**

## Originality
The novelty of the method is in the use of the PI controller which helps stabilize the learning dynamics. Otherwise, proving theorems in the limit of small nudges, similarly showing a connection with GN is not new.

## Quality
Overall this is a good paper. It is grounded in theory, guaranteeing performance in certain limits. The assumptions (but not the limits) of the theorems are more or less well explored. However it suffers from some weaknesses that if addressed can drastically improve the quality and the impact of the work. Here are a number of shortcomings in no particular order.

* The exact influence of the PI controller is not clear. It seems like Theorems 2 and 3 would hold also for purely proportional control (setting $\tau_u=0$), therefore the only effect of the integrative part is in stabilizing dynamics in stabilizing the dynamics such that $Q$ can be learned (sections 4 and 5). Given that without the integral part - that is just having just proportional feedback - brings the submission much closer to prior work, the authors should further clarify exactly where the integral part is essential, and what parts of their arguments would also hold for prior work. (The section in the appendix on related works is an attempt in this direction but still does not clarify where the differences are coming from. Is it in the architecture? In the controller? In the update rules?)

* There are some recent developments missing from the related work section and should be included (e.g  arxiv: 2011.15031).

* In lines (350-351) is is said that forward weight updates are not exactly GN etc. Similarly in Fig. 3 it is said that DFC-SSA is the ideal _damped_ MN update. First, the damped MN or GN update is nowhere defined, but I assume it refers to the fact that theorems 2,3 apply for update in Eq. (9) not the proposed update of Eq. (5). The effect of this seemingly adhoc dampening is mentioned in passing in lines 189 and expanded a little for MN optimization in the appendix, but never verified experimentally. This can be fixed by included MN and GN in Fig 3 and Tab 1.

* Similar to above, theorems 2,3 apply in the limit of $\alpha,\lambda\to0$. Unfortunately this aspect of the algorithm is never explored. In fact, the actual value of the $\lambda$ (is it actually small?) in experiments is not even found in the 45 page appendix! In biological networks, neither of these parameters can be too small. Because of noise and the limited dynamic range of neurons the nudging would have to be finite. Overall, the authors spend more time examining whether or not the conditions of their theorems are satisfied, than convincing the reader that the theorems are still approximately valid in realistic situations. If the authors spend a little bit of time exploring this, the potential impact of the paper would increase dramatically.

* Comparison to prior work (TP, DTP,  the hybrid GN-SGD of [20] etc) and non-damped GN and MN update. The authors do not answer the question of 'Does DFV perform better than the many (or even any) prior incarnations of TP?'

## Clarity

Overall the paper is clear. There are some small points.should be clarified).

* As mentioned above, the differences with prior work (especially [20]) should be further clarified. For experts actively working on this problem, the submission might look like a big departure from prior work, but for general computational neuroscience audience, it looks like a small evolution of [20,21], with many arguments seemingly applicable to it as well. A clear section, mentioning 1. what exactly is different (architecture, learning rule etc) and 2. what is the effect of each departure from previous work (whether it improves stability or is required for relationship to GN etc.).

* It seems the influence of the controller is a little exaggerated. For example in line 142-143, it is mentioned that the controller dynamically inverts the output error. Judging by Fig. 1, the controller (the box at the top of the circuit) just integrates the error. The dynamical inversion is done in the body of the circuit (even if the controller was just spitting out the error itself).


## Significance

The paper is an evolution of the TP framework. With added clarity and extra experiments it can have a not-insignificant impact.

**Time Spent Reviewing:**

6

---

> ### Author Response · Authors · 2021-08-10
> **General Response**
>
> We thank the reviewer for the thorough reading of the manuscript and proofs, for the detailed and constructive feedback, and for providing us with the opportunity to significantly improve the paper. An overarching question in the comments was the relation of DFC to previous work and, more specifically, to recent work connecting Target Propagation (TP) to Gauss-Newton (GN) optimization [20,21]. We will argue in Comment *Clarity and related work* that, although both DFC and TP are connected to GN, DFC is not merely an evolution of the TP framework because: (i) DFC combines many insights from different lines of work, not only TP; (ii) the inner workings of DFC and TP are very different; and (iii) upon closer inspection, DFC and TP are related to significantly different instantiations of GN optimization. Moving to the more specific comments and suggestions of the reviewer, we address each in a separate comment. In comment *Influence of the controller type*, we compare the effects of a PI controller versus a proportional controller on DFC and argue that both controller types lead towards significant novelty compared to prior work. In comment *Effects of the nonlinearity in the learning rule*, we discuss new experiments showing that introducing the nonlinearity $\\phi$ into the learning rule (5) for the forward weights significantly improves the performance of DFC. In comment *Robustness to* $\\lambda$ *and* $\\alpha$, we show new experiments that indicate the robustness of DFC for a wide range of $\\lambda$s and $\\alpha$s. In comment *Empirical comparison to TP*, we provide new experimental results showing that DFC performs better compared to TP variants. Finally, in comment *Originality and significance*, we conclude this author response and summarize the core contributions of DFC. With the help of the useful comments and suggestions of the reviewer, we significantly improved our paper and we hope that the reviewer finds it worthy for acceptance.

---

> > ### Author Response · Authors · 2021-08-10
> > **Influence of the controller type**
> >
> > We thank the reviewer for this helpful comment and would like to clarify the influence of the PI controller in the DFC framework. As we will discuss at the end of this comment, DFC introduces several novel insights and unique properties to the field of biologically plausible CA, without hinging on the specifics of the controller. The reviewer sharply noted that many of our theoretical results could also hold for proportional control instead of proportional integral (PI) control, which could make the DFC framework more generally applicable to different controller types. In the following, we will provide a new theoretical analysis exploring this suggestion.
> >
> > By making a first-order Taylor approximation of the network dynamics with only proportional control (putting $K_I=0$ in Eq. (4)), we obtain the following steady-state solution:
> >
> > $$
> > \\mathbf{v}_ {\\mathrm{ss}} = \\mathbf{v}_{\\mathrm{ss}}^{\\mathrm{ff}} + (QJ + \\frac{1}{k_p}  I)^{-1}Q\boldsymbol{\\delta}_L + \\mathcal{O}(\\lambda^2),
> > $$
> >
> > with $k_p$ the proportional gain, i.e., $\\mathbf{u} = k_p \\mathbf{e}$. When comparing with Eq. (6), we see that the steady-state solution with proportional control has a similar structure to the one with PI control, with $\\tilde{\alpha}$ replaced by $\\frac{1}{k_p}$. Note, however, that the damped inverse of $QJ$ is taken instead of $JQ$. By using a similar proof technique as for Lemma S2, we can show that $\\lim_{k_p\\rightarrow \\infty} (QJ + \\frac{1}{k_p}  I)^{-1}Q = J^{\\dagger}$ iff Condition 2 holds. Consequently, Theorem 2 and 3 and Proposition 4 hold also for proportional control, if the limit of $\\alpha$ to zero is replaced by the limit of $k_p$ to infinity. Furthermore, the main intuitions of Theorem 6 for training the feedback can be applied to proportional control, given that one finds a way to keep the network stable during the initial feedback weights training phase. We will include these new findings in a revised version of our paper and clarify that DFC is compatible with both proportional and PI control.
> >
> > Despite these theoretical similarities between proportional and PI control in DFC, there are some significant practical differences. First, for finite $k_p$ in proportional control, there is always a residual error that remains and hence the output target will never be exactly reached. Second, if noise is present in the network, it gets amplified by the same factor $k_p$. Hence, using a high $k_p$ in proportional control makes the controlled network sensitive to noise. Adding an integral control component can alleviate these issues by replacing the need for a large gain, $k_p$, with the need for a good integrator circuit (i.e., low $\\alpha$) [33], for which a rich neuroscience literature exists [73, 74, 75, 76, 77]. This way, we can use a smaller gain, $k_p$, without increasing the residual error and consequently make the network less sensitive to noise. This is also interesting from a biological point of view since biological networks are considered to be substantially noisy.
> >
> > Further, we highlight that the major novel contributions of DFC do not depend on whether proportional or PI control is used. Although the connection between learning with inverses and GN optimization was already made before, Theorems 2 and 3 are novel and unique to DFC and do not hold for previous work on TP [20,21] (see Comment *Clarity and related work* for a detailed discussion). Furthermore, the proportional control methods to train non-hierarchical RNNs [27,28,29,30] have so far only been used to train single-layer RNNs with fixed output and feedback weights. DFC introduces the 'learning through control approach' to DNNs by identifying flexible conditions on the feedback weights needed for principled CA (Condition 2) and introducing a feedback learning rule to adjust the feedback weights during training, which is crucial for DNNs. We note that our theoretical results connecting the 'learning through control approach' to GN and the theoretical results on the feedback weight learning rule, using stochastic differential equations, are also novel to the field. In summary, CA in DNNs through controlling neural activations is a novel contribution of DFC, both for a PI controller and proportional controller.

---

> > > ### Author Response · Authors · 2021-08-10
> > > **Effects of the nonlinearity in the learning rule**
> > >
> > > We thank the reviewer for this helpful comment. There are three main reasons why the DFC weight updates are not exactly equal to MN updates. **First**, we use the nonlinearity $\\phi$ in the DFC learning rule (5), hereafter referred to as the *nonlinear learning rule*, in contrast to the *linear learning rule* in Theorem 2 and 3 (Eq. (9)). **Second**, $\\alpha$ and $\\lambda$ are not infinitesimal in practice. **Third**, the feedback weights do not exactly fulfill Condition 2 for each datasample, due to limited training iterations and limited capacity of the linear feedback mappings. Because we are interested in modelling the learning behavior of DFC as close as possible, we crudely model these effects by adding a damping constant to the pseudoinverse of $J$ in the MN updates against which we compute the alignment of DFC (see Eq. (211) and (212) and lines 1200-1221 for a discussion). To avoid introducing too many complexities in the main paper that might distract the reader, we recomputed and updated the alignment measures in Fig. 3 now with the undamped MN and GN updates, corresponding to Theorem 3 and 2, to show how well DFC approximates the exact theorems (said figures will be included in the revised paper and can be shared in an anonymous drive upon request of the reviewer). The same conclusions on the figure remain valid (lines 290-309).
> > >
> > > Encouraged by the reviewer's comment, we also studied in detail the experimental consequences of using the nonlinear learning rule (5) instead of the linear learning rule (9). We adopt the following approach: first, we show that when the assumptions of Theorem 3 are perfectly satisfied, the nonlinear learning rule (5) has no significant benefit over the linear learning rule (9). Second, we show that in the realistic case of DFC when the assumptions are not satisfied perfectly, the nonlinear learning rule has a significant benefit in performance over the linear learning rule. These results, which we will include in the revised paper, highlight that adding the appropriate nonlinearities to the learning rule helps compensating for imperfect conditions of DFC.
> > >
> > > We start with the ideal case where Condition 2 is perfectly satisfied and in the limit of $\\lambda$ and $\\alpha$ to zero, which results in MN updates (211) if the linear learning rule is used and in the following update when the nonlinear learning rule is used:
> > >
> > > $$
> > >     \\Delta \\bar{W} = RDJ^T(JJ^T)^{-1}\\boldsymbol{\\delta}_L,
> > > $$
> > >
> > > with $D$ a diagonal matrix with $\\partial \\phi(v_j) / \\partial (v_j)$ for each neuron in the network on its diagonal and $R$ as defined in Eq. (211). To show that in this ideal case, the nonlinear learning rule has no significant benefit, we performed a new MNIST experiment comparing the linear and nonlinear learning rule in this ideal case, resulting in a test error of $2.18^{\\pm 0.14}\\%$ and $2.11^{\\pm 0.10}\\%$, respectively.
> > >
> > > Next, we investigate the influence of the nonlinear learning rule in the practical case when Condition 2 is not exactly satisfied. We performed a new hyperparameter search on MNIST for DFC-SSA, now with the linear learning rule (9), resulting in a test error of $5.28^{\\pm 0.14}\\%$. Comparing this with the corresponding test performance in Table 1 ($2.29^{\\pm 0.097}\\%$ test error), we see that introducing the nonlinearity in the weight update improves results significantly. Hence, the main effect on the performance of the nonlinear learning rule is to compensate for the feedback weights, which do not perfectly satisfy Condition 2. This result experimentally validates the introduction of the chosen nonlinearities in the learning rule (5) for DFC.
> > >
> > > Finally, to investigate where this performance gap originates from, we performed another toy experiment similar to Fig. 3 for the linear versus nonlinear learning rule in DFC. The new results show that the updates resulting from the nonlinear learning rule are much better aligned with the undamped MN and GN updates, compared to the linear learning rule, explaining its better performance (said figures will be included in the revised paper and can be shared in an anonymous drive upon request of the reviewer). Overall, we conclude that introducing the nonlinearity in the learning rule, which prevents saturated neurons from updating their weights, is a useful heuristic to improve the alignment of DFC with the undamped MN and GN updates and consequently improve its performance, when Condition 2 is not perfectly satisfied. We will add these new insights to the revised paper accordingly.

---

> > > > ### Author Response · Authors · 2021-08-10
> > > > **Robustness to lambda and alpha**
> > > >
> > > > We thank the reviewer for this great suggestion to improve the completeness of our paper. To address the question of how DFC behaves when $\\alpha$ and $\\gamma$ are not in the limit to zero, we performed new experiments on MNIST while varying the values of $\\alpha$ and $\\lambda$. These new results show that the performance of DFC and DFC-SS is robust to a wide range of values for $\\alpha$ and $\\lambda$ and are contained in the two tables below (test error on MNIST with variable $\\alpha$ and $\\lambda$).
> > > >
> > > > Classification error on MNIST with variable $\\lambda$ values and $\\alpha = 0.0015$.
> > > >
> > > >
> > > > | $\\lambda$ |         DFC-SS         |          DFC          |
> > > > |:----------:|:----------------------:|:---------------------:|
> > > > |  $1e^{-3}$ |  $2.21^{\\pm 0.03}\\%$ | $2.26^{\\pm 0.06}\\%$ |
> > > > |  $1e^{-2}$ |  $2.28^{\\pm 0.10}\\%$ | $2.20^{\\pm 0.08}\\%$ |
> > > > |  $1e^{-1}$ |  $2.30^{\\pm 0.13}\\%$ | $2.30^{\\pm 0.10}\\%$ |
> > > > |  $0.5$       |  $2.22^{\\pm 0.10}\\%$ | $2.37^{\\pm 0.17}\\%$ |
> > > >
> > > > Classification error on MNIST with variable $\\alpha$ values and $\\lambda=0.08$.
> > > >
> > > > | $\\alpha$ |         DFC-SS        |          DFC          |
> > > > |:---------:|:---------------------:|:---------------------:|
> > > > | $1e^{-4}$ | $2.31^{\\pm 0.12}\\%$ | $2.28^{\\pm 0.06}\\%$ |
> > > > | $1e^{-3}$ | $2.28^{\\pm 0.15}\\%$ | $2.31^{\\pm 0.11}\\%$ |
> > > > | $1e^{-2}$ | $2.26^{\\pm 0.05}\\%$ | $2.32^{\\pm 0.12}\\%$ |
> > > > | $1e^{-1}$ | $2.28^{\\pm 0.11}\\%$ | $2.34^{\\pm 0.16}\\%$ |
> > > >
> > > > For reference, we also provide in the table below the values of $\\lambda$ and $\\alpha$ that were used for the DFC results of Table 1, which we will include in the revised paper, complementary to the hyperparameter search intervals that were already included in App E.4.
> > > >
> > > > Values of $\\lambda$ and $\\alpha$ used for Table 1. Note that for DFC, DFC-SS, and DFC-SSA these values are the same.
> > > >
> > > > |            |     MNIST    | Fashion-MNIST | MNIST autoencoder | MNIST (train loss) |
> > > > |:----------:|:------------:|:-------------:|:-----------------:|:------------------:|
> > > > |  $\\lambda$ | $7.79e^{-2}$ |  $5.61e^{-2}$ |     $2.03e^{-2}$   |    $1.45e^{-2}$    |
> > > > | $\\alpha$ | $1.39e^{-3}$ |  $2.06e^{-4}$ |    $3.69e^{-2}$   |    $8.21e^{-3}$    |
> > > >
> > > > To investigate how robustly the theory describes DFC for varying $\\lambda$ and $\\alpha$, we performed new toy experiments for different values of $\\lambda$ and $\\alpha$ and created figures that show the alignment with GN and MN (said figures will be included in the revised paper and can be shared in an anonymous drive upon request of the reviewer). The figures show that the alignment measures remain the same for a wide range of values for $\\alpha \\in [10^{-4}: 10^{-1}]$, highlighting that our theory explains the behavior of DFC robustly when the limit of $\\alpha$ to zero does not hold. We also observe that the alignment of DFC with MN and GN remains robust for realistic values of $\\lambda \\in [0.001-0.1]$ in this toy example, but decreases when we perform a strong nudging of the output target, i.e., putting the output target $\\mathbf{r}_L^*$ equal to the output label, corresponding to $\\lambda=0.5$. When considering this new regime of 'strong nudging', i.e., $\\lambda=0.5$, the decrease in alignment can be explained by the fact that: (i) the linearization of the network used in Lemma 1 to describe the steady-state solution and consequently the post-synaptic plasticity signals becomes inaccurate in this strong nudging regime; and (ii) the neural activities change significantly during the feedback phase of DFC in this strong nudging regime, thereby changing the pre-synaptic plasticity signals compared to GN and our theory. Note, however, that our results on MNIST show robust performance even for big $\\lambda$s, suggesting that DFC can also provide principled CA in this new regime of strong nudging. In summary, experimental results suggest that DFC performs principled CA for a wide range of $\\alpha$s and for both soft nudging and strong nudging of the output. Our current theory describes the DFC well for a wide range of $\\alpha$s and modest values of $\\lambda$, and we are currently working on a follow-up project addressing a new theory that describes the strong nudging regime in more detail. We will include the new findings of this comment appropriately in the revised paper.

---

> > > > > ### Author Response · Authors · 2021-08-10
> > > > > **Recent literature developments**
> > > > >
> > > > > We thank the reviewer for pointing out this recent development related to our work (arxiv: 2011.15031), which we will of course include in the revised paper. We will also proceed with another thorough literature review so that, if new papers were published in the meantime, we can include any additional developments related to this work.

---

> > > > > > ### Author Response · Authors · 2021-08-10
> > > > > > **Empirical comparison to TP**
> > > > > >
> > > > > > Thank you for this suggestion to expand the experimental results of our work. We performed new experiments evaluating DTP [19], and DDTP-linear [20] (the best performing variant of TP in [20]) on MNIST and MNIST-autoencoder, for the same architectures as used for Table 1. The following table presents the results (test error).
> > > > > >
> > > > > > The classification error and loss on MNIST and MNIST autoencoder, respectively, for DTP and DDTP-linear.
> > > > > >
> > > > > > |             |         MNIST         |          MNIST autoencoder         |
> > > > > > |:-----------:|:---------------------:|:----------------------------------:|
> > > > > > |     DTP     | $2.61^{\\pm 0.13}\\%$ | $22.36^{\\pm 0.59} \\cdot 10^{-2}$ |
> > > > > > | DDTP-linear | $2.22^{\\pm 0.22}\\%$ | $14.60^{\\pm 0.10} \\cdot 10^{-2}$ |
> > > > > >
> > > > > > Comparing these results to the ones in Table 1, we see that DFC outperforms DTP on both datasets and DDTP-linear on MNIST autoencoder. These encouraging results suggest that the closer connection of DFC to GN, when compared to the one of DDTP-linear to GN (see comment *Clarity and related work*), leads to practical improvements in performance. We will add these results to the revised paper and complement them with results on Fashion MNIST and MNIST (train).

---

> > > > > > > ### Author Response · Authors · 2021-08-10
> > > > > > > **Clarity and related work**
> > > > > > >
> > > > > > > Thank you for these valuable suggestions which helped to improve the clarity of the paper. The dynamical inversion is indeed the result of the interplay between the network dynamics and a feedback controller, which we will clarify in the revised paper. Upon request of the reviewer, we wrote a more detailed related work section comparing DFC with TP and its variants. We will include this in the revised paper.
> > > > > > >
> > > > > > > Recent work [20,21] discovered that learning through inverses of the forward pathway can lead in certain cases to an approximation of Gauss-Newton (GN) optimization. Although this finding inspired our theoretical results on the CA capabilities of DFC, there are fundamental differences between DFC and TP, indicating that DFC is more than a mere evolution of the TP framework. The main conceptual difference between DFC and the variants of TP [18-21] is that DFC uses the combination of network dynamics and a controller to dynamically invert the forward pathway for CA, whereas TP and its variants learn parametric inverses of the forward pathway, encoded in the feedback weights. Although dynamic and parametric inversion seem closely related, they lead to major methodological and theoretical differences.
> > > > > > >
> > > > > > >
> > > > > > > ### Methodological differences between DFC and TP.
> > > > > > >
> > > > > > > **First**, for TP and its variants, the task of approximating the inverse of the forward pathway is completely put onto the feedback weights, resulting in a need for a strict relation between the feedforward and feedback pathway at all times during training. DFC, in contrast, reuses the forward pathway to dynamically compute its inverse, resulting in a more flexible relation between the feedforward and feedback pathway, described by Condition 2. To the best of our knowledge, DFC is the first method that approximates a principled optimization method for feedforward neural networks of arbitrary dimensions, compatible with a wide range of feedback connectivity. The recent work of Bengio [21] iteratively improves the inverse and, hence, can compensate for imperfect parametric inverses. However, this method is developed only for invertible networks, which require all layers to have equal dimensions.
> > > > > > >
> > > > > > > **Second**, DFC drives the neural activations to target values simultaneously, hence letting 'target activations' from upstream layers influence 'target activations' from downstream layers. TP, in contrast, computes each target as a (pseudo)inverse of the output target independently. This is a subtle yet important difference between DFC and TP, which leads to significant theoretical differences, on which we will expand later. To gain intuition, consider the case where we update the weights of both DFC and TP to reach exactly the local layer targets. In TP, if we update the weights of a hidden layer to reach its target, all downstream layers will also reach their target without updating the weights. Hence, if we update all weights simultaneously, the output will overshoot its target. DFC, in contrast, takes the effect of the updated target values of upstream layers already into account, hence, when all weight updates are done simultaneously, the output target is reached exactly (in the linearized dynamics, c.f. Theorem 3).
> > > > > > >
> > > > > > > **Third**, DFC needs significantly less external coordination compared to the recent TP variants. The new variants of TP with a link to GN [20] need highly coordinated noise phases for computing the Difference Reconstruction Loss (one separated noise phase for each layer). For DTP [19], a similar coordination is needed if noisy activations are used for computing the reconstruction loss, as proposed by the authors. The iterative variant of TP [21] needs coordination in propagating the target values, as the target iterations for a layer can only start when the iterations of the downstream layer have converged. As DFC uses dynamic inversion instead of parametric inversion, possible learning rules for the feedback weights do not need to use the Difference Reconstruction Loss [20] or variants thereof, opening the route to alternative, more biologically realistic learning rules. We propose a first feedback learning rule compatible with DFC, that makes use of noise and Hebbian learning, without the need for extensive external coordination (see also App. C.3 that merges feedforward and feedback weight training in a single phase).
> > > > > > >
> > > > > > > **Finally**, DFC uses a multi-compartment neuron model closely corresponding to recent models of cortical pyramidal neurons, to obtain plasticity rules fully local in space and time. Presently, it is unclear whether there exist similar neuron and network models for TP that result in plasticity rules local in time.
> > > > > > >
> > > > > > > ### Theoretical differences between DFC and TP.
> > > > > > >
> > > > > > > **First**, computing layerwise inverses, as is done in TP [18], DTP [19], and iterative TP [21], can only be linked to GN for invertible networks but breaks down for non-invertible networks, as shown by Meulemans et al. [20]. Both DFC and the DRL variants of TP [20] establish a link to GN for both invertible and non-invertible feedforward networks of arbitrary dimensions. However, the DRL variants of TP are linked to a hybrid version of GN and gradient descent, whereas DFC, under appropriate conditions, is linked to pure GN optimization on the parameters. Our Theorems 2 and 3 cannot be applied to the DRL variants of TP [20] due to the fact that: (i) the DRL variants compute targets for the post-nonlinearity activations and the DFC target activations, $\\mathbf{v}_i$, are pre-nonlinearity activations; and (ii) the DRL variants compute the targets for each layer independently, whereas DFC dynamically computes the targets while taking into account the changed target activations of other layers. We continue with expanding on this second point.
> > > > > > >
> > > > > > > As explained intuitively before, TP and its variants compute each layer target independently from the other layer targets. Consequently, to link their variants of TP to GN optimization, Meulemans et al. [20] and Bengio [21] need to make a block-diagonal approximation of the GN curvature matrix, with each block corresponding to a single layer. As off-diagonal blocks are put to zero, influences of upstream target values on the downstream targets are ignored. The block-diagonal approximation of the GN curvature matrix was proposed in studies that used GN optimization to train deep neural networks with big minibatch sizes [59,60]. However, similar to DFC, TP is connected to GN with a minibatch size of 1. In this case, the GN curvature matrix is of low rank, and a block-diagonal approximation of this matrix will change its rank and hence its properties. In the analysis of DFC, in contrast, we do not need to make this block-diagonal approximation, as the target activations, $\\mathbf{v}_i$, influence each other. Consequently, DFC has a closer connection to GN optimization than the TP variants [20,21].
> > > > > > >
> > > > > > > **Finally**, DFC does not use a reconstruction loss to train the feedback weights but instead uses noise and Hebbian learning. Our Theorem 6 that analyzes the behavior of this learning rule using stochastic differential equations is also a novel theoretical contribution to the field, to the best of our knowledge.

---

> > > > > > > > ### Author Response · Authors · 2021-08-10
> > > > > > > > **Originality and Significance**
> > > > > > > >
> > > > > > > > In summary, we argue that DFC merges various insights from different fields resulting in a novel biologically plausible CA technique with unique and interesting properties that transcend the sheer sum of its parts. Consequently, we do not see DFC as a mere evolution of the TP framework. Although DFC has the conceptual similarity to TP of using inverses for CA and both methods can be linked to GN optimization, the inner workings of DFC and TP are completely different, and upon closer inspection, both methods are linked to significantly different instantiations of GN optimization and CA in general. To clarify the novelty of our work, we summarize here again the core contributions of DFC.
> > > > > > > > - DFC extends the idea of using a feedback controller to adjust network activations to track a desired output target for providing CA to DNNs, opening a new route for designing principled CA methods for DNNs.
> > > > > > > > - To the best of our knowledge, DFC is the first method that approximates a principled optimization method for feedforward neural networks of arbitrary dimensions, while allowing for a wide range of feedback connectivity, in contrast to a single allowed feedback configuration.
> > > > > > > > - The learning rules of DFC for the forward and feedback weights are fully local in time and space, in contrast to many other biologically plausible learning rules. Furthermore, DFC does not need highly specific connectivity motives nor tightly coordinated plasticity mechanisms and can have all weights plastic simultaneously, if the adaptations explained in appendix C.3 are used.
> > > > > > > > - The multi-compartment neuron model needed for DFC naturally corresponds to recent multi-compartment models of pyramidal neurons.
> > > > > > > >
> > > > > > > > Finally, we hope that our comments sufficiently addressed the main concerns of the reviewer and that they clarified that DFC is a novel and significant contribution to the field.

---

> > ### Comment · Reviewer_nh8b · 2021-08-11
> > **Post-rebuttal comment**
> >
> > I thank the authors for their detailed response. Because of the increased quality of the work stemming from more theoretical arguments and more experiments, I have increased my score from 5 to 8.

---

### Official Review · Reviewer_f7V6 · 2021-07-16

**Rating:** 7
**Confidence:** 2

**Summary:**

This paper proposes a deep feedback control (DFC) method for deep networks, which is biological plausibility, and the control signal can be used for credit assignment. Experiments on MNIST and Fashion MNIST dataset demonstrate the effectiveness of DFC, and the performances is competitive to back-propagation based learning algorithms.

**Ethical Concerns:**

No.

**Limitations And Societal Impact:**

Yes.

**Main Review:**

The proposed deep feedback control approach for network learning is interesting and conveys innovate ideas both theoretically and algorithmically. The DFC method can be implemented with local learning, which is biological plausibility. It is also interesting to ask what are the possible applications of DFC based networks? Can the proposed model help explain biological mechanisms in the brain?

=============================


Post-rebuttal:

The authors have addressed most of my concerns. I have increased my score from 6 to 7.



**Time Spent Reviewing:**

3

---

> ### Author Response · Authors · 2021-08-10
> **General Response**
>
> We thank the reviewer for the feedback and the interest in this work's applications and its connection to biological mechanisms in the brain. Regarding the possible applications of DFC-based networks, the DFC framework can present a suitable, efficient family of algorithms for neuromorphic hardware and the authors are already working on adaptations of DFC to this type of bio-inspired hardware. On the other hand, regarding its connection to biological mechanism in the brain, we would like to point out the following interesting biological properties of our model: it allows for learning the synapses with a temporal and spatial local learning mechanism based solely on neuronal activity; its dynamics allows for the feedback to influence the neuronal activities; and it has an intriguing natural connection to recently developed multi-compartment models of cortical pyramidal neurons [22,24,25]. Whether specific aspects of DFC can be used to explain biological phenomena, currently unexplained by other models, is an exciting direction for future work.
>
> Again, we thank the reviewer for the helpful feedback and we are already excited to continue to work on the biological implications and predictions of this model beyond the current work.

---

> > ### Comment · Reviewer_f7V6 · 2021-08-31
> > **post-rebuttal comment**
> >
> > Thanks for the response. The authors have addressed most of my concerns. I have increased my score from 6 to 7.

---

### Official Review · Reviewer_Syop · 2021-07-20

**Rating:** 9
**Confidence:** 3

**Summary:**

This paper builds on recent insights in applying adaptive control theory, Gauss-Newton optimization, target propagation, and dynamical inversion methods to the issue of local and bio-plausible credit assignment in neural networks. The authors propose deep feedback control (DFC) as a principled approach to learning feedforward and feedback weights via local plasticity rules to approximate Gauss-Newton optimization of an output loss function for a neural network.

**Limitations And Societal Impact:**

Yes

**Main Review:**

This is a masterful paper that lucidly combines several recent insights towards solving the issue of credit assignment via local plasticity rules in the brain and neuromorphic hardware. A number of theorems/propositions are proved along the way to showing the conditions and approximations needed. (I have not checked the proofs, but the theorems and the conditions appear plausible.) A bio-plausible three-compartment neural implementation is also presented. The method is tested on simple benchmarks and shows good performance compared to backpropagation. I expect this work to open new directions as an alternative to approximating backpropagation to obtain brain-like learning rules.

Minor:
lines 5 and 56: motives -> motifs.

**Time Spent Reviewing:**

2.5

---

> ### Author Response · Authors · 2021-08-10
> **General Response**
>
> We thank the reviewer for the encouraging review and look forward to work on the possible directions of this work that can lead to other new biologically plausible learning rules for DNNs.

---

### Decision · Program_Chairs · 2021-09-27

**Decision:**

Accept (Spotlight)

**Comment:**

The paper introduces a new framework for biological plausible credit assignment algorithms in neural networks. The main idea consists in sending a top-down 'feedback' signal that drives the network to output the desired values, and update weights to make the feedback correction unneeded. They show a connection between the resulting algorithm and Gauss-Newton optimization. The work has connection to previous biological CA literature, in particular work on target propagation and feedback alignment.
Reviewers thought the contribution was insightful, and well written. Initial reviews brought about some concerns regarding novelty and what the precise contributions of some components were (whether they were important or needed at all, like the PI controller), which were extensively discussed in the rebuttal period. Please adjust the final version of the paper by taking these discussions into account.
A final improvement could include a more formal, clear theorem connecting DFC to GN: at the moment, it feels that one needs to connect theorem 1 and 2 to fully get the connection between the two. The relation between DFC and GN would ideally be made clear and rigorous in a single theorem, without having to connect dots.